# Endoplasmic reticulum disruption stimulates nuclear membrane mechanotransduction

Zhouyang Shen ®[1,2,3,4], Zaza Gelashvili ®[1,2,4] & Philipp Niethammer ®[1] ✉

Cytosolic phospholipase A2 (cPLA$_2$) controls some of the most powerful inflammatory lipids in vertebrates by releasing their metabolic precursor, arachidonic acid, from the inner nuclear membrane (INM). Ca$^{2+}$ and INM tension (T$_{INM}$) are thought to govern the interactions and activity of cPLA$_2$ at the INM. However, as compensatory membrane flow from the contiguous endoplasmic reticulum (ER) may prevent T$_{INM}$, the conditions permitting nuclear membrane mechanotransduction by cPLA$_2$ or other mediators remain unclear. To test whether the ER buffers T$_{INM}$, we created the genetically encoded, Ca$^{2+}$-insensitive T$_{INM}$ biosensor amphipathic lipid-packing domain inside the nucleus (ALPIN). Confocal time-lapse imaging of ALPIN– or cPLA$_2$–INM interactions, along with ER morphology, nuclear shape/volume and cell lysis revealed a link between T$_{INM}$ and disrupted ER–nuclear membrane contiguity in osmotically or ferroptotically stressed mammalian cells and at zebrafish wound margins in vivo. By combining ALPIN imaging with Ca$^{2+}$-induced ER disruption, we reveal the causality of this correlation, which suggests that compensatory membrane flow from the ER buffers T$_{INM}$ without preventing it. Besides consolidating the biomechanical basis of cPLA$_2$ activation by nuclear deformation, our results identify cell stress- and cell death-induced ER disruption as an additional nuclear membrane mechanotransduction trigger.

Cells constantly reshape and hence must tell innocuous shape changes from deformations that endanger their integrity. Plasma membrane mechanotransduction is tuned for sensitivity: stretch-activated channels, integrin adhesions and other mechanisms sense minute strains at the cell body surface and its periphery. By contrast, nuclear membrane mechanotransduction (NMMT) engages when the cell body is subjected to severe confinement, stark osmotic swelling or other stresses that pressurize and stretch the nuclear envelope (NE)[1]. In other words, the nuclear membrane is tailored to detect bulk deformations that may threaten cell and tissue viability and call for an extension of its adaptive range, emergency repair or inflammatory signalling. One of the most powerful and versatile bioactive lipid signalling mechanisms involved in all these aspects, the eicosanoid cascade[2], is controlled by NMMT. Specifically, inner nuclear membrane tension (T$_{INM}$)-dependent cytosolic phospholipase A2 (cPLA$_2$) activation leads to arachidonic acid release from the NE. Arachidonic acid-derived prostaglandins, leukotrienes, oxoeicosanoids and other bioactive lipids control diverse physiological processes ranging from inflammation to musculoskeletal adaptation[3,4].

[1]Cell Biology Program, Memorial Sloan Kettering Cancer Center, New York, NY, USA. [2]Gerstner Sloan Kettering Graduate School of Biomedical Sciences, New York, NY, USA. [3]Present address: Bloomberg-Kimmel Institute for Cancer Immunotherapy, Department of Oncology, Johns Hopkins University School of Medicine, Baltimore, MD, USA. [4]These authors contributed equally: Zhouyang Shen, Zaza Gelashvili. ✉e-mail: niethamp@mskcc.org

The cPLA$_2$–NMMT pathway has been reported to control the osmotic surveillance of epithelial barrier integrity by leukocytes; mitotic entry; confined migration of embryonic, immune and cancer cells; dendritic cell chemotaxis; and cell differentiation[5–10]. Via conserved hydrophobic residues in its Ca$^{2+}$-dependent membrane-binding C2 domain and its catalytic site, cPLA$_2$ adsorbs to T$_{INM}$-induced lipid-packing defects (LPDs)[11]. Notably, a similar (that is, LPD) mechanism mediates curvature sensing by amphipathic lipid-packing sensor (ALPS) domains, for example, of ARFGAP1, on nanometre-sized vesicles[12,13]. Although the concept of LPD-dependent mechanosensing is well established[14–16] and has been thoroughly explored in the context of membrane curvature sensing[13,17,18], its physiological relevance in the context of NMMT is just emerging.

NMMT assumes that strong nuclear deformation causes T$_{INM}$, but this premise has been questioned. The endoplasmic reticulum (ER) is contiguous with the nuclear membrane, forming a large membrane reservoir. Before T$_{INM}$ can expose a hydrophobic binding surface, nuclear membrane invaginations and evaginations must be unfolded. If a contiguous ER were to behave like a large nuclear membrane reservoir, T$_{INM}$ may not develop until this reservoir is exhausted. If the ER were to instantly quench T$_{INM}$ by compensatory membrane flow, NMMT would be impossible. But whether or how the ER affects T$_{INM}$ and NMMT has never been experimentally tested. To this end, we developed the Ca$^{2+}$-independent, intranuclear lipid-packing probe 'amphipathic lipid-packing domain inside the nucleus' (ALPIN), which allowed us to monitor T$_{INM}$ dynamics during nuclear swelling in the presence or absence of a contiguous ER.

## Results

### T$_{INM}$ rises with ER–nuclear membrane disruption during cell stress

The ER is normally contiguous with the nuclear membrane, but different types of stress can disrupt this connection through Ca$^{2+}$-dependent ER vesiculation[19–25]. Intriguingly, simultaneous imaging of eGFP–KDEL (ER lumen), eGFP–SEC61B (ER membrane) and cPla$_2$–mKate2–inner nuclear membrane (INM) adsorption during hypo-osmotic swelling showed that cPla$_2$–INM adsorption and ER disruption (ER area↓ and ER circularity↑) were highly correlated (Fig. 1a, Extended Data Fig. 1a and Supplementary Video 1). Confocal time-lapse analysis revealed that transient nuclear volume increases and envelope smoothening coincided with ER vesiculation and cPla$_2$ adsorption to the INM (Fig. 1b and Extended Data Fig. 1b–d). These acute osmotic responses ('shock

phase) reversed within ~10–15 min probably through regulatory volume decrease (RVD)[26]. It is known that the cell nucleus behaves like an osmometer that follows cytoplasmic volume changes upon osmotic shock and RVD, respectively[27–30].

The RVD-related nuclear volume decrease was accompanied by lamina shrivelling and cPla$_2$–INM desorption, in line with NE relaxation (Extended Data Fig. 1b–d). Thereafter, the nuclear volume remained steady for ~1–2 h despite the ongoing osmotic stress, while ER vesiculation proceeded. As ER vesiculation plateaued, nuclear swelling, NE smoothening and cPla$_2$–INM adsorption rebounded, indicating T$_{INM}$. This second nuclear swelling was perhaps caused by RVD failure related to cell damage and/or metabolic exhaustion. The progressive nuclear shrinkage that followed markedly differed from RVD-related nuclear shrinkage: it took hours not minutes and did not involve nuclear shrivelling or cPla$_2$ desorption, arguing against NE relaxation. Conceivably, nuclear volume loss, unlike during RVD, was accompanied by a nuclear surface decrease of unknown mechanistic origin. Whether the ER–outer nuclear membrane (ONM) vesiculation directly contributed to the putative surface loss remains an interesting question for future research.

After ~3 h of osmotic differential ($\Delta\Pi$) = 270 mOsm, cell lysis became visible by nuclear Sytox blue positivity (or earlier after $\Delta\Pi$ = 303 mOsm shock). Nuclear rupture sometimes preceded plasma membrane rupture. In such cases, cytoplasmic cPla$_2$ fluorescence became an inverse cell lysis marker, whose loss mirrored the rise of the Sytox blue signal (Extended Data Fig. 1b). At $\Delta\Pi$ = 220 mOsm, ER vesiculation rarely occurred, at $\Delta\Pi$ = 303 mOsm it did not reverse (Fig. 1b). Apparently, ER vesiculation is linked to critical (that is, outside the adaptive range), rather than mild, nuclear volume perturbations.

Super-resolution imaging (Fig. 1c, Extended Data Fig. 1e and Supplementary Videos 2 and 3) confirmed that ER vesiculation split the ER from the NE. Furthermore, fluorescence loss in photobleaching (FLIP) experiments revealed that diffusional eGFP–KDEL or eGFP–SEC61B marker exchange between the vesiculated ER and the NE (Fig. 1d,e) was completely abrogated, in line with previous findings[25]. By severely disrupting ER–NM contiguity, far more than any previously described ER shape factor mutation, this phenomenon offered itself to probe the role of the ER in T$_{INM}$/NMMT regulation.

### ALPIN monitors T$_{INM}$ dynamics independent of Ca$^{2+}$

The LPD-binding capacity of zebrafish cPla$_2$ renders it sensitive to T$_{INM}$, but only in the presence of Ca$^{2+}$ (refs. 5,11), which we wanted to

**Fig. 1 | cPla$_2$–INM adsorption correlates with ER–NM disruption during osmotic stress. a**, Left: representative confocal midplane slices of U2OS cells expressing the luminal ER marker eGFP–KDEL and zebrafish cPla$_2$–mKate2 at baseline and 5 min after $\Delta\Pi$ = 270 mOsm osmotic shock. Scale bar, 20 μm. Top middle: a plot of normalized ER surface area versus time after hypo-osmotic shock treatment ($\Delta\Pi$ = 270 mOsm). The data represent six independent experiments. The line shows the average and error bars are the s.e.m. Bottom middle: a plot of normalized ER circularity, and cPla$_2$–mKate2–INM adsorption versus time after hypo-osmotic shock ($\Delta\Pi$ = 270 mOsm). The line shows the average and error bars are the s.e.m. Note that the bottom middle image shows the result for a subset of the data presented in the top middle image. The data represent three independent experiments. Top right: a scatter plot showing the correlation between cPla$_2$–mKate2–INM adsorption and ER vesiculation (as measured by a decrease of ER surface area) in cells after being exposed to hypo-osmotic ($\Delta\Pi$ = 303 mOsm, $n$ = 33) or isosmotic medium ($\Delta\Pi$ = 0 mOsm, $n$ = 34) for 5 min. Bottom right: a scatter plot showing the correlation between cPla$_2$–mKate2–INM adsorption and ER fragmentation (as measured by an increase of ER network circularity) in cells after being exposed to hypo-osmotic ($\Delta\Pi$ = 303 mOsm, $n$ = 33) or isosmotic solution ($\Delta\Pi$ = 0 mOsm, $n$ = 34) for 5 min. The data represent one independent experiment performed on cells from two distinct biological sources (separate frozen vials). **b**, Left: representative, confocal midplane slices of U2OS cells expressing the ER membrane marker eGFP–SEC61B acquired at the indicated times after $\Delta\Pi$ = 270 mOsm or 303 mOsm

hypo-osmotic shock. Right: time-lapse quantification of ER circularity upon osmotic shock. The lines show the average and the shaded areas are the s.e.m. Data represent two independent experiments for 222 mOsm, three independent experiments for 270 mOsm and two independent experiments for 303 mOsm. Scale bar, 20 μm. **c**, Left: representative lattice SIM²apotome super-resolution maximum intensity projections (MIPs) of U2OS cells expressing the ER membrane marker eGFP–SEC61B ($\Delta\Pi$ = 0 mOsm, $n$ = 6). Right: a representative super-resolution MIP of U2OS cells 10 min after hypo-osmotic shock ($\Delta\Pi$ = 270 mOsm, n = 4). Scale bars, 5 μm. **d**, A cartoon scheme of the FLIP experiment. The ER–NE sites that are bleached (red, flash) or measured (green) are indicated. **e**, Fluorescence dynamics in response to bleaching the indicated sites. Left: FLIP of eGFP–KDEL at baseline ($\Delta\Pi$ = 0 mOsm, $n$ = 10) or during osmotic shock ($\Delta\Pi$ = 270 mOsm, $n$ = 8). Right: FLIP of membrane, eGFP–SEC61B at baseline ($\Delta\Pi$ = 0 mOsm, $n$ = 6) and osmotic shock ($\Delta\Pi$ = 270 mOsm, $n$ = 10). The line shows the average and error bars the 95% CI. The data represent three independent experiments. If not otherwise indicated, $n$ denotes the number of analysed cells or nuclei. Note that ER morphology measurements (normalized circularity/area) are always analysed on an entire FOV (that is, no structure segmentation, $n$ represents FOV, 1–8 cells per FOV). For time-lapse movies, values were normalized to baseline at $t$ = 0. For single images captured at the indicated time point, values were normalized to measurements obtained under isosmotic conditions. $A$, area; $S$, ratiometric signal; Cir, circularity; $I$, intensity; Ctr, non-targeted/bleached neighbouring cell control in whole field of view.

use to disrupt ER–NM contiguity[22]. The codependence of $Ca^{2+}$ for both sensing $T_{INM}$ and disrupting the ER renders the relationship between both parameters hard to disentangle by using the INM adsorption or enzymatic activity of cPla₂ as $T_{INM}$ proxy. To obtain a $Ca^{2+}$-independent $T_{INM}$ readout, we turned to the ALPS domain of ARFGAP1. Its $T_M$ sensitivity has been previously confirmed by micropipette aspiration

(cycling between 0.1 and 0.3 mN m⁻¹) and osmotic swelling experiments using giant unilamellar vesicles (GUVs)[11,13]. Each ALPS motif forms an amphipathic helix that inserts its hydrophobic face into LPDs created by membrane tension (Fig. 2a), positive curvature or conical lipids[17]. As predicted by its lack of known $Ca^{2+}$ binding sites, the tension dependent GUV interactions of ARFGAP1–ALPS were $Ca^{2+}$ insensitive

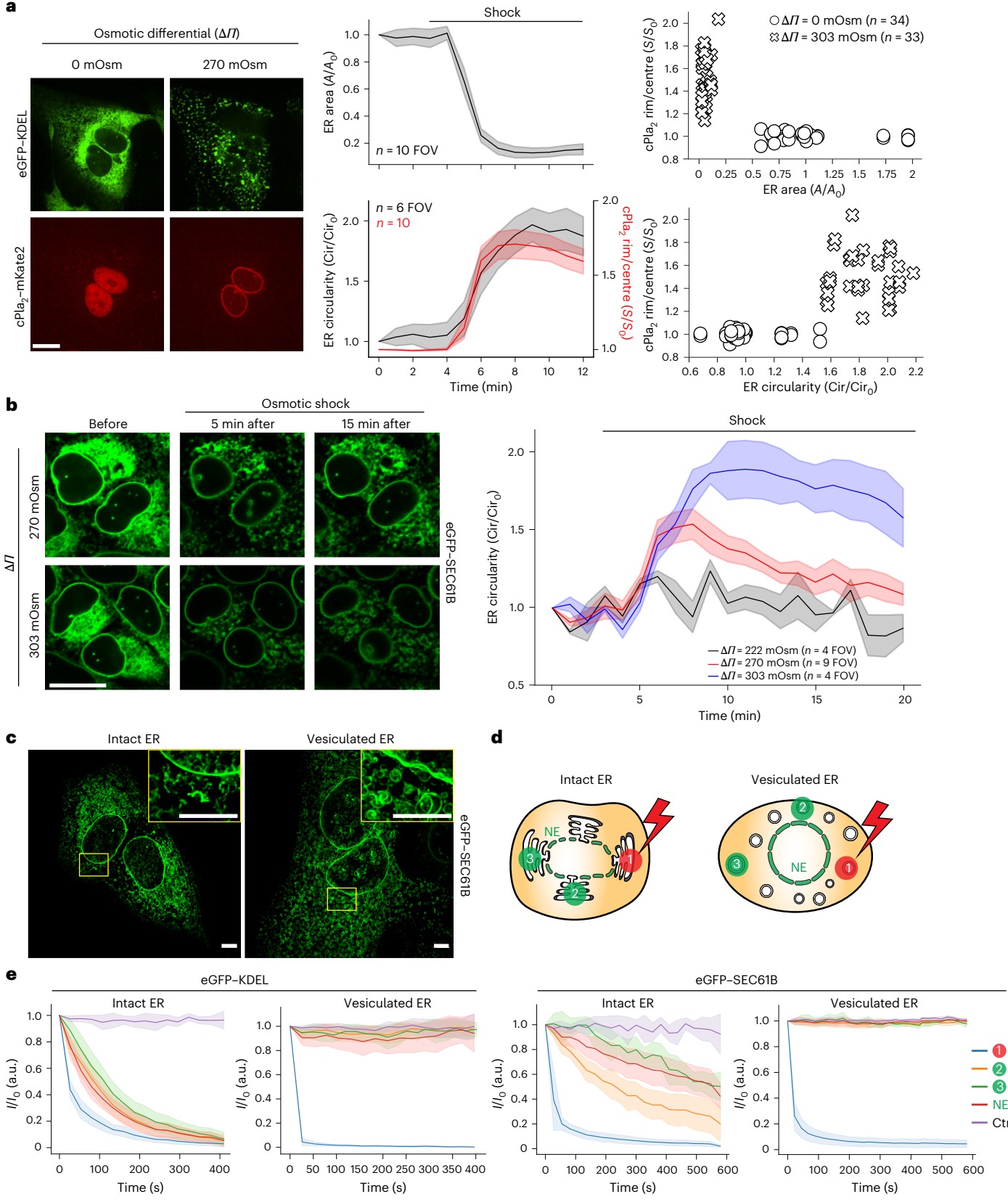

(Extended Data Fig. 2a). Just like the C2 domain of cPla$_2$, ALPS motifs target LPDs but because nuclear membranes are mostly flat, tension—not curvature—is the dominant generator of LPDs in this compartment. On stretched synthetic bilayers, ARFGAP1–ALPS competes with cPla$_2$ for the same LPD sites, confirming a shared T$_{INM}$-sensing mechanism. However, its apparent affinity for GUVs ($K_d' \approx 150$ nM) is roughly eight-fold lower than that of the cPla$_2$ C2 domain ($K_d' \approx 20$ nM at 20 μM Ca$^{2+}$, $\Delta\Pi = 240$ mOsm)[11]. Nevertheless, computational image processing of confocal midsections allowed us to robustly measure ALPS recruitment to the nuclear rim despite its lower sensitivity (Methods).

We fused either one (ALPIN1) or both (ALPIN2) ALPS motifs of ARFGAP1 to a triple nuclear-localization signal (3×NLS) to create nuclear-targeted sensors (Fig. 2b). Both constructs located to the nucleoplasm and—unlike cPla$_2$—also to nucleoli (Extended Data Fig. 2b). ALPS motifs attain their α-helical amphipathic conformation upon binding LPDs of stretched or curved membranes[31]. If no LPDs are available, ALPS domains are largely unstructured, which may favour their nucleolar sequestration[32,33].

Hypotonic shock caused partial nucleolar dissolution as previously reported[34,35] along with ALPIN1 adsorption to the INM (Fig. 2b, Extended Data Fig. 2b,c and Supplementary Video 4). Compared with ALPIN1, ALPIN2 showed a stronger baseline INM binding along with constitutive cytoplasmic vesicle association (Fig. 2b and Supplementary Video 5). These perinuclear ALPIN signals distracted our single-channel T$_{INM}$ detection method (nuclear edge segmentation along with rim:centre normalization based on the ALPIN channel; Methods), which allowed us to use the green and blue channel for simultaneous imaging of ER morphology and cell lysis (Sytox blue), respectively. When fluorescence channels are not limiting, ALPIN2 imaging with normalization by NE marker (eGFP–Lmnb1) fluorescence might yield more sensitive T$_{INM}$ detection, given its higher membrane affinity (Fig. 2b and Extended Data Fig. 2d).

Like cPla$_2$, ALPIN1 showed reversible membrane adsorption during the acute phase and persistent INM-binding during and beyond the prelytic phase of hypotonic shock. Again, ER vesiculation and INM adsorption were correlated (Fig. 2c and Extended Data Fig. 2e) as previously observed for cPla$_2$ (Fig. 1a). Apart from its nucleolar localization and lower membrane binding, ALPIN1 mimicked the cPla$_2$ response to osmotic shock, but without the Ca$^{2+}$ requirement.

By contrast, the cytosolic ALP-sensor amphipathic lipid-packing sensor domain inside the cytosol (ALPIC2; ARFGAP1–ALPS without NLS fusion) showed little ONM binding upon osmotic shock (Extended Data Fig. 2f). The discrepancy between the shock-induced INM or ONM adsorption of ALPIN or ALPIC (compare Fig. 2b and Extended Data Fig. 2f), respectively, could be due to a ONM–INM membrane tension gradient. Perhaps compensatory lipid flow between the INM and ER–ONM is throttled by the nuclear pores[36]. It has previously been shown that the nuclear shape-sensing function of cPla$_2$ depends on its nuclear localization[6]. Consistent with these results, our data highlight the INM as a 'privileged' mechanosensing surface.

## The ER serves as limited T$_{INM}$ buffer during nuclear swelling

To more closely examine how ER–NM continuity shapes T$_{INM}$, we tracked ALPIN1 in a permeabilized cell assay where Ca$^{2+}$ levels and colloid osmotic stress ($\Delta\Pi_{colloid}$) can be independently altered (Fig. 2d)[5,11]. Using low-dose digitonin, we removed cholesterol from the plasma membrane, perforating it while leaving the cholesterol-poor nuclear membrane intact. The permeabilization buffer contains 360-kDa polyvinylpyrrolidone (PVP) to balance extra- and intranuclear colloid osmotic pressure upon lysis; stepwise dilution of PVP then produces controlled nuclear swelling at adjustable Ca$^{2+}$ concentrations. In permeabilized cells, micromolar Ca$^{2+}$ disrupts the ER, unlike nuclear swelling (Fig. 2e and Extended Data Fig. 2g). Even with a contiguous ER, we observed ALPIN1–INM adsorption responding to colloid osmotic shock (Fig. 2f), suggesting that compensatory membrane flow from the ER does not prevent the INM from becoming tense and developing LPDs during nuclear swelling.

This raised the question of whether the ER buffers T$_{INM}$ at all. To address this, we measured ALPIN1 adsorption upon increasing $\Delta\Pi_{colloid}$ in the presence (0 μM Ca$^{2+}$) or absence (50 μM Ca$^{2+}$) of an intact ER (Fig. 2g). The ALPIN1–INM adsorption, despite being itself Ca$^{2+}$ independent (Extended Data Fig. 2a), was consistently higher when the ER membrane reservoir was severed from the nucleus by micromolar Ca$^{2+}$. These data argue that the ER incompletely buffers T$_{INM}$, even when ER–NM contacts are intact. Despite its overall structural contiguity, only parts of the ER seem to serve as nuclear membrane reservoir. Accordingly, we never witnessed an overall collapse of the ER into the nucleus.

**Fig. 2 | ALPIN reveals T$_{INM}$ buffering by the ER. a**, Top: a cartoon scheme of the 199–223 sequence of ARFGAP1 (part of the ALPS1 motif). Yellow, hydrophobic; purple, serine and threonine; grey, glycine and alanine; blue, basic residues. Bottom: a hypothetical cartoon scheme of ALPS (orange) adsorption to stretch-induced LPDs (red). ALPS adopts a helical, amphipathic conformation upon insertion into LPDs but remains largely unstructured in solution. **b**, Left: cropped representative, confocal midplane slices of ALPIN1 (ALPS1–mKate2 3XNLS) and ALPIN2 (ALPS1–2–mKate2 3XNLS)–INM interactions before and after hypo-osmotic shock ($\Delta\Pi = 270$ mOsm). Scale bars, 20 μm. Right top: a scheme of the ALPS-based T$_{INM}$ biosensor used in this study. Right bottom: time series analysis showing normalized sensor adsorption (left, ALPIN1, $n = 26$ cells from 5 independent experiments; right, ALPIN2, $n = 15$ cells from 2 independent experiments) to the INM after hypo-osmotic shock ($\Delta\Pi = 270$ mOsm). Note that ALPIN1 was co-expressed with eGFP–SEC61B, and portions of the ALPIN1 data presented here and the 270 mOsm condition in Fig. 1b were reanalysed from the same experiment. ALPIN2 was co-expressed with eGFP–Lmnb1. The line shows the average and error bars are the s.e.m. **c**, Left: representative confocal midplane slices showing ALPIN1–INM interactions and ER morphology (eGFP–SEC61B) at baseline (0 min) and 5 min after hypo-osmotic shock ($\Delta\Pi = 270$ mOsm). Scale bar, 20 μm. Right: a scatter plot showing the correlation between ALPIN1 NM adsorption and ER disruption in cells treated for 5 min with $\Delta\Pi = 0$ mOsm (open circles, $n = 52$), 270 mOsm (blue crosses, $n = 38$) or 303 mOsm (red squares, $n = 39$). The data represent one independent experiment performed on cells from two distinct biological sources (separate frozen vials). **d**, A cartoon scheme of the cell permeabilization experiment. **e**, Representative SIM midplane slices of U2OS cells permeabilized with the indicated lysis buffer supplements acquired at 60 nm *xy* resolution 60 min after the start of the experiment. Scale bar, 20 μm. Note that the representative slices were taken from the data analysed in Extended Data Fig. 2g representing one independent experiment performed on cells from two distinct source vials. **f**, Time series quantification of ALPIN1 adsorption to the INM after dilution of [PVP360] from 2.5% to 1.9% ($n = 23$ from 2 independent experiments), 1.25% ($n = 24$ from 4 independent experiments), 0.6% ($n = 26$ from 4 independent experiments) or 0% ($n = 25$ from 4 independent experiments) under Ca$^{2+}$-free conditions (Methods). The lines show the average and the shaded regions show the s.e.m. \*\*\*$P = 0.0005$, \*\*$P = 0.003$ and \*$P = 0.02$, determined by a one-way Welch's ANOVA (unequal variance) with post hoc Games–Howell test on ALPIN1 adsorption at $t = 15$ min. **g**, Left: representative confocal midplane slices of ALPIN1 NM adsorption at the indicated colloid stress and Ca$^{2+}$ conditions at baseline (0 min) and 2 min after the shift. Scale bar, 20 μm. Middle: time series quantification of ALPIN1 adsorption to the INM at the indicated conditions. The data represent five independent experiments for the middle graph and three independent experiments for the right graph. The line shows the average and error bars are the s.e.m. Insets show the adsorption rates determined by OLS fitting of the linear portion of the plots (between $t = 4$ min and 5 min). The shaded regions show the s.e.m. \*\*\*\*$P = 2.5 \times 10^{-5}$ (middle) and $P = 1.5 \times 10^{-5}$ (right), determined by a two-sided Student's *t*-test (equal variance; middle) and a two-sided Welch's *t*-test (unequal variance; right) on ALPIN1 adsorption at $t = 20$ min. If not otherwise indicated, $n$ represents the number of analysed cells or nuclei. For time lapse movies, the values in subsequent time points were normalized to values at $t = 0$. For single time point images, values from hypo-osmotic conditions were normalized to isosmotic controls.

## ER fragmentation and NMMT are triggered by critical stress in vitro and in vivo

Even when digitonin-permeabilized cells were kept at colloid osmotic equilibrium (2.5% PVP) so that their nuclei did not show overt swelling,

$Ca^{2+}$-dependent ER vesiculation promoted ALPIN1–INM adsorption (Extended Data Fig. 2h). This suggests that $T_{INM}$ rises not only when the nuclear water influx outpaces the ER-supplied membrane area, but also when ER–ONM vesiculation withdraws membrane from

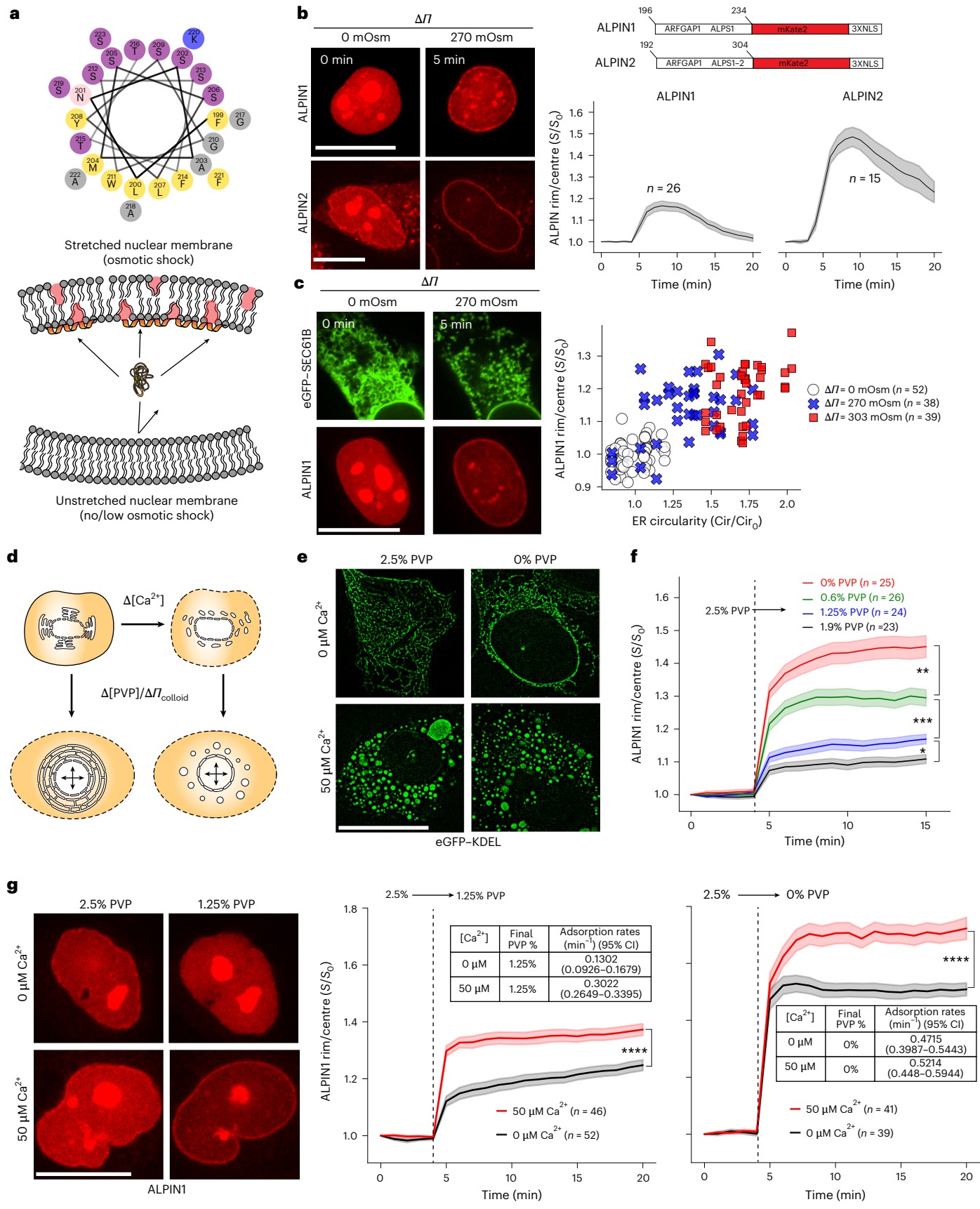

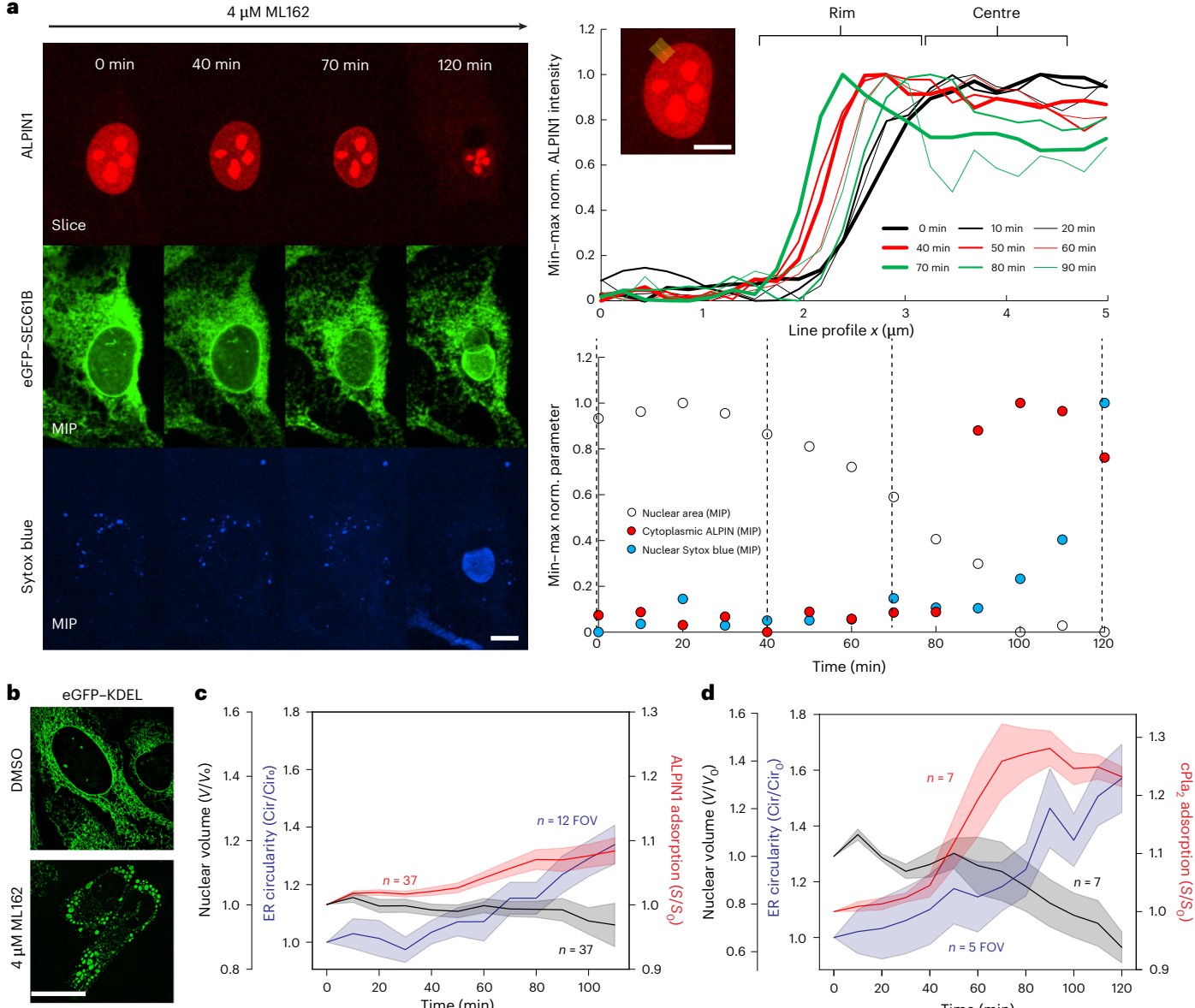

**Fig. 3 | Ferroptosis disrupts $T_{INM}$ buffering by the ER. a**, Left: a representative time lapse montage of ALPIN1–INM adsorption, ER morphology (eGFP–SEC61B) and cell lysis (Sytox blue) in a cell undergoing ferroptotic cell death upon exposure to the GPX4 inhibitor ML162. Scale bar, 10 μm. Top right: nuclear rim profiles (20 pixels wide) of ALPIN1 midplane fluorescence at the indicated location (inset, yellow line). Temporal order is indicated by line width and colour code. Inset scale bar, 10 μm. Bottom right: the temporal profile of the segmented nuclear area (open circles), cytoplasmic ALPIN1 (ROI on the extranuclear area, red circles) and nuclear Sytox blue signal (ROI on the intranuclear area, blue circles) as determined from confocal MIPs. Note that the representative time lapse montage was taken from the data analysed in Fig. 3c and represents two

independent experiments. **b**, Representative SIM midplane slices at 60 nm *xy* resolution of U2OS cells, expressing the luminal ER marker eGFP–KDEL, treated with the GPX4 inhibitor ML162 acquired 120 min after drug treatment. Scale bar, 20 μm. The data represent two independent experiments. **c,d**, Time lapse analysis of ER circularity (blue), nuclear volume (black) and ALPIN1–INM adsorption (red) (**c**) or cPla₂–INM adsorption (**d**) at the indicated times after ML162 treatment. The line shows the average and the shaded areas are the s.e.m. ALPIN1 data represent two independent experiments. Note that because ALPIN1 leaked from most cells after lysis, only movie segments acquired before leakage were analysed as leakage would otherwise result in broken tracks. *V*, volume.

the contiguous ONM–perinuclear ER, reducing the accessible area required to accommodate a given nuclear volume. Accordingly, ER vesiculation may directly elevate $T_{INM}$ and engage NMMT, even in the absence of overt nuclear swelling. Since ER vesiculation (equal to surface loss) has been observed under pathophysiological stress conditions that boost cytoplasmic $Ca^{2+}$, and since such conditions are tightly linked to cell damage and death[22,24], we wondered whether $T_{INM}$ and NMMT may be more broadly linked to critical cell stress and death, beyond the previously characterized mechanical perturbations ($\Delta\Pi$, compression).

Indeed, we had previously observed cPla₂–INM adsorption in the prelytic phase of ferroptotic cell death caused by the glutathione peroxidase 4 inhibitor ML162 (ref. 37); however, we did not consider possible contributions of the ER at this time. Therefore, we repeated the experiment, using eGFP–KDEL and eGFP–SEC61B as ER markers, and cPla₂ and ALPIN1 as $T_{INM}$ sensors.

Starting ~40 min after ML162 exposure, we observed continuous nuclear shrinkage and ER vesiculation, along with ALPIN1/cPla₂ adsorption to the INM (Fig. 3a–d and Extended Data Fig. 3a,b), reminiscent of the prelytic phase of osmotic stress (Extended Data Fig. 1c,d). Nuclear

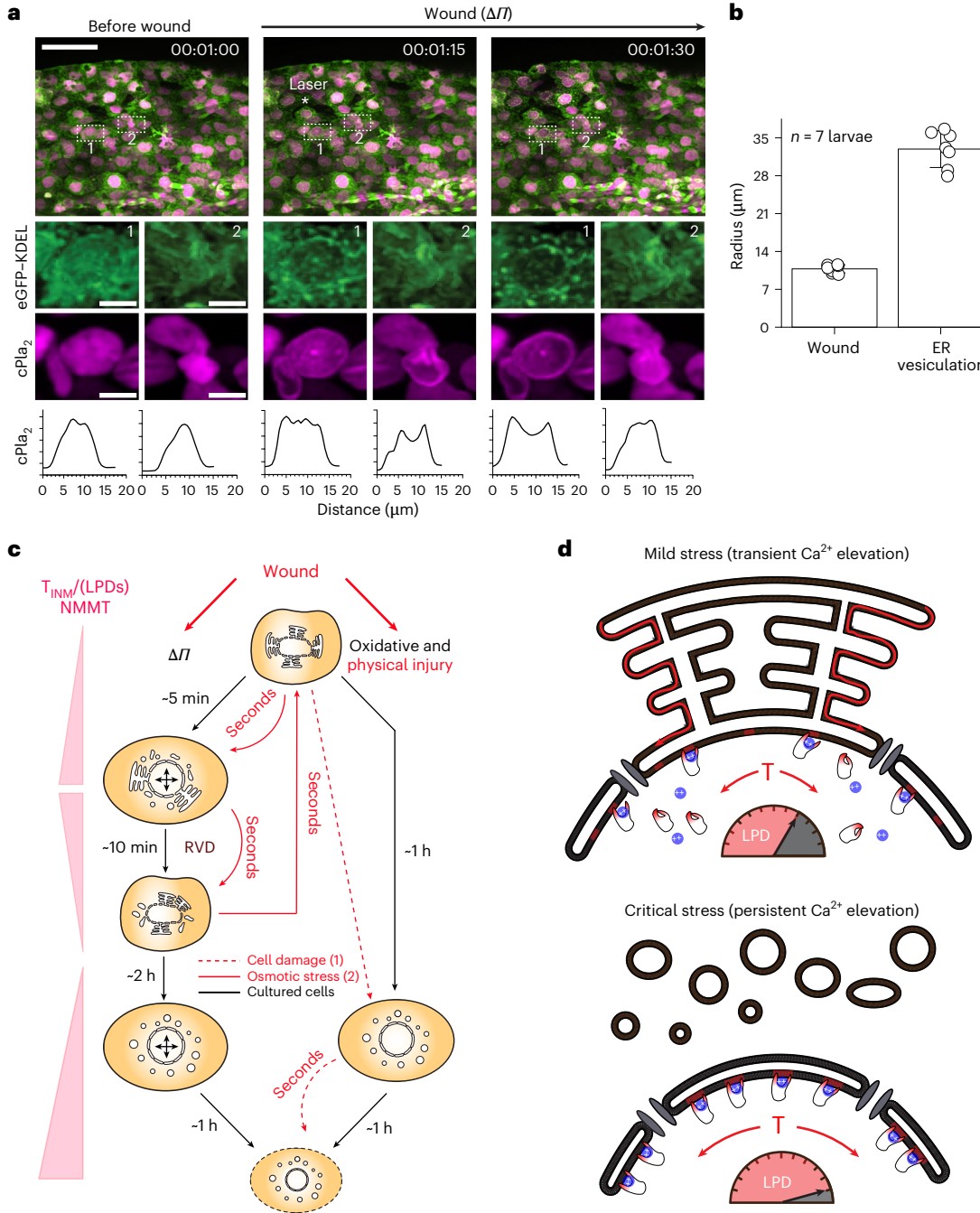

**Fig. 4 | ER–nuclear membrane network buffers $T_{INM}$ in response to tissue injury.** **a**, Top: confocal maximum intensity projection of cPla$_2$–mKate2 and luminal ER marker eGFP–KDEL in latrunculin A-pretreated zebrafish larvae after wounding under hypo-osmotic conditions ($\Pi$ ~ 10 mOsm) at $t$ = 00:01:15. Scale bar, 50 µm. Middle: magnification of eGFP–KDEL or cPla$_2$ in individual cells within (position 1) and just outside (position 2) the wound region. Inset scale bars, 10 µm. Bottom: profile plots of cPla$_2$–mKate2 emission of the depicted nuclei. Constitutive cPla$_2$ translocation with ER fragmentation at the wound margin was observed in all wounded fish ($n$ = 5). Reversible cPla$_2$ translocation without ER fragmentation (in the second or third cell layer) in four of five fish. Time stamp, hh:mm:ss. *indicates centre of the UV-laser blast. **b**, The average radius of the laser wound and ER vesiculation zone from $n$ = 7 larvae. These radii values were approximated from polygon area measurements from the dataset shown in **a**. **c**, An overview cartoon scheme outlining the approximate timing of stress responses to osmotic ($\Delta\Pi$), oxidative (ML162) or physical (laser wound) insults in vitro (black) or in vivo (red). Moderate osmotic shock ($\Delta\Pi$ = 270 mOsm) causes rapid nuclear swelling, partial ER fragmentation and $T_{INM}$, which is recovered by RVD. Under prolonged osmotic stress, cells enter the prelytic stage, with strong ER vesiculation and

$T_{INM}$, before they eventually lose plasma membrane integrity. Strong osmotic shock ($\Delta\Pi$ = 303 mOsm), oxidative stress, and physical injury cause irreparable cell damage (dashed line), which directly initiates the prelytic stage. Left-column engagement of NMMT during each stage. As a death-transcendent 'danger-signalling' mechanism, NMMT via cPla$_2$ is initiated before lysis and persists beyond. **d**, A hypothetical scheme of NMMT buffering by the ER. Middle: osmotic shock ($\Delta\Pi$) must unfold nuclear membrane invaginations before tension (T) can develop. Top: when the ER is contiguous with the nuclear membrane during homeostasis or mild stress, a compensatory lipid flow (dashed red line) partially relieves T. This amounts to less LPDs (shaded red regions/gauge) on the INM and less, or more reversible, hydrophobic cPla$_2$–INM interactions, as observed in cultured cells exposed to moderate osmotic stress, in distal wound cells (**a**, position 2), or in perivascular macrophages (see ref. 43) in vivo. Bottom: when the ER is vesiculated by high and persistent Ca$^{2+}$ after critical stress, there is no compensatory lipid flow to attenuate T and LPDs. This amounts to high (and more stable) cPla$_2$–INM interactions. The respective cell is turned into a constitutive, inflammatory lipid signalling hub. This is observed, for example, in wound margin cells (**a**, position 1), prelytic osmotically stressed, ferroptotic cells, and post-lytic (that is, permeabilized) cell corpses.

swelling after ML162 treatment, if at all noticeable (Extended Data Fig. 3a), was milder and more inconsistent than upon osmotic shock (Fig. 3a,c,d and Extended Data Figs. 1b and 2c). Akin to the prelytic phase of osmotic shock (Extended Data Fig. 1c), nuclear shrinkage after ML162 treatment was accompanied by NE smoothening (Extended Data Fig. 3b), arguing against relaxation. Towards the end of the experiment and just before plasma membrane failure, we frequently noticed pronounced NE blistering, consistent with previous reports[38]. These blisters contained eGFP–KDEL and appeared to expand at the expense of nuclear volume (Fig. 3a and Extended Data Fig. 3a). Ferroptosis, like other necrotic or hypo-osmotic insults, produces prelytic cell swelling[37,39], which dilutes cytosolic colloids and—provided NE area can be supplied—should drive nuclear volume expansion. However, depletion of the NE area reservoir (for example, via ER–ONM vesiculation or NE blistering) may reduce the envelope's accessible area and elevate $T_{INM}$ at near-constant (or even reduced) nuclear volume, thereby engaging NMMT, as observed.

The original evidence for the pathophysiological relevance of NMMT in vivo comes from the zebrafish wound model[40]. Zebrafish are freshwater animals, and when their epidermal barrier is damaged, cells and nuclei around the wound site swell by hypotonic shock[5,41]. The wound-induced osmotic shock elicits wave-like cPla$_2$ activation (travelling at ~50 μm s$^{-1}$ through the tissue) with irreversible cPla$_2$–INM adsorption observable exclusively at the wound margin. By contrast, reversible cPla$_2$–INM adsorption is typically seen farther away, in cells not directly compromised by the injury procedure itself, such as perivascular macrophages[5,42,43]. The hypotonic shock causes cPla$_2$-dependent arachidonic acid release[5,44,45]. Enzymatic or non-enzymatic lipid peroxidation converts arachidonic acid into chemotactic oxoeicosanoids that attract distant immune cells to the wound margin and regulate mucosal resilience[46–49]

To test whether the correlation between ER vesiculation and NMMT holds in live tissue, we turned to the well-established zebrafish tail fin wounding model. Using confocal microscopy, we simultaneously imaged cPla$_2$ translocation (cPla$_2$–mKate2) and ER morphology (eGFP–KDEL) in vivo. In larvae, transient cPla$_2$ adsorption to the INM occurs extremely fast and lasts only ~10 s. To better capture transient INM translocation at the reduced time resolution of two-channel acquisition (that is, 15 s per frame), we depolymerized F-actin with latrunculin, which prolongs the osmotic swelling response to injury[5], perhaps in part by modulating regulatory volume regulation[50].

Epithelial cells directly adjacent to the blast zone showed instant, irreversible ER fragmentation with permanent cPla$_2$ translocation, as in the prelytic phase of osmotic shock and ferroptosis (Figs. 4a,b, Extended Data Fig. 3c and Supplementary Video 6). In line with previous work[5,42,43], cells further away displayed transient cPla$_2$ recruitment without ER disruption. Expectedly, the reversible cPla$_2$–INM adsorption was harder to temporally discern without latrunculin treatment, but the correlation of ER vesiculation and cPla$_2$translocation directly at the wound margin was nevertheless evident (Extended Data Fig. 3c). This lasting cPla$_2$ activation may be responsible for residual neutrophil recruitment, independent of hypotonic shock[42]. Transient cPla$_2$ translocation to the INM, by contrast, requires osmotic shock. It mobilizes the majority of first-wave leukocytes and—via macrophages—promotes rapid serum exudation from nearby blood vessels, as reported previously[42,43].

Depending on the amplitude or duration, different kind of cell stresses (osmotic, oxidative or laser) cause (1) reversible ER fragmentation with reversible $T_{INM}$/NMMT or (2) persistent ER fragmentation with persistent $T_{INM}$/NMMT, with the latter being indicative of prelytic cell damage (Fig. 4c). Importantly, cPla$_2$ signals from stressed, severely damaged or dead cells alike, as long as Ca$^{2+}$ and $T_{INM}$ are maintained[5]. The removal of nuclear membrane slack by ER vesiculation thus enhances $T_{INM}$ and contingent inflammatory NMMT, irrespective of a cell's vital state (Figs. 2g and 4d).

## Discussion

By harnessing the Ca$^{2+}$-independent $T_{INM}$ sensor ALPIN, this study reveals that the ER serves as a stress/Ca$^{2+}$-gated nuclear membrane buffer, quenching NMMT under homeostatic conditions. This NMMT break is released upon critical stress. Persistent ER vesiculation in damaged cells locks the nucleus in a high-tension state that stabilizes cPla$_2$ on the INM—this occurs while the cell is technically still alive and it persists beyond cell death: placing swollen HeLa cell nuclei next to an isotonically balanced zebrafish wound mediates cPla$_2$-dependent neutrophil recruitment just like hypotonic shock at wound sites[5]. Thus, the cPla$_2$–NMMT pathway is a cell death-transcendent mechanism, not exclusively tied to a particular type of stress (for example, $\Delta\Pi$), nor to cell viability or cell death—it is a simple, physical process that just relies on Ca$^{2+}$ and concomitant nuclear pressurization. It can trigger restitutive mechanisms (repair, inflammation and so on) hours before the irreversible loss of plasma membrane integrity releases immunogenic molecules into the extracellular space that promote canonical damage-associated molecular pattern signalling.

$T_{INM}$ is never fully abrogated by the ER, suggesting that compensatory lipid flow from this organelle into the INM is throttled by obstacles that remain to be described. Anchored transmembrane proteins inhibit lipid flow, as previously proposed for Lem2 in yeast[51]. It is also conceivable that the nuclear pores act as topological bottlenecks for nuclear membrane/ER area redistribution[36]. The incompleteness of $T_{INM}$ buffering by an intact ER permits reversible cPla$_2$–NMMT in cells distal to the wound margin, such as perivascular macrophages[43].

To probe ER–NM interorganelle mechanics, we vesiculated the ER with Ca$^{2+}$, mainly because the perturbation of known ER shape genes[52,53] does not yield a comparable disruption. The molecular mechanisms underlying ER vesiculation deserve further investigation. Identifying the putative Ca$^{2+}$-regulated vesiculation factor(s), could inform therapeutic strategies that attenuate sterile inflammatory signalling by preserving the ER–NM contiguity in damaged cells.

## Online content

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

## Methods

### Cell culture and generation of stable cell lines

U2OS (HTB-96, ATCC) cells were grown in Dulbecco's modified Eagle medium (DMEM) supplemented with 10% fetal bovine serum, 2 mM L-glutamine and 1% penicillin–streptomycin. Lipofectamine 3000 (Thermo Fisher) was used to transfect U2OS cell lines according to the manufacturer's instructions. To establish stable transgenic cell lines, cells were co-transfected with SB100X *Sleeping Beauty* transposase and a pSB/MCS/Puro transposon vector encoding the genes of interest under the control of the EF1α or CMV promoter in 2:1 ratio. At 48 h after transfection, cells were grown in complete growth medium with 1 µg ml⁻¹ puromycin for 1–2 weeks. mKate2 and eGFP double-positive cells were sorted by flow cytometry. For imaging experiments, cells were incubated with L-15 LEIBOVITZ (Thermo Scientific, 21083027), which is buffered by phosphate salts, free-base amino acids, and 0.9 g l⁻¹ galactose (in place of glucose), with 0.55 g l⁻¹ sodium pyruvate and 2 mM L-glutamine.

### Lipids

The 1,2-dipalmitoyl-*sn*-glycero-3-phosphocholine (DPPC) and 1,2-dioleoyl-*sn*-glycero-3-phosphoethanolamine-*N*-(lissamine rhodamine B sulfonyl) (18:1 Liss Rhod PE) were purchased from Avanti Research.

### Expression vector construction

For bacterial expression, human ARFGAP1 ALPS1–2-eGFP (human ARFGAP1, amino acids AA192–304) were cloned into a modified pGEX 6P-1 backbone via EcoRI and NotI, leading to N-terminal GST fusion with Human Rhinovirus-3C (HRV-3C) protease cleavage sites.

For mammalian expression, rat ARFGAP1 ALPS1-mKate2 with 3XNLS (rat ARFGAP1, amino acids AA196-234) and human ARFGAP1 ALPS1–2-mKate2 with or without 3XNLS (human ARFGAP1, amino acids AA192–304) were fused to eGFP-Lmnb1 or eGFP-SEC61B through P2A peptides on the C-terminus and subsequently cloned into the pSB/CMV/MCS/Puro transposon plasmid. Within this plasmid, an expression cassette containing genes of interests is driven by a constitutive CMV promoter, along with another expression cassette that encodes a puromycin selection marker under the control of a synthetic SV40 promoter. Both expression cassettes are flanked by inverted terminal repeats that contain transposase recognition sites and allow transposition into host genome when co-expressed with the hyperactive *Sleeping Beauty* transposase (SB100X). For ALPS fusing with zebrafish eGFP-Lmnb1, ALPS sequences were inserted into the backbone harbouring eGFP-Lmnb1 via NheI and BshTI. For ALPS fusing with human eGFP–SEC61B, the plasmids harbouring both ALPS motifs and eGFP-LMNB1 were digested with BglII and XbaI, and SEC61B was ligated in to replace Lmnb1. mCherry-SEC61B and eGFP-Lmnb1 were also fused and subsequently cloned into the same transposon plasmid to investigate lamina shrivelling during hypo-osmotic stress and the prelytic phase of ferroptotic cell death. Briefly, mCherry-SEC61B was inserted into the vector backbone harbouring eGFP-Lmnb1 via NheI and EcoRI.

Zebrafish cPla₂-mKate2 was fused to eGFP-KDEL through a P2A peptide and subsequently cloned into a similar pSB/EF1α/MCS/Puro transposon plasmid, in which the genes of interests were under the control of a constitutive and mammalian-endogenous EF1α promoter.

Helical wheel representations of full-length ALPS1 and ALPS2 motifs are provided in Extended Data Fig. 4a. Detailed maps of pSB/CMV/MCS/Puro transposon plasmid are provided in Extended Data Fig. 4b. The full lists of plasmids and PCR primers used in this study are provided in Supplementary Tables 1 and 2.

### Preparation of GUV membranes

To generate GUV membranes composed of 99.5% DPPC and 0.5% 18:1 Liss Rhod PE (mol/mol), we used a standard electroformation protocol[11,54]. Briefly, 0.11 µM of lipid mixture dissolved in chloroform

was spread on 30–60 Ω indium tin oxide coverslips (Structure Probe Inc. supplies) and dried under a N₂ stream. To completely remove chloroform, the cover slips were exposed to vacuum overnight. Indium tin oxide coverslips were assembled into an electroformation chamber with 500 mM sucrose solution inside and incubated at 60 °C. The chambers were connected to a function generator (RIGOL DG1022). A sine wave with a frequency of 10.0 Hz and peak-to-peak voltage at 1.41 V was applied for 4–6 h followed by a square wave with a frequency of 4.5 Hz and peak-to-peak voltage at 2.12 V for 30 min. After electroformation, GUVs were extracted from the chamber using a gel-loading tip. The chamber was washed two times with 500 mM glucose solutions to collect residual GUVs. GUVs were used for experiments within 5 days after preparation.

### Recombinant protein expression and purification

The pGEX 6P-1 expression vector harbouring tagged ARFGAP1-ALPS1–2-eGFP was transformed into Rosetta 2 pLySs cells (Novagen). This vector is used to express GST fusion proteins with a PreScission protease site. Expression of ARFGAP1-ALPS1–2-eGFP was induced by 0.2 mM IPTG at 37 °C under vigorous shaking (250 rpm) for 3 h in LB medium supplemented with ampicillin (100 µg m⁻¹) at OD₆₀₀ = 0.8. After expression, bacteria were first centrifuged at 2,600 rcf for 30 min, pelleted and frozen at −80 °C overnight. Afterwards, bacteria were resuspended in 25 mM HEPES and 0.1% Triton X-100 with Complete protease inhibitors (Roche) at pH 7.4. Sonication was conducted on an ice/water mixture for 7–10 min with the following settings: 50% amplitude, 1 s on/2 s off pulses (Thermo Fisher Scientific, Sonic Dismembrator Model 500). After sonication and centrifugation at 24,000 rcf for 30 min at 4 °C (Sorvall RC 6 PLUS, Rotor SS-34), the supernatants were collected and incubated with glutathione sepharose 4B gel beads (Cytiva) overnight at 4 °C. The beads were then washed three times in a buffer containing 25 mM HEPES and 200 mM KCl (pH 7.4). Beads with GST-tagged ARFGAP1-ALPS1–2-eGFP were incubated with HRV-3C protease (PreScission protease, Cytiva) in a buffer containing 25 mM HEPES and 200 mM KCl (pH 7.4) overnight at 4 °C to elute ARFGAP1-ALPS1–2-eGFP. Eluates were concentrated with Amicon Ultra-2 10 K centrifugal filter devices (Millipore Sigma). The total protein concentration was determined using the Pierce BCA Protein Assay Kit (Thermo Fisher Scientific). Protein purity was determined by SDS–PAGE and Coomassie blue staining. Bands were quantified in Fiji. Aliquots were snap frozen in liquid nitrogen and stored at −80 °C.

### GUV equilibrium binding experiments

The binding experiments were performed according to a standard protocol[11]. In summary, concentrated GUVs were transferred to a hypo-osmotic binding buffer containing 0 or 20 µM Ca²⁺ (Supplementary Table 3) and ALPS1–2–eGFP domains at varying concentrations (ranging from 0 to 500 nM) in BSA-coated glass-bottom dishes (20 mg ml⁻¹ BSA solution was added 1–2 h before the experiment to reduce GUV membrane ruptures during prolonged incubation). Specifically, GUVs were loaded into hypotonic solution by gently dispensing them with the pipette tip directed onto the glass surface to ensure the vesicles settled at the bottom. GUVs were left in the binding solution for 2 h until protein recruitment approached an equilibrium. Then GUV membrane and protein fluorescence images were simultaneously acquired using a 40× objective and the protein adsorption fluorescence was quantified, normalized and adsorption isotherms were fitted as described in 'Image acquisition and computational analysis for cell culture and GUV experiments' section. All steps, including incubation, were carried out at room temperature to minimize the effects of thermal drifts during image acquisition. For binding quantification, the ALPS1–2–eGFP was titrated on GUVs and binding isotherms extracted (0 nM ALPS1–2–eGFP: $n = 47$ for 0 µM Ca²⁺ and $n = 41$ for 20 µM Ca²⁺; 25 nM ALPS1–2–eGFP: $n = 14$ for 0 µM Ca²⁺ and $n = 28$ for 20 µM Ca²⁺; 50 nM ALPS1–2–eGFP: $n = 28$ for 0 µM

$Ca^{2+}$ and $n = 39$ for 20 µM $Ca^{2+}$; 100 nM ALPS1–2–eGFP: $n = 57$ for 0 µM $Ca^{2+}$ and $n = 54$ for 20 µM $Ca^{2+}$; 200 nM ALPS1–2–eGFP: $n = 61$ for 0 µM $Ca^{2+}$ and $n = 73$ for 20 µM $Ca^{2+}$; 300 nM ALPS1–2–eGFP: $n = 87$ for 0 µM $Ca^{2+}$ and $n = 86$ for 20 µM $Ca^{2+}$; 400 nM ALPS1–2–eGFP: $n = 89$ for 0 µM $Ca^{2+}$ and $n = 73$ for 20 µM $Ca^{2+}$; 500 nM ALPS1–2–eGFP: $n = 74$ for 0 µM $Ca^{2+}$ and $n = 64$ for 20 µM $Ca^{2+}$).

#### Hypo-osmotic treatment experiments
Ninety-six-well glass-bottom dishes (Cellvis) were coated with 0.1% gelatin in PBS without $Ca^{2+}$ and $Mg^{2+}$ for at least 15 min at 37 °C before washing 3× with PBS. U2OS cells stably expressing fluorescent markers were then seeded in 96-well dishes (~$3 \times 10^4$ cells per well) 24 h before the start of the imaging experiments in the specified cell culture solutions (see above). Before the experiment, the cell culture medium was exchanged to Leibovitz's L-15 medium (Thermo Fisher). Under isosmotic conditions, U2OS cells were incubated in L-15 stock culture medium to obtain a final osmotic pressure of 341 mOsm. To induce different levels of hypo-osmotic shock, the cell imaging medium was diluted by the addition of double-distilled water supplemented with 1.26 mM $CaCl_2$ (Supplementary Table 4). The wells were buffered by a combination of phosphates, free base amino acids (2 mM L-glutamine) and galactose, and did not rely on sodium bicarbonate and $CO_2$ pH buffering. For standard time-lapse imaging, cells were first recorded under isosmotic conditions for 4–5 min before hypo-osmotic shock was applied by dilution. They were then imaged continuously for a maximum of 20 min ($t = 1$ min). In extended time-lapse experiments, cells were imaged for three frames (6 min total, $t = 2$ min) under isosmotic conditions before applying the hypo-osmotic treatment and imaged for 6 h. The humidity and temperature of the imaging chamber was supported by a TOKAI-HIT substage heater, calibrated and set to 37 °C at least 30 min before the start of the experiment. During imaging, cells that were in the central region of the well were chosen for image acquisition. To evaluate plasma membrane integrity during osmotic swelling, 50 nM Sytox Blue (Molecular Probes) was included in the culture medium and the dilutant. For single time point experiments, U2OS cells expressing ALPIN1/cPla$_2$–mKate2 and ER markers were exposed to hypo-osmotic solutions for 5 min before being imaged for nuclear membrane binding and ER morphology.

#### Nucleolus detection and hypo-osmotic treatment
U2OS cells were seeded in 96-well glass-bottom plate ($1 \times 10^5$ cells per well) the night before the experiment, at least 16 h before the start of the imaging experiment. To visualize the nucleolus, the adherent U2OS cells expressing either of ALPIN1 (ALPS1–mKate2 3XNLS) or cPla$_2$–mKate2 were stained live with NUCLEOLAR-ID (Enzo, ENZ-51009-500) by incubating the cells with green detection reagent dissolved in L-15 media for 30 min at 37 °C. Subsequently, the cells were transferred to a confocal microscope and left untreated or shocked with hypo-osmotic dilution for 15–20 min ($\Delta\Pi = 270$ mOsm) and imaged.

#### Cell permeabilization experiments
The permeabilization protocol was adapted from an established method[5]. A base medium (BM) was prepared under $Ca^{2+}$-free conditions for all permeabilization experiments, consisting of 123 mM KCl, 12 mM NaCl, 1 mM $KH_2PO_4$, 1.94 mM $MgCl_2$, 10 mM MOPS and 10 mM EGTA/Na$^+$, adjusted to pH 7.3–7.4. To perform permeabilization in the presence of $Ca^{2+}$, a modified version of BM was used in which all components remained unchanged except for the EGTA/Na$^+$ concentration, which was reduced from 10 mM to 1 mM. $CaCl_2$ was added to the modified BM to raise the free $Ca^{2+}$ concentration to 50 µM according to the CHELATOR algorithm[55]. Unless otherwise specified, cells were permeabilized in either BM or modified BM supplemented with 12.5 µg ml$^{-1}$ Digitonin (Cayman Chemical) and 2.5% PVP360 (Millipore Sigma) for over 60 min before imaging, ensuring that the NE was fully relaxed at

the start of acquisition. To induce and control $T_{INM}$ (tension-induced nuclear membrane deformation), the concentration of PVP360 was systematically reduced.

To assess the behaviour of the ALPS1–mKate2-3xNLS (ALPIN1) sensor and its binding dynamics to the NE during increasing colloid osmotic pressure (Fig. 2f), permeabilized U2OS cells expressing ALPIN1 and eGFP–Lmnb1 were first incubated and imaged in BM supplemented with 2.5% PVP360 for 5 min. Cells were then washed twice with PBS and subsequently incubated and imaged in BM containing decreasing concentrations of PVP360 (ranging from 0% to 2.5%) for an additional 10 min.

To systematically test the effects of $Ca^{2+}$ concentration and colloid osmotic stress on ER structures (Extended Data Fig. 2g), U2OS cells expressing ER markers (eGFP–KDEL) were imaged after incubating for 1 h under four experimental conditions: (1) BM with 2.5% PVP360 ($Ca^{2+}$ free, no colloid osmotic stress), (2) BM alone ($Ca^{2+}$ free, while inducing colloid osmotic stress), (3) modified BM with 2.5% PVP360 and 50 µM $CaCl_2$ (presence of $Ca^{2+}$, but no colloid osmotic stress) and (4) modified BM with 50 µM $CaCl_2$ alone (presence of $Ca^{2+}$ and colloid osmotic stress).

To evaluate $T_{INM}$ responses during increasing colloid osmotic pressures in cells with either intact or disrupted ER networks (Fig. 2g), permeabilized cells expressing ALPIN1 and eGFP–Lmnb1 were imaged for 5 min in BM containing 2.5% PVP360 (intact ER) or in modified BM with 2.5% PVP360 and 50 µM free $Ca^{2+}$ (fragmented ER network). Subsequently, the PVP360 concentration was diluted to either 1.25% or 0%, while maintaining constant $Ca^{2+}$ levels. Imaging was continued for an additional 15 min. For diluting the PVP360 concentration from 2.5% to 1.25% or 0%, permeabilized cells were washed two to three times before incubation. Without washes, residual high molecular weight and viscous PVP360 would differentially delay colloid osmotic shock, confounding the interpretation. Imaging was paused during the wash steps and resumed after permeabilized cells were incubated in the diluting medium. Note that cells remained only loosely adherent after permeabilization; therefore, all wash steps were performed gently. Medium was added and removed by dispensing against the well wall rather than directly onto the cells to minimize detachment.

To evaluate whether ER vesiculation alone can induce a $T_{INM}$ response under osmotically balanced conditions (Extended Data Fig. 2h), single time point images of U2OS cells expressing ALPIN1 and eGFP–Lmnb1 were taken after permeabilizing in BM containing 2.5% PVP360 (intact ER) or in modified BM with 2.5% PVP360 and 50 µM free $Ca^{2+}$ (fragmented ER network) for 30 min.

**FLIP.** For testing the contribution of the ER network to NMMT, membrane (eGFP–SEC61B) or lumen (eGFP–KDEL) localization was used as a proxy for ER dynamics in the U2OS cells. The cells were seeded in DMEM media at a density of $5 \times 10^5$ cell ml$^{-1}$ in 96-well plates that were precoated with gelatin. The media were exchanged to L-15 and cells were imaged with the FLIP approach. FLIP imaging was performed on a Leica TCS SP8 Confocal Laser Scanning system equipped with a Leica white light laser (1.5 mW: 470–670 nm, 78 Mhz, output power of 70.00%), argon lasers (65 mW: 458, 476, 488, 496 and 514 nm continuous) that were used for excitation and photobleaching and a Plan Apochromatic objective (Leica HC PL APO CS2, 63×/NA 1.40 oil). The emission was detected on internal spectral HyD SMD2 sensors for mKate2 (618–660 nm, gain 100/offset −0.22) and eGFP 2× liquid-cooled PMT (498–540 nm eGFP gain 835.8/offset −0.22), respectively. The light path was mounted with a ×2 lens changer (CS2 UV Optics 1) and emission notch filter set and polarization FW mirror (NF 488/561/633). The fluorescent images were acquired at 1,024 × 1,024 pixel resolution in *xyt* scan mode at 600 Hz scan speed (pixel dwell time of 0.4 µs) at the midplane of the cells of interest in the field of view (FOV) or 184.52 × 184.52 µm at pixel size (voxel 0.18 µm) in *x* and *y* direction. The imaging set up was optimized for minimal fluctuations in baseline fluorescence for

the duration of imaging, therefore power for lasers were adjusted to 1.5% for 488 nm excitation (eGFP–ER) and 2.5% for 561 nm (mKate2), where the $xy$ scan was bidirectional and pinhole was adjusted to 191 µm (PinholeAiry of 2 AU) with line averaging of 3–4, and data were collected at 12-bit resolution. The photobleaching was achieved by ablating a 2.5-µm radius circle in the FOV with an argon bleach laser line (488 nm, output power of 19.84%) and a simple beam expander for FRAP booster mode. Each FRAP was set up in the Leica Microsystems software (Leica LAS X 3.5.7.23225) using the LAS X FRAP AB module where images were acquired once as a single frame for pre-bleach followed by a series of continuous photobleaching iterations in the same region for three cycles (loops), followed by full FOV acquisition. This sequence was set to repeat until the complete depletion of target cells with the fluorophore of interest (10–25 repeats). The pre-bleach took 5–7 s, while acquisition of FRAP sequence, including three loops per subsequent frame, took approximately 20 s for all imaging experiments. The settings that were determined for the ER network at baseline were applied to the hypo-osmotically shocked cells, with $CaCl_2$ dilutant as described above as the depletion control for critically stressed cells in luminally (KDEL–eGFP) or membrane (eGFP–SEC61B) expressed eGFP variants. For ER vesiculation, the cells were hypotonically shocked ($\Delta \Pi = 270$ mOsm) and imaged 5–10 min after treatment. The obtained .lif files were then loaded into Fiji (ImageJ) and stitched, and an invariant circle region of interest (ROI) (width, 5 µm; length, 5 µm) was overlaid on the eGFP (ER) channel and analysed for ROI 1, ROI 2 and ROI 3, where ROI 1 corresponds to photobleached area, the median axis in plane with regard to the target position and the opposite location to the photobleached area with respect to the nucleus. The nucleus was segmented using binary (otsu) thresholding and the segmentation mask was overlaid to the eGFP (ER) channel for quantification of nuclear ER intensity over time. The baseline fluorescence intensity fluctuation in non-targeted cells is recorded as Ctr ROI, referring to non-targeted neighbouring cell control region within the field of view. The data were extracted and analysed in a custom Python script (v3.9).

**Super-resolution fluorescence microscopy.** For delineating the states of ER network or its vesiculation, we distinguished these two morphologies with super-resolution fluorescence microscopy. Structured-illumination microscopy (SIM) systems, which achieve a nominal resolution of 60 nm in $xy$ were used in this study. One such example is the Zeiss Elyra 7 microscope equipped with a Plan-Apochromat 63×/1.4 oil DIC M27 objective and 2× pco.edge sCMOS (version 4.2 CL HS, Innovatek LCS-BU) cameras. The 60-nm super-resolution cell images were acquired at baseline ($\Delta \Pi = 0$ mOsm) and 10 min after hypo-osmotic shock ($\Delta \Pi = 270$ mOsm) and imaged using the Zeiss Elyra 7 microscope with the following build. The system contained 488 nm and 561 nm OPSL lasers. The light path was equipped with a reflector turret mounted with FW1:LBF (405/488/561/642 laser-blocking filter) beam splitters and filters for red (TV1:LP570) or green (TV2:BP 495–550) emission. The data were recorded in apotome $SIM^2$ in non-delay (five phases for each of the $z$ sections) with 3D leap mode and a FOV of 80.14 µm × 80.14 µm (or 1,280 ×1,280 pixels), 0.063 µm pixel size, 0.329 µm $z$ step, stack 5–12 µm and 50–80 ms exposure time for both colours (camera sensor temperature, air 7 °C) and collected at 16-bit resolution. The acquisition of individual $z$ stacks takes about 12–20 s. The mKate2 and eGFP fluorophores are excited with the 561 nm (1.6% power) and 488 nm (4.1% power) laser lines, respectively. We also generated single time point images in Lattice $SIM^2$ (13 phases) with 3D leap mode with a FOV of 80.14 µm × 80.14 µm (or 1,280 ×1,280 pixels), 0.063 µm pixel size, 0.110–0.150 µm $z$ step and 80 ms exposure time for both colours (camera temperature set to 7 °C). Overall, for each focal plane, 5 or 13 phase images were acquired. Exposure time and laser power were balanced for each fluorescence channel individually to minimize bleaching and exposure time. SIM and Lattice SIM reconstruction was performed with the SIM processing

tool of ZEN 3.0 SR FP2 (64-bit, 16.0.20.306 black) software for applying $SIM^2$ image reconstruction on the ER channel as follows: the advanced filter was set with best fit and Gauss (sectioning = 100), we did not use any detrending for either channel or scale to raw image, the set regularization weight was 0.015, with proc.Sampling and Out.Sampling set to (×2 or ×4) and the algorithm was run for 20 iterations. After image reconstruction, the voxel size decreased to 0.031 × 0.031 × 0.329 µm (Out.Sampling ×2) or 0.016 × 0.016 × 0.110 µm (Out.Sampling ×4). The images (.czi format) were then converted to .ims files and 3D rendered in Imaris with the filament and surface GUI function, where surface grain size (0.093 µm) and the diameter of the largest sphere (0.93 µm) were segmented automatically. The segmentation masks were exported from imaris (Imaris v10.2, Bitplane) as .IV. The 3D segmentation mask was then triangulated and unpacked as a mesh of vertices and faces with a custom Python pipeline. The interactive data and the method of extraction from $SIM^2$ are available at https://github.com/zazadovv/ER_Nucleus (representative interactive ER rendering data files are available in Demo_file_3D folder and extracted.html and can be viewed from any internet web browser). The mesh array was decimated to maintain only 70% of vertices (30% discarded) and transformed into an interactive 3D STL in a plotly.js (v3.0.1) model for the qualitative evaluation of nuclear ER connections or the absence of a fragmented state.

**Ferroptosis induction.** To study ER fragmentation during ferroptosis, cells were imaged after incubating in L-15 medium supplemented with DMSO or 4 µM glutathione peroxidase 4 inhibitor ML162 (Selleckchem) for 2 h. For the temporal correlation of nuclear membrane binding, ER fragmentation and cell death, the time-lapse movies of ML162 were acquired on U2OS cells expressing eGFP–KDEL, cPla$_2$–mKate2 or eGFP–SEC61B, ALPIN1 and Sytox blue. For ML162 the cells were seeded in a gelatin precoated (0.1% gelatin in PBS without $Ca^{2+}$ and $Mg^{2+}$) 96-well imaging plate (Cellvis, P96-1.5H-N) at a density of $1 \times 10^4$ cells ml$^{-1}$ and allowed to adhere overnight. Before imaging, the DMEM culture medium was exchanged to L-15 supplemented with 50 nM Sytox blue and transferred to a 37 °C pre-heated microscope chamber. During imaging, after the first time point, the wells were either incubated with 4 µM ML162 or DMSO and 50 nM Sytox blue (Thermo Scientific, S11348) for 2 h, where each $z$ stack ($z = 17$ steps) was collected every 10 min ($t = 10$ min, 13 frames) on a spinning-disc confocal microscope.

**Zebrafish husbandry.** Adult transgenic Casper Zebrafish (*Danio rerio*) strains were maintained by adhering to the guidelines of the care and use of laboratory animals of the National Institutes of Health and Office of Laboratory Animal Welfare recommendations. The animal procedures were in compliance with an animal protocol approved by the Institutional Animal Care and Use Committee of the Memorial Sloan Kettering Cancer Center (protocol no. 11-01-002). The adult Tg(*hsp70i*:cPla$_2$-mKate2-P2A-eGFP-KDEL) fish were reared in 2.8 l polycarbonate tanks at animal density 10 fish per litre. The anaesthesia for the offspring was all conducted with 0.2 mg ml$^{-1}$ 3-amino benzoic acid ethyl ester (Sigma, MS-222, E10521), (pH 7.0), buffered in 0.5 mg ml$^{-1}$ anhydrous sodium phosphate dibasic (Fisher, BP332-500). The animals were staged by days post-fertilisation (d.p.f.). Sex was indeterminate at 2.5–4 d.p.f. and all in vivo experiments were conducted at these larval stages. The embryos were collected from natural spawning and raised in standard hypo-osmotic E3 containing 0.1% (w/v) methylene blue (Sigma-Aldrich, M9140) for first 24 h followed by E3 medium (5 mM NaCl (Sigma-Aldrich, S7653), 0.17 mM KCl (Sigma-Aldrich, P9333), 0.33 mM $CaCl_2$ (Sigma-Aldrich, C5670) and 0.33 mM $MgSO_4$ (Sigma-Aldrich, M7506)) in 100 mm Petri dishes (Fisher Scientific, FB0875713) in complete darkness.

**Transgenesis, plasmid construction and in vitro transcription.** Fertilized Casper zebrafish embryos were collected and injected at the one-cell stage into the cytoplasm with a Nanoject II microinjector

(Drummond Scientific). Plasmids were assembled with the Gateway multisite cloning kit using LR Clonase (Invitrogen, C12537-023) and multicloning sites between the gateway att sites flanking the cassette of p5E (hsp70i), pME (cPla2-mKate2) and p3E (P2A-eGFP-KDEL) plasmids that were recombined into Tol2kit Destination pDESTc-rybb1 for generating Tg(hsp70i:cPla2-mKate2-P2A-eGFP-KDEL). The middle entry pME vector was assembled from the cPla2 (Ensembl: ENSDARG00000024546) open reading frame amplified by PCR and ligated into pDONOR221 containing the pME gateway compatible att sites and subsequently fused in frame to mKate2 (Evrogen) as a C-terminus fluorescent tag with a 15 amino acid GS-enriched linker. The p3E entry vector bears a self-cleaving peptide in the $N_{TD}$ of eGFP that is followed by the ER localization signal KDEL and SV40 pA. For transgenesis, Tol2kit transposase mRNA was transcribed from the NotI linearized pCS2FA-transposase plasmid with the mMESSAGE mMACHINE SP6 reverse transcription kit (Thermo Scientific, AM1340). To generate the in vivo construct, the pDest-Tol2pA2 was used as the backbone bearing a (attR4/attR3) cassette for multisite LR recombination. The entry clones were flanked with the multisite-compatible attL4-attB cassette in p5E-HSP70i (attL4/attR1), pDonor-cPla2_MK2_Nonstop (attL1/attL2) and p3E-p2A-KDEL-eGFP (attB2/attB3) to yield the destination construct (Extended Data Fig. 5a).

The final transgenesis construct was injected into one-cell stage zebrafish embryos (~2.7 nl per embryo) at a concentration of 25 ng μl⁻¹ along with 25 ng μl⁻¹ Tol2kit transposase mRNA. Among the injected animals, fluorescence-positive siblings in the eye were selected and raised in husbandry and backcrossed at sexual maturity to Casper fish and their progeny were used for larvae experiments. A detailed map of the Tg(hsp70i:cPla2-mKate2-P2A-eGFP-KDEL) plasmid and restriction digests results are provided in Extended Data Fig. 5b.

### Intravital confocal microscopy and laser wounding of zebrafish larvae

For laser-wounding experiments, at 2 d.p.f., Tg(hsp70i:cPlate2-P2A-eGFP-KDEL) were heat shocked at 37 °C for 2 h to induce transgene expression. The fluorescent larvae were sorted for experiments at 12–18 h after heat treatment using a coaxial dissection stereomicroscope (MVX10, Olympus) equipped with a mercury lamp (LM200B1-A, Prior Scientific) and set dichroic mirrors (MVX-RFA; 540/35 U-MGFPHQ/XL, 625/55 U-MRFPHQ/XL). The selected larvae were anaesthetized in hypo-osmotic E3 medium[56] containing 0.2 mg ml⁻¹ 3-amino benzoic acid ethyl ester (Sigma, MS-222, E10521) (pH 7.0), buffered in 0.5 mg ml⁻¹ anhydrous sodium phosphate dibasic (Fisher, BP332-500). Using ~200 μl 1% low-melting (LM) agarose dissolved in E3, the anaesthetized 3 d.p.f. embryos were embedded on their right side in a 60 mm plastic Petri dish (Corning, 351007). The solidified LM agarose was covered with ~2–3 ml of standard E3 medium, or pre-incubated for 30 min in E3 supplemented with ethanol as vehicle (1% EtOH in E3) or latrunculin A (2.5 μM, Sigma-Aldrich, L5163) to create a submerging environment for the 25× objective lens (NA 1.1 ∞/0–0.17 WD 2 μm Water Dipping, Nikon) on an upright microscope. The samples were excited with 488 and 561 nm diode lasers (Andor Revolution XD). Excitation intensity/exposure was set to 35%/80 ms (488 nm) and 30%/80 ms (561 nm). ER and cPla2–mKate2 fluorescence were acquired using a Andor iXon3 897 EMCCD camera mounted on a Nikon Eclipse FN1 microscope (25× objective lens; NA 1.1, ∞/0–0.17 WD 2 μm, Nikon) equipped with a Yokogawa CSU-X1 spinning-disc unit (pinhole of 50 μm) and NIS imaging software (NIS Elements, 3.22.14). Emission was acquired through band-pass filters for green (525/40, Semrock, FF02-525/40-25) or red (617/73, Semrock, FF02-617/73-25) fluorescence emission spectra of confocal stacks, collected at 1.5 μm z step size and repeated in no-delay intervals per position for up to 5 min. The wounds were induced at t - 1 min with several successive laser pulses targeted at the periphery of the epithelium

using a microscope-mounted 435 nm micropoint laser (Andor). The experimental scheme for heat shock-induced expression is shown in Extended Data Fig. 5c.

### Two-photon laser wounding of zebrafish larvae

For multiphoton imaging experiments, 2 d.p.f. Tg(hsp70i:cPla2-mKate2-P2A-eGFP-KDEL) were heat shocked at 37 °C for 2 h approximately 24 h before the start of the imaging experiment. The heat-shocked larvae were anaesthetized in hypo-osmotic E3 medium containing tricaine analgesic. The larvae were prepared using the same protocol as for UV-laser wounding experiments except for latrunculin A pre-treatment. The anaesthetized live samples were embedded in 35-mm glass-bottom dishes (MatTek Corporation, P35G-1.5-14-C) before being transferred to a pre-heated Leica Stellaris 8 Dive two-photon microscope. The microscope is equipped with two tunable infrared lasers: Mai Tai HP (690–1,040 nm, continuous) and Insight X3 (680–1,300 nm, continuous), both capable of delivering up to 2.4 W average maximum power. Tissues were wounded using the FRAP ab1 XYZT module in galvanometric scanning mode with bidirectional scan direction, a scan speed of 1,000 Hz, and a pixel dwell time of 0.3508 μs. A HC FLUOTAR L VISIR 25×/0.95 WATER objective was used with 2.25× zoom and acquired at 15-s intervals. Images were collected at Nyquist resolution with a pixel size of 0.25 μm, yielding a FOV of 206.7 × 206.7 μm (840 × 840 pixels), and a z-stack depth of 10 μm with a step size of 2 μm per slice. For dual-colour excitation, the Mai Tai HP laser was tuned to 924 nm (MP1, 5.5% power) for eGFP excitation, and the Insight X3 laser was tuned to 1,118 nm (MP2, 2.6% power) for mKate2. Wounding was performed using the FRAP module with a sequence of four pre-bleach iterations, followed by six bleach iterations and 20 post-bleach iterations. For laser ablation, bleaching intensity was set to 100% on MP2 (1,118 nm) and 20% on MP1 (924 nm), with a total dwell time of 2 s at the mid plane of the tissue, targeting a circular ROI (radius 25 μm) for localized wounding. Emitted fluorescence was filtered using the Galvo Resonant Pan module (multifunction port RSP 1005) with SP815 and SP800 short-pass filters, in combination with a Notch FW 2 (OPO SP800) filter and a variable beam expander (expansion factor of 1.03, z-colour correction offset of 0). Detection was performed using external spectral non-descanned hybrid detectors (HyD NDD1 and NDD2) operating in photon-counting mode, with NDD1 configured for eGFP detection (480–527 nm) and NDD2 for mKate2 (620–732 nm). All imaging data were saved and processed using Leica LAS X software (Leica LAS X, 4.8.1.29271).

### Image processing for zebrafish experiments

The 4D image stacks of Tg(hsp70i: cPla2-mKate2-P2A-eGFP-KDEL) were corrected by 3D deconvolution, where sample PSF was derived and computed automatically with depth calibration and without image intensity subtraction or pre-processing in Nikon NIS elements (5.21.03, Build 1489). The deconvolution process was carried by the default method using the landweber algorithm for spinning-disc modality with a 50 μm pinhole size with an immersion refractive index 1.33 (water) for both 488 and 561 nm fluorescent light emission. The deconvolved or MP image stacks were denoised using the denoise.ai tool for improving the signal-to-noise in Nikon NIS elements (5.21.03). The obtained files were loaded in Fiji (ImageJ, 1.53p) and the last frame was used for extraction of wound core and ER vesiculation area. Specifically, the polygon tool was used to trace parameter and set-scaled images for the dark area after 435 nm–UV harmonic laser wounding in tissue, and was designated as wound core in all biologically independent replicates. The ER vesiculation was traced by the polygon tool outside of the wound core until no visual vesiculation patterns were observed deep within the tissue. The obtained values were recorded per pixel of segmented polygon in Fiji (ImageJ, 1.53p) and converted to micrometres squared. The radius and correlation quantification were subsequently extracted, and box plots and Pearson tests with 95% linear regression were plotted in Python3 vector graphics.

## Image acquisition and computational analysis for cell culture and GUV experiments

The dynamics of cPla$_2$ and ALPIN binding and lamin B1 (Lmnb1) and ER structures were measured using a Nikon Eclipse Ti inverted spinning-disc confocal microscope with a Plan Apo 100×/1.45 oil objective, a Plan Apo 60×/1.4 oil objective or a Plan Apo 40×/0.95 air objective (cells were imaged with a 60× or 100× objective, while GUVs were imaged with a 40× objective) and equipped with either (1) a Yokogawa CSU-X1 spinning-disc unit, an ANDOR iXon ULTRA 897BV EMCCD camera, 405 nm, 488 nm and 561 nm solid-state laser lines (Andor Revolution XD, LCS-501A) and NIS Elements Software (4.13.04) or (2) a Yokogawa CSU-W1 spinning-disc unit, a Teledyne Photometrics Prime-BSI sCMOS camera and 405, 488 and 561 nm multidiode AOTF-controlled laser modules (Nikon Instruments, model LUN-F; F6A 250 V) with NIS-Elements software (version 5.21.03). mKate2 and 18:1 Liss Rhod PE emission was excited at 561 nm and collected through a 620 band-pass emission filter (590–650 nm, Chroma Technology., 49005, ET-DsRed; 88000v2 Quad Set). eGFP fluorescence was excited at 488 nm and collected through a 525 band-pass emission filter (515–555 nm, Chroma Technology, 49002, ET-GFP;88000v2 Quad Set). The Sytox blue fluorescence was excited at 405 nm and collected through a 425 band-pass emission filter (435–478 nm, Chroma Technology, 49000, ET-DAPI;88000v2 Quad Set). Identical microscope settings were used throughout imaging. Experiments involving GUVs and permeabilized cells took place at room temperature. Live imaging of cultured cells was performed at 37 °C. Nikon's Perfect Focus System was enabled throughout the imaging session. For short-term time lapse and permeabilized cell imaging experiments, z-stacks (1–1.7 μm steps over 30 μm) were acquired each minute for a maximum of 20 min. For extended time-lapse experiments, z-stacks (2 μm over 40 μm) were acquired every 2 min for total of 6 h (frames 181, step t interval 2 min) in Nikon NIS elements (5.21.03). For the ML162 treatment, the z-stacks (1 μm step over 40 μm) were acquired every 10 min. For single time point acquisition, single z-stacks (1–1.4 μm steps over 30 μm) were acquired immediately after the end of the incubation. For high-resolution imaging of the ER network, we used a structured-illumination microscope (27.5 μm G5). Images were acquired using Elyra 7 Zeiss Microscope with a 63× objective (Plan-Apochromat 63× NA 1.4 ∞/0.17 WD 0.19 mm oil, 420782-9900-000 Zeiss) oil immersion lens. Emission was excited at 488 and 561 nm with beam splitters (SBS LP 560) and a laser-blocking filter (LFB 405/488/561/642) and sampled through 495–550 nm (BP 495–550) and 570–620 nm (LP 570) band-pass filters. Each emission for red and green were collected on two separate sCMOS camera (Excelitas technology, pco.edge 4.2 M) and aligned during pre-processing in Zeiss software (Zen Black 3.0 SR FP2, v16.0.20.306). A single z-stack (0.8 μm steps over 30 μm) was acquired at the end of the experiment. All images were taken in the identical microscope settings. The SIM reconstruction was conducted using the Zeiss Software (Zen Black 3.0 SR FP2). All 3D segmentation and analysis were performed with the Anaconda distribution of Python (Python ≥3.11). Specifically, customized Python scripts were developed using the Numpy (v1.23.5)[57], Scipy (v1.15.2)[58], Scikit image (v0.20.0)[59], Allen Cell Structure Segmenter (v0.5.0)[60], trackpy (v0.6.4)[61,62] and Napari (v0.5.5) libraries[63].

To quantify binding of ALPS1–2–eGFP to GUV membranes, a custom, semi-automatic Python script was developed. First, a line was manually drawn by the user from the centre of GUV to the edge to crop a selected GUV from the whole image stack. Next, intensity normalization and a 3D Gaussian blur (3-pixel radius) were applied to the cropped GUV stack to remove background noise. A marker-controlled watershed algorithm was used on each slice of the cropped stack to segment the GUV. The slice with largest segmented GUV area was defined as the middle section. Based on the watershed segmentation result, a three-pixel-wide contour of the GUV was drawn onto the GUV middle section. Last, the rim-binding intensity was calculated as the median of all intensities along the contour (Extended Data Fig. 6a).

To quantitatively measure the affinity of ALPS1–2–eGFP to GUV, the apparent dissociation constant $K'_d$, maximum binding intensity $B_{max}$ and the apparent Hill coefficient $H$ were determined by nonlinear least-square analysis of bound-protein fluorescence ($B_{bound}$) versus the total domain concentration ([Domain]) using the Langmuir adsorption isotherm with Hill expansion (MATLAB Curve Fitting Toolbox v25.1 R2025a)

$$B_{bound} = B_{max} \left( \frac{[Domain]^H}{[Domain]^H + {K'_d}^H} \right).$$

A similar, fully autonomous Python pipeline was developed to segment the nucleus in 3D and quantify both the volume of nucleus and bindings of cPla$_2$–mKate2 and ALPIN sensor. If cells expressed eGFP–Lmnb1, this marker was used for nuclear segmentation. Otherwise, cPla$_2$–mKate2 or the ALPIN sensor was used for segmentation. First, intensity normalization, 3D Gaussian blur (1-pixel radius) and rolling-ball algorithms were applied to the images to remove uneven background noise. Second, a masked-object thresholding (this algorithm was employed to separate foreground objects from the background of the 3D image stack through an automatically determined threshold and to generate a marker image that distinguishes individual nucleus) followed by marker-controlled watershed algorithms were used to segment the NE within the 3D image stack. Since the movement of cell nuclei was limited (less than 50 μm) during the duration of the experiment, a standard particle-tracking algorithm was utilized to track individual nucleus over time. Nuclear volume was calculated by the summation of all voxels contained within the segmented binary mask. Ratiometric analysis was used to quantify cPla$_2$–mKate2 and ALPIN nuclear membrane-binding dynamics. For each nucleus, the slice with largest segmented nuclear area was selected as the representative, middle section for each nucleus. Based on the watershed segmentation result of the middle section, a 3-pixel-wide contour ('membrane contour') was drawn onto the NE and another 3-pixel-wide contour ('background contour') was drawn to mark the centre of the nucleus (nucleoplasm). The binding was calculated by dividing the medians of all intensities along the 'membrane contour' by the median of all intensities along the 'background contour' in the channel where protein fluorescence was measured. One caveat for this type of imaging analysis is that an increase of binding ratio could be due to a decrease of intensity in nucleoplasm instead of an increase in protein binding to the NE. To account for this issue, an alternative ratiometric binding was calculated between the median intensity along the 'membrane contour' of the 561 channel (protein of interests tagged with mKate2) and the median fluorescence intensity across the segmented nuclear middle section in the 488 nm channel (the eGFP-tagged Lmnb1 nuclear marker, whose intensity remained constant throughout the imaging session). A detailed scheme of nuclear membrane binding and volume analysis is shown in Extended Data Fig. 6b.

To quantify individual NE shrivelling/invaginations, a filament-based filter algorithm was performed on the previously selected middle section from the eGFP–Lmnb1 image stack to create a binary mask that preserved the detailed structures of nuclear membranes. Note, to preserve nuclear membrane details and minimize processing-induced artefacts, only Gaussian blurring was applied to the eGFP–Lmnb1 stack before applying the filament-based filter algorithm. The thickness parameter for the filament-based algorithm was set to 1.6 (~3–4 pixels). This was followed by a skeletonization algorithm to produce a precise 1-pixel outline of NE. Shrivelling often created branch points in the skeleton images. Therefore, the degree of nuclear invagination was approximated as the number of branch points in the skeleton outlines, which can be detected if a particular skeleton pixel contained more than three neighbours. Finally, a condensation algorithm was utilized to merge multiple nearby branch points that appeared owing to noise affecting accuracy of skeleton outlines. A scheme of the NE invagination analysis is shown in Extended Data Fig. 7a.

To quantify ER structure, a Python-based automatic image analysis pipeline was developed. ER images or movies were first pre-processed with intensity normalization and 3D Gaussian blur (1-pixel radius). The segmentation algorithm of choice was slightly different depending on the specific tag used to label the ER. With luminal labelling (for example, eGFP–KDEL), ER was segmented from the 3D stack by a marker-controlled watershed algorithm the same way as NE segmentation. If ER membranes were labelled (for example, SEC61B), the image was processed by a filament filter algorithm that segments dense filamentous and network-like structures in the cell. The algorithm's thickness parameter was set to 1.6, corresponding to ER networks or vesicles approximately 3–4 pixels thick. For 3D ER metrics, calculations were technically and optically challenging. The slice with largest segmented area along the $z$ axis was selected as the representative mid-section for downstream quantifications to compute surface area and circularity. The surface area of each segmented ER membrane in the mid-section was extracted from the corresponding segmentation mask and the circularity was calculated as Circularity $= 4\pi \times \frac{\text{Surface area}}{\text{Perimeter}^2}$ (0 for a highly non-circular object and 1 for a perfect circle). For each time point, ER surface area and circularity were measured for all individually segmented ER membranes within a single FOV. To enable the quantitative comparison across samples and time, these measurements were summarized by recording the largest individual ER surface area and the mean circularity of all ER membranes in that FOV. This approach produced one surface area value and one circularity value per FOV at each time point, which were then used for normalization, plotting and statistical analysis. As ER membranes from neighbouring cells were often in close proximity and plasma membrane labelling was absent, segmentation at the single-cell level was not feasible; therefore, ER metrics were quantified per FOV. A detailed scheme of quantifying ER circularity and area are provided in Extended Data Fig. 7b.

## Statistics and reproducibility

To ensure the comparability of results obtained across different experimental batches and days, multiple normalization methods were applied. For GUV experiments, raw rim binding of each domain was normalized by the fluorescence signal of a 1 μM protein solution from the same purification batch to reduce variabilities in fluorescence from different purification batches. For time-lapse experiments, numerical values for the fluorescent marker nuclear rim-to-centre ratio or rim-to-nuclear Lmnb1 (for example, cPla$_2$, ALPIN and ALPIC), ER surface area, circularity, nuclear volume and nuclear folds at each subsequent time point were normalized to the corresponding value at $t = 0$ min. For FLIP experiments, the intensity of fluorescent ER markers at each subsequent time point was similarly normalized to the $t = 0$ min value. Notably, although ER circularity values inherently range from 0 to 1, normalization to the mean circularity at $t = 0$ min shifted the range to 1–2, such that a value of 1 represents the baseline circularity and values greater than 1 indicate an increase relative to baseline. For single time point osmotic shock experiments (Figs. 1a right, and 2c and Extended Data Fig. 1a), cPla$_2$ and ALPIN1 nuclear rim-to-centre ratio, ER area and ER circularity under hypo-osmotic condition values were normalized to the corresponding measurements obtained under isosmotic condition. For single time point permeabilization experiments (Extended Data Fig. 2g,h), values for ER circularity, ER area and ALPIN1 nuclear rim-to-centre ratios were normalized to measurements obtained under 0 μM Ca$^{2+}$ and osmotic-balanced (2.5% PVP) conditions. As different normalization schemes were applied to time-lapse and single time point datasets, the resulting value ranges differed and they were not directly comparable.

The results presented in this study were performed as multiple, biologically independent experiments, as indicated in figure legends, and no inconsistent results were observed. For experiments involving cell cultures, biological replicates were obtained using cells derived from distinct frozen vials and/or cultured at different times.

For experiments involving GUVs, biological replicates were obtained using GUVs synthesized from independently prepared lipid mixtures. In animal experiments, each larva was considered to be an independent biological replicate, and at least six animals were used per imaging modality (spinning-disc confocal and two-photon microscopy).

For the fitted coefficients of ARFGAP1 ALPS1-2–eGFP GUV equilibrium binding, the minimum and maximum values denote the 95% confidence interval (CI), while the error bars for individual GUV binding measurements indicate s.d. For FLIP experimental results, the shaded area represents the 95% CI. For time-lapse comparisons of ER circularity and area, nuclear volume and folds, and cPla$_2$ and ALPIN binding, the error bars denote the s.e.m. For single time point comparisons, error bars represent the s.d. The statistical analysis was performed in Python (v3.11) using Scipy (v1.15.2) and Pingouin (v0.5.5) packages[58]. The Shapiro–Wilk test and the Levene test were used to test for normality and equal variance between groups, respectively. For comparing statistical significance between two groups, a parametric method (an unpaired two-tailed Student's t-test if two groups have equal variance, otherwise an unpaired two-tailed Welch's t-test) was used if the data passed the normality test. A non-parametric method (two-tailed Mann–Whitney $U$ test) was used for non-normally distributed data. For the comparison of more than two groups, a one-way Welch's ANOVA (data did not pass Levene's test for equal variance) followed by a post hoc Games–Howell test were used if the data followed normal distribution. Otherwise, statistical significance was calculated with the Kruskal–Wallis test followed by a post hoc Dunn's test for multiple comparison. $P$ values ≤0.05 were accepted (*$P \le 0.05$, **$P \le 0.01$, ***$P \le 0.001$ and ****$P \le 0.0001$). ALPIN binding rates in permeabilized nuclei were estimated via ordinary least squares (OLS) on the linear part of the binding curves from Python's Statsmodel package[64]. Specifically, OLS regression models were fitted in Python 3.11 (statsmodels v0.14.4). Since there are only two time points ($t = 4$ min and $t = 5$ min), assumptions for linearity and multicollinearity were not applicable. The Breusch–Pagan test was used to assess heteroscedasticity; the Shapiro–Wilk test and $Q$–$Q$ plots of standardized residuals were used to evaluate normality of residuals. If either test returned a $P < 0.05$, the model was re-estimated with heteroscedasticity-consistent (HC3) robust standard errors; otherwise, classical OLS standard errors were retained. The Durbin–Watson statistic was used to evaluate autocorrelations between variables. Estimated coefficients, s.e.m., 95% CIs, coefficient of determination ($R^2$ and adjusted $R^2$) and $P$ value are reported in Source data. Graphical plots were generated using Python 3.11 Matplotlib (v3.10.3)[65] and seaborn (v0.13.2) libraries[66]. The helical-wheel representation of ALPS1 motif structure was created using the Python 3.11 Matplotlib library. The equilibrium GUV fluorescence-binding graph was generated using Curve Fitting Box (v25.1) from MATLAB R2025a. Since the ER circularity/area was quantified per FOV while the binding ratio of cPla$_2$–mKate2 and the ALPIN sensor was calculated per nucleus, different cells within the same FOV were given the same ER circularity/area value in plotting the scatter plot in Figs. 1a and 2c. All plots and statistical analyses were generated using the standard packages listed above (version listed), with no additional custom plotting code.

No statistical methods were used to predetermine sample size or effect size. Statistical analyses were performed across biological replicates. Replication included both variation in biological source material and/or independent imaging sessions conducted on separate experimental days. Analysis was performed objectively without blinding.

Time-lapse U2OS cell experiments were performed with 2–5 biological replicates, with key experiments repeated ≥3 independent replicates to ensure reproducibility. Single time point U2OS cell assays were conducted in one independent experiment using cells from two independent frozen vials. Equilibrium GUV experiments were conducted from two independently prepared lipid mixtures. For zebrafish imaging, each experiment represents a biologically independent replicate using animals derived from a distinct clutch of eggs.

Experiments were repeated across multiple days, with one larva imaged per experiment for each UV- or infrared-induced tissue-wounding condition and repeated on more than six different live larvae to ensure reproducibility.

Data exclusion criteria were predefined. Imaging datasets were excluded if they were affected by artificial fluorescence fluctuations (for example, immersion oil bubbles) or incompletely acquired $z$-stacks, and broken tracks were either cropped or completely discarded. All raw imaging files of cell experiments were loaded into custom written automatic programs for analysis, unless otherwise stated. For the nuclear membrane analysis program, segmented objects with a size less than 8,000 pixel$^2$ (~140 μm$^2$) and/or touching the borders of the image boundary were excluded from the analysis. Nuclei were removed from the analysis when segmented objects moved more than 130 pixels (~20 μm) between successive frames or broken tracks appeared due to leakage of nucleoplasmic fluorescent markers (cPla$_2$–mKate2 and ALPIN) when these markers were used for nuclear segmentation. Multiple tracks that were detected as single were deleted from the analysis. For the ER analysis program, segmented ER vesicles size less than 30–50 pixel$^2$ (~1 μm$^2$) were precluded from the analysis. Since the ER was analysed per FOV, no tracks were involved, and all time points were included in the analysis unless ER segmentation was precluded by the leakage of eGFP–KDEL. In long-term cell imaging experiments, datasets were excluded only when imaging artefacts prevented reliable analysis and data extraction. Specifically, exclusions were made if (1) fluorescent markers leaked from their designated organelle localization into the surrounding cytoplasmic space during acquisition, (2) cells became clumped or (3) automated tracking produced broken or fragmented tracks. For GUV experiments, the raw imaging files were loaded into a semi-automatic custom written analysis program where the individual GUV was randomly selected by the user for analysis. For zebrafish intravital imaging, datasets were excluded if tailfin tissue drift exceeded 100 pixels (~50 μm) in the $xy$ plane or 20 pixels (~10 μm) in the $z$ direction following UV-laser damage.

No data were excluded based on distribution (no outlier exclusion). Typically, experiments were repeated at least on two separate days (using cells thawed from different vials). Only the experiments shown in Figs. 1a, right, and 2c,e and Extended Data Figs. 1a and 2a,g,h were not replicated on different days; nevertheless they were conducted using cells or GUVs prepared from at least two independent stock sources, that is, storage vials or lipid mixtures, respectively. The details of statistical analyses used, exact $P$ values, number of biological independent experiments and sample sizes for all graphs are listed in the figures or figure legends. The investigators were not blinded during the experiments and analysis. Specimens (U2OS cells, GUVs and zebrafish larvae) were randomly chosen for imaging without bias or pre-selection.

### Reporting summary
Further information on research design is available in the Nature Portfolio Reporting Summary linked to this article.

## Data availability
Data supporting this work are available in this article and Supplementary Information. All other data supporting the findings of this study are available from corresponding author upon reasonable request. Specifically, the time-lapse and super-resolution imaging data can be made available only upon request due to its proprietary file formats, very large file sizes (>10 TB) and repository storage limitation. The plasmids, cell lines and zebrafish strains created in this study are also available upon reasonable request. Source data are provided with this paper.

## Code availability
The code for ER FLIP analysis and viewing 3D super-resolution segmentation masks is available as an open-source download via GitHub at https://github.com/zazadovv/ER_Nucleus.git. The code for analysing GUV protein binding and adsorption fitting is publicly available via GitHub at https://github.com/joeshen123/GUV-Protein-Binding-Analysis-Program.git. The code for analysing protein binding to the nuclear envelop, nuclear volume and invaginations is available via GitHub at https://github.com/joeshen123/Nuclear-Membrane-Binding-Analysis.git. The code for analysing ER areas and curvatures is available via GitHub at https://github.com/joeshen123/ER-Structure-Analysis.git.

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

## Acknowledgements
This research was supported by the NIH/NIGMS grant R35GM140883 to P.N., and in part by the NIH/NCI Cancer Center Support grant P30CA008748. Z.S. was supported by a Bruce Charles Forbes Graduate Fellowship. Z.G. was supported by Harold E. Varmus Graduate Fellowship. We thank M. Lengyel, Y. Ma and S.G. Rajan for editing and insightful comments on the draft.

## Author contributions
P.N., Z.S. and Z.G. conceived of the study and designed the experiments. Z.S. and Z.G. conducted the experiments. Z.S. developed the image analysis scripts and analysed the data. Z.S., Z.G. and P.N. prepared the figures. Z.S., Z.G. and P.N. wrote the paper.

## Competing interests
The authors declare no competing interests.

## Additional information
**Extended data** is available for this paper at https://doi.org/10.1038/s41556-025-01820-9.

**Correspondence and requests for materials** should be addressed to Philipp Niethammer.

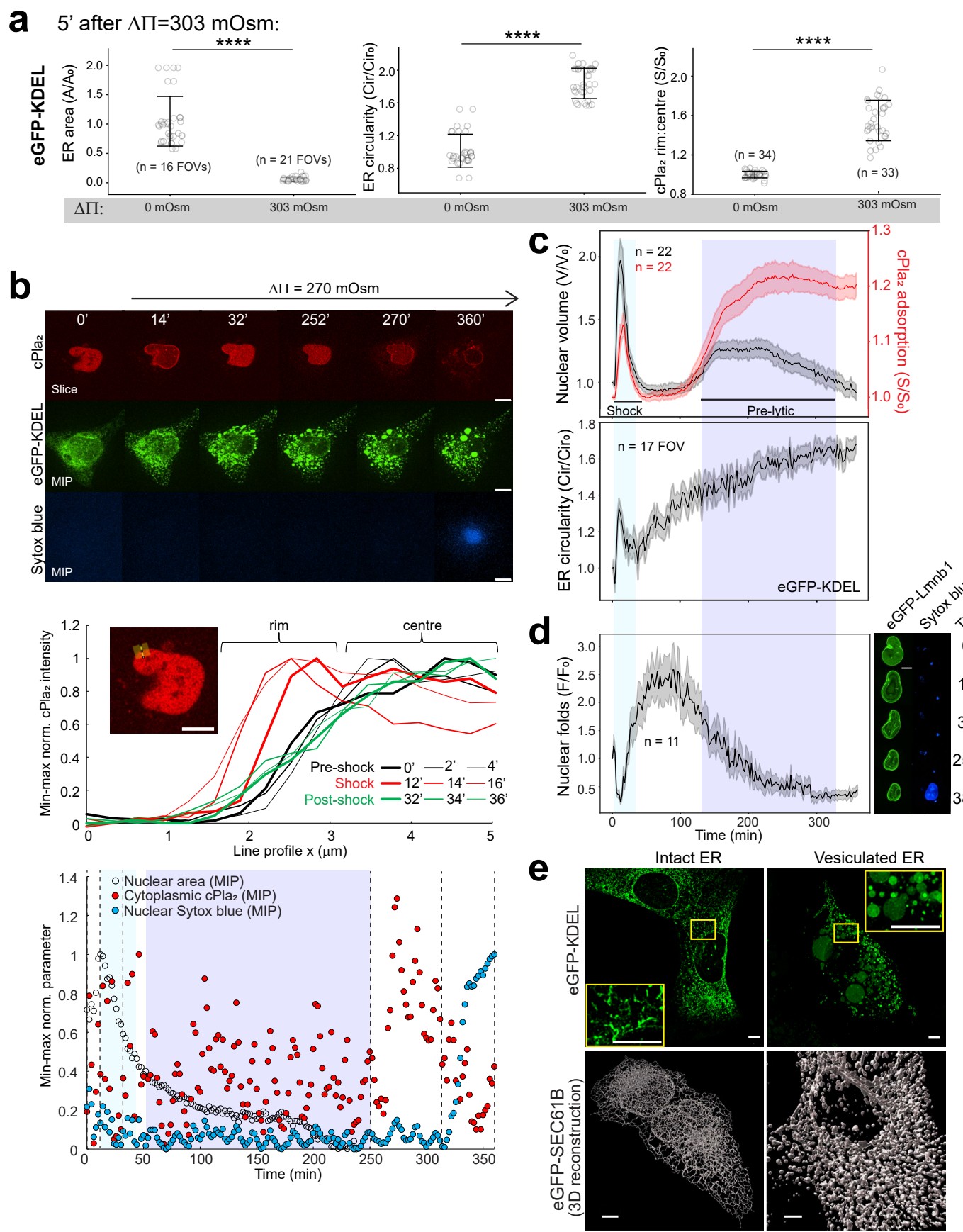

**Extended Data Fig. 1 | See next page for caption.**

**Extended Data Fig. 1 | ER, cPla$_2$ and Lmnb1 dynamics in hypoosmotic shock.**
**(a)** Left panel, ER network area of U2OS cells at ΔΠ = 0 mOsm or 303 mOsm, respectively. Error bars, SD. n, FOVs. Middle panel, ER circularity of U2OS cells at ΔΠ = 0 mOsm or 303 mOsm, respectively. Error bars, SD. n, FOVs. Same FOVs were analyzed for normalized area and circularity. Right panel, cPla$_2$-mKate2-INM adsorption in cells in the above FOVs at ΔΠ = 0 mOsm or ΔΠ = 303 mOsm, respectively. Error bars, SD. n, number of nuclei analyzed. ****, p = 2.8 ×10$^{-7}$ for both the left and the middle graph and p = 7.8 ×10$^{-16}$ for the right graph as determined by a two-sided Mann Whitney U-test (data are not normally distributed) for both the left and middle graph and a two-sided Welch's $t$-test for the right graph. Note: Both Extended Data Fig. 1a and the right panel of Fig. 1a are different representations of the same data set. The data represent one independent experiment performed on cells from two distinct biological sources (separate frozen vials). **(b)** Top panel, representative time-lapse montage of cPla$_2$-INM adsorption, ER morphology (eGFP-KDEL), and cell lysis (Sytox blue) upon exposure to the hypotonic solution (ΔΠ = 270 mOsm). Middle panel, nuclear rim profiles (20 pixels wide) of cPla$_2$ midplane fluorescence at indicated location (inset, yellow line). Temporal order is indicated by line width and color code. Bottom panel, temporal profile of segmented nuclear area (open circles), cytoplasmic cPla$_2$ (ROI on extranuclear area, red circles), and nuclear Sytox blue signal (ROI on intranuclear area, blue circles) as determined from confocal MIPs. Scale bars, 10 µm **(c)** Top panel, time-lapse analysis of nuclear volume (black) and normalized cPla$_2$-INM adsorption (red). Light blue shade, osmotic shock and RVD phase. Dark blue shade, pre-lytic phase. Lines, average. Shaded error, SEM. n, number of analyzed nuclei. Bottom panel, corresponding time-lapse analysis of ER circularity (black). Lines, average. Shaded error, SEM. n, number of corresponding FOVs. The data represent three independent experiments. **(d)** Left panel, time-lapse analysis of nuclear envelope invaginations. Lines, average. Shaded error, SEM. n, number of analyzed nuclei. Right panel, representative confocal MIPs of eGFP-Lmnb1 and Sytox blue channel. Scale bar, 10 µm. The data represent two independent experiments. **(e)** The top panel, super resolution lattice-SIM$^2$-Apotome representative MIP image of U2OS cells for the ER lumen network at baseline (ΔΠ = 0 mOsm, n = 6) and 10 min after hypoosmotic shock (ΔΠ = 270 mOsm, n = 6). Bottom panel, the super resolution (XYZ, 60, 60, 150 nm) segmentation masks of eGFP-SEC61B at baseline (left, n = 4) and 10 min after hypoosmotic shock (right, n = 4). Scale bars, 5 µm. For time-lapse movies, values in subsequent time points were normalized to baseline at T = 0. For single time point images, values from hypoosmotic conditions were normalized to isosmotic controls. A, area. Cir, circularity. S, ratiometric signals. V, volume. F, folds. Statistical source data are provided in Statistical Source Data Extended Data Fig. 1.

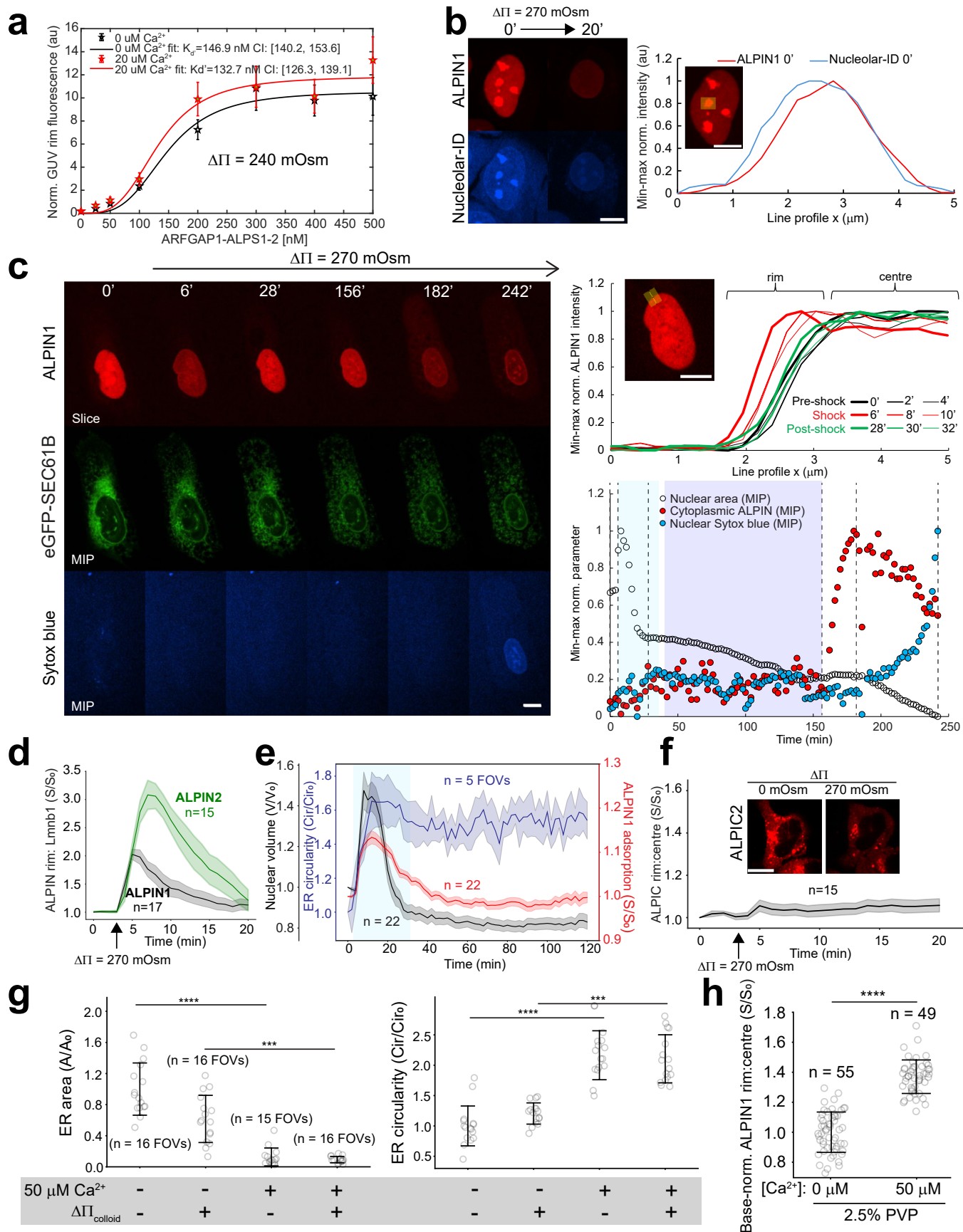

**Extended Data Fig. 2 | See next page for caption.**

**Extended Data Fig. 2 | Characterization of ALPIN1/2 in nuclear membrane tension sensing. (a)** Equilibrium GUV binding isotherms of ARFGAP1 ALPS 1–2-eGFP at hypotonic solution (ΔΠ = 240 mOsm) containing 0 µM (black lines) or 20 µM (red lines) Ca$^{2+}$. Error bar, SD of normalized single GUV binding measurements per domain concentration. Binding isotherms were derived by titrating ALPS 1-2-eGFP (0–500 nM) on GUVs under hypotonic conditions (ΔΠ = 240 mOsm) in the presence of either 0 or 20 µM Ca$^{2+}$. For each condition, individual GUVs counts were analyzed as follows (0/20 µM Ca$^{2+}$):n = 47/41 (0 nM), 14/28 (25 nM), 28/39 (50 nM), 57/54 (100 nM), 61/73 (200 nM), 87/86 (300 nM), 89/73 (400 nM), and 74/64 (500 nM). Fitted lines, Langmuir equations with Hill expansion fit (methods, GUV equilibrium binding experiments). Legend, apparent dissociation constants (Kd') and associated 95% confidence interval determined from the above Langmuir/Hill equations fitting. N, number of GUVs analyzed. The data represent one independent experiment using GUVs from two distinct lipid mixtures. **(b)** Left panel, representative confocal MIPs depicting nucleolar colocalization of ALPIN1 and nucleolus marker (Nucleolar-ID) acquired before and 15 min after ΔΠ = 270 mOsm shock. Right panel, profile plot quantification (see inset). Inset scale bar, 10 µm. **(c)** Top panel, representative timelapse montage of ALPIN1–INM adsorption, ER morphology (eGFP-SEC61B), and cell lysis (Sytox blue) upon exposure to the hypotonic solution (ΔΠ = 270 mOsm). Top right panel, nuclear rim profiles (20 pixel wide) of ALPIN1 midplane fluorescence at indicated location (inset, yellow line). Temporal order is indicated by line width and color code. Bottom right panel, temporal profile of segmented nuclear area (open circles), cytoplasmic ALPIN1 (ROI on extranuclear area, red circles), and nuclear Sytox blue signal (ROI on intranuclear area, blue circles) as determined from confocal MIPs. Inset scale bar, 10 µm. **(d)** Timelapse comparison of ALPIN1- (black) and ALPIN2- (green) INM adsorption upon ΔΠ = 270 mOsm shock as normalized by Lmnb1 fluorescence across the nuclear

midsection. Lines, average. Shaded error, SEM, n, number of nuclei analyzed. Note: Both Fig. 2b and this figure present different type of analysis performed on the same ALPIN2 movie. Both ALPIN1 and ALPIN2 were co-expressed with eGFP-Lmnb1. The data for ALPIN1 and ALPIN2 represent three and two independent experiments, respectively. **(e)** Timelapse of ER circularity (blue), nuclear volume (black), and ALPIN1 adsorption (red). n = nuclei or FOVs. Lines, average. Shaded error, SEM. The data represent two independent experiments. **(f)** Inset, representative confocal midplane-slices of ALPIC2 (ALPS1-2-mKate2)-ONM adsorption acquired before and after (t = 5 min) ΔΠ = 270 mOsm shock. Scale bar, 20 µm. Bottom, timeseries quantification. Line, average. Shaded error, SEM. n, number of nuclei analyzed. The data represent 4 independent experiments. **(g)** Quantification of ER area (left panel) and circularity (right panel) after digitonin-permeabilization as a function of [Ca$^{2+}$] and [PVP360]. Error bars, SD. ****, p = 4.8 ×10$^{-7}$, ***, p = 0.0004 for ER area; ****, p = 1.8 ×10$^{-7}$, ***, p = 0.0005 for ER circularity, as determined by a Kruskal–Wallis Test followed by post-hoc Dunn's tests (the data was not normally-distributed). n, number of FOVs analyzed (containing ~1-10 cells each). The data represent one independent experiment performed with cells from two distinct biological sources. **(h)** Normalized ALPIN1-INM adsorption 30 min after permeabilization with 0 or 50 µM Ca$^{2+}$. Error bar, SD. ****, two-sided Welch's t-test (unequal variance): p = 3.9 ×10$^{-28}$. n, number of nuclei analyzed. The data represent one independent experiment performed with cells from two distinct biological sources. For time courses, data normalized to T = 0; GUV data normalized to 1 µM ALPS1-2-eGFP control. For single timepoint permeabilization experiments, values were normalized to measurements obtained under 0 µM Ca$^{2+}$ and osmotic-balanced (2.5% PVP) conditions. A, area. Cir, circularity. S, ratiometric signals. V, volume. Statistical source data are provided in Statistical Source Data Extended Data Fig. 2.

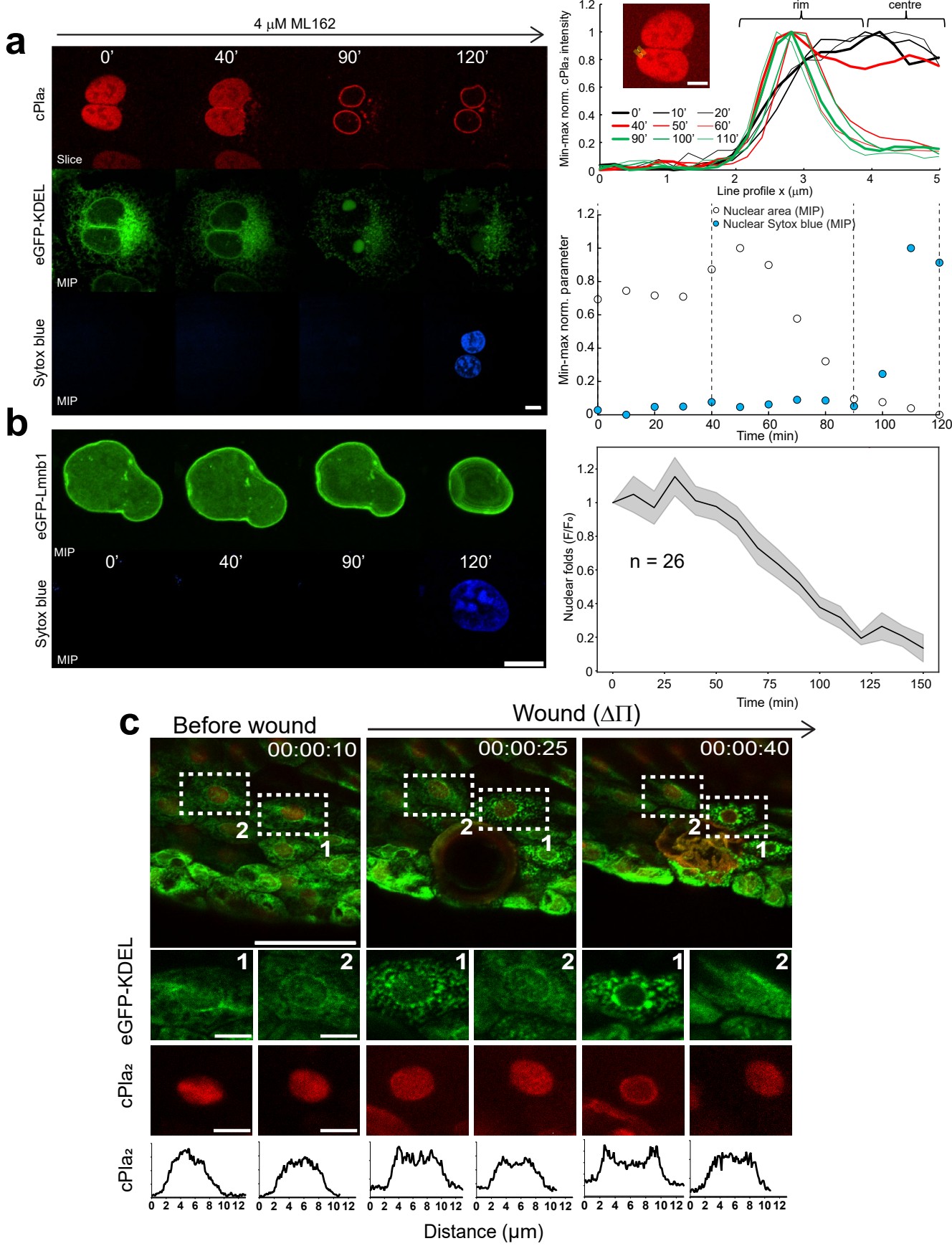

**Extended Data Fig. 3 | See next page for caption.**

**Extended Data Fig. 3 | ER network disruption in ferroptotic cell death and tissue wounding. (a)** Left panel, representative timelapse montage of cPla$_2$-INM adsorption, ER morphology (eGFP-KDEL) and cell lysis (Sytox blue) in a cell undergoing ferroptotic cell death upon exposure to the GPX4 inhibitor ML162. Scale bar, 10 μm. Top right panel, nuclear rim profiles (20 pixel wide) of cPla$_2$ midplane fluorescence at indicated location (inset, yellow line). Temporal order is indicated by line width and color code. Bottom right panel, temporal profile of segmented nuclear area (open circles), cytoplasmic cPla$_2$ (ROI on extranuclear area, red circles), and nuclear Sytox blue signal (ROI on intranuclear area, blue circles) as determined from confocal MIPs. **(b)** Left panel, representative timelapse montage of eGFP-Lmnb1 and cell lysis (Sytox blue) in ferroptotic cell death with GPX4 inhibitor ML162. Right panel, time-lapse analysis of nuclear envelope invaginations. Shaded error, SEM. n, number of analyzed nuclei. Scale bar, 10 μm. The data represent two independent experiments. F, folds. **(c)** Two-photon MIPs of cPla$_2$-mKate2 and luminal ER marker eGFP-KDEL in live zebrafish larvae at baseline (before wounding) and after wounding under hypoosmotic conditions – larvae are wounded (Π - 10 mOsm) at t = 00:00:25. Scale bar, 50 μm. Middle panels, magnification of eGFP-KDEL or cPla$_2$ in individual cells within (position 1) and just outside (position 2) the wound region. Inset scale bars, 10 μm. Bottom panels, profile plots of cPla$_2$-mKate2 emission of the depicted nuclei. Constitutive cPla$_2$ translocation with ER fragmentation was observed in every (n = 10) wounding experiments. Statistical source data are provided in Statistical Source Data Extended Data Fig. 3.

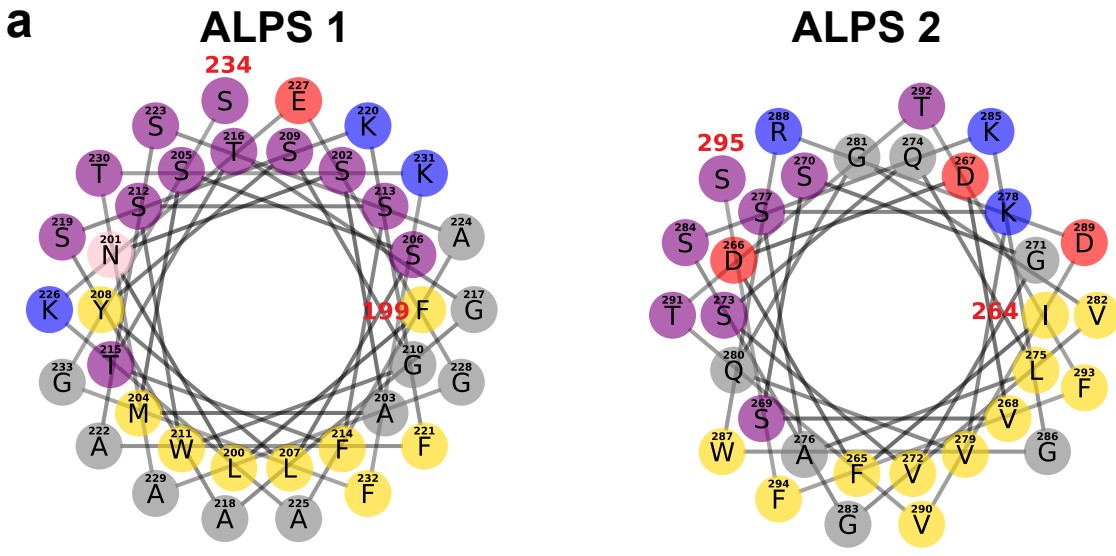

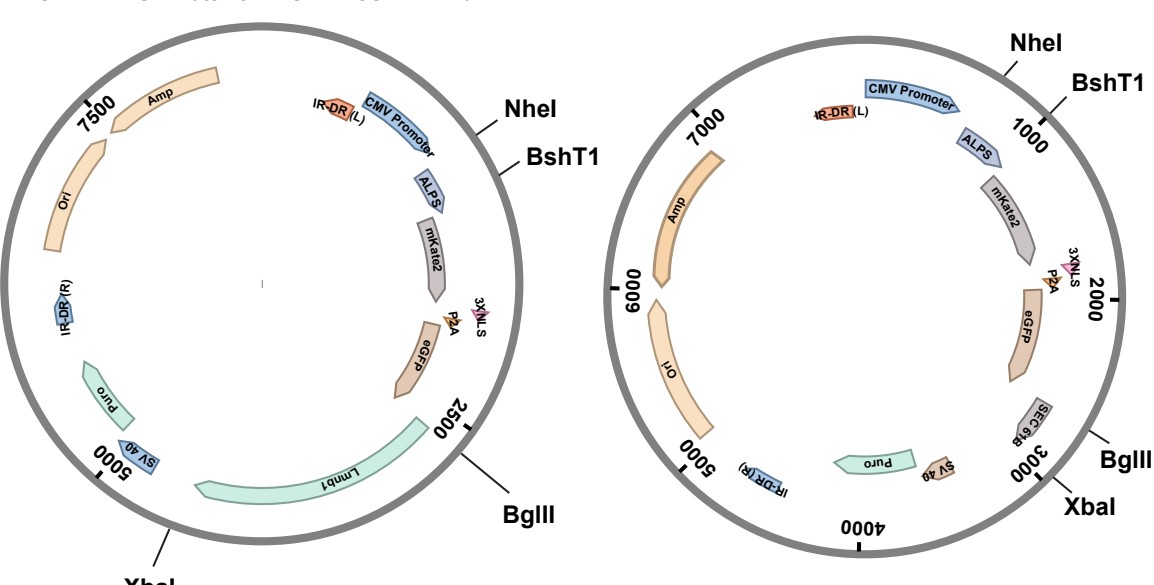

**Extended Data Fig. 4 | ALPS helical wheel and plasmid map. (a)** Helical wheel representation of ARFGAP1 ALPS 1 [199-234] and ALPS 2 [264-295] motifs. Yellow, hydrophobic; purple, serine and threonine; grey, glycine and alanine; blue, basic residues; red, acidic residues. **(b)** Circular plasmid maps of PSB/CMV/ MCS constructs used to express ALPIN 1 or ALPIN 2 in U2OS cells together with nuclear (Lmnb1, left) or ER (SEC61B, right) markers (both protein sequences are linked via P2A sequence). To generate the constructs containing both ALPIN and nuclear marker, ARFGAP1 ALPS1 and ALPS 1-2 were inserted into the vector

containing both eGFP-Lmnb1 and NLS tagged mKate2 sequences via NheI and BshT1 restriction sites. To generate the constructs containing both ALPIN and ER markers, the Lmnb1 sequence from the previously generated ALPIN-P2A-eGFP-Lmnb1 (left) was replaced by SEC61B via BgIII and XbaI restriction cut sites. The expression of the Puromycin resistant genes was under control of a SV40 promoter. Two inverted internal repeats sequences (IR-DR) flank the expression cassettes and allow the integration into host genomes when co-transfected with *Sleeping Beauty* transposase.

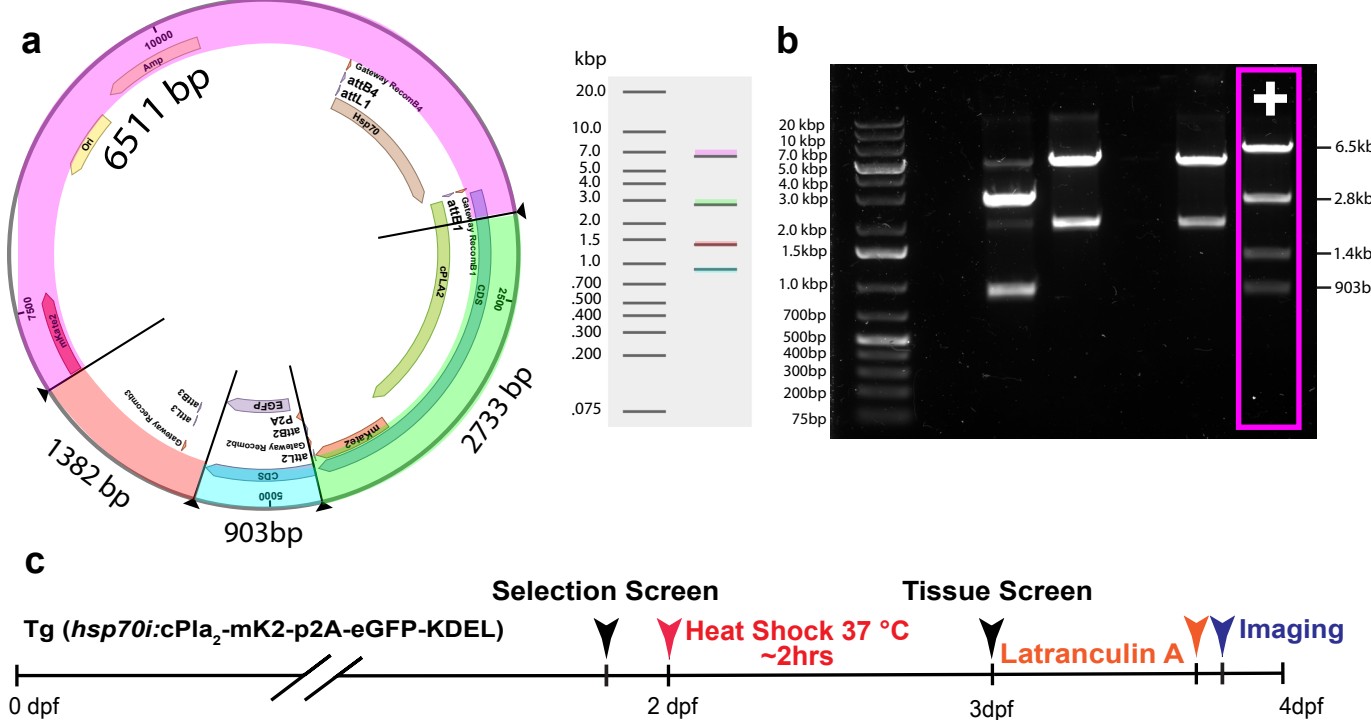

**Extended Data Fig. 5 | Transgenesis and scheme for heat-shock induced expression. (a)** Left panel, circular DNA map of zebrafish transgenesis plasmid, pDest cryybb1:mKate2-P5E-HSP70i_pME-Ctd-mK2-cPla₂_P3E-P2A-eGFP-KDEL overlaid with predicted restriction enzyme, BamHI digest cut sites as indicated by black arrow heads and black lines. Right panel, legend of virtual restriction enzyme digest predicting 4 bands, the color map corresponds to digest fragment size (magenta, 6511 bp; green, 2773 bp; red 1382 bp; cyan, 903 bp). **(b)** The DNA gel showing the restriction enzyme digest bands, magenta highlights the predicted DNA pattern shown in (a). **(c)** Scheme for laser wounding, Zebrafish Casper embryos are reared and selected for fluorescent selection marker, such as red eye. The embryos were heat-shocked in water bath at 37 °C for 2 h and subsequently returned to 28 °C. The following day, larvae are selected for pharmacological pre-treatment with Latrunculin A for 20-30 min and then wounded during confocal imaging with UV laser.

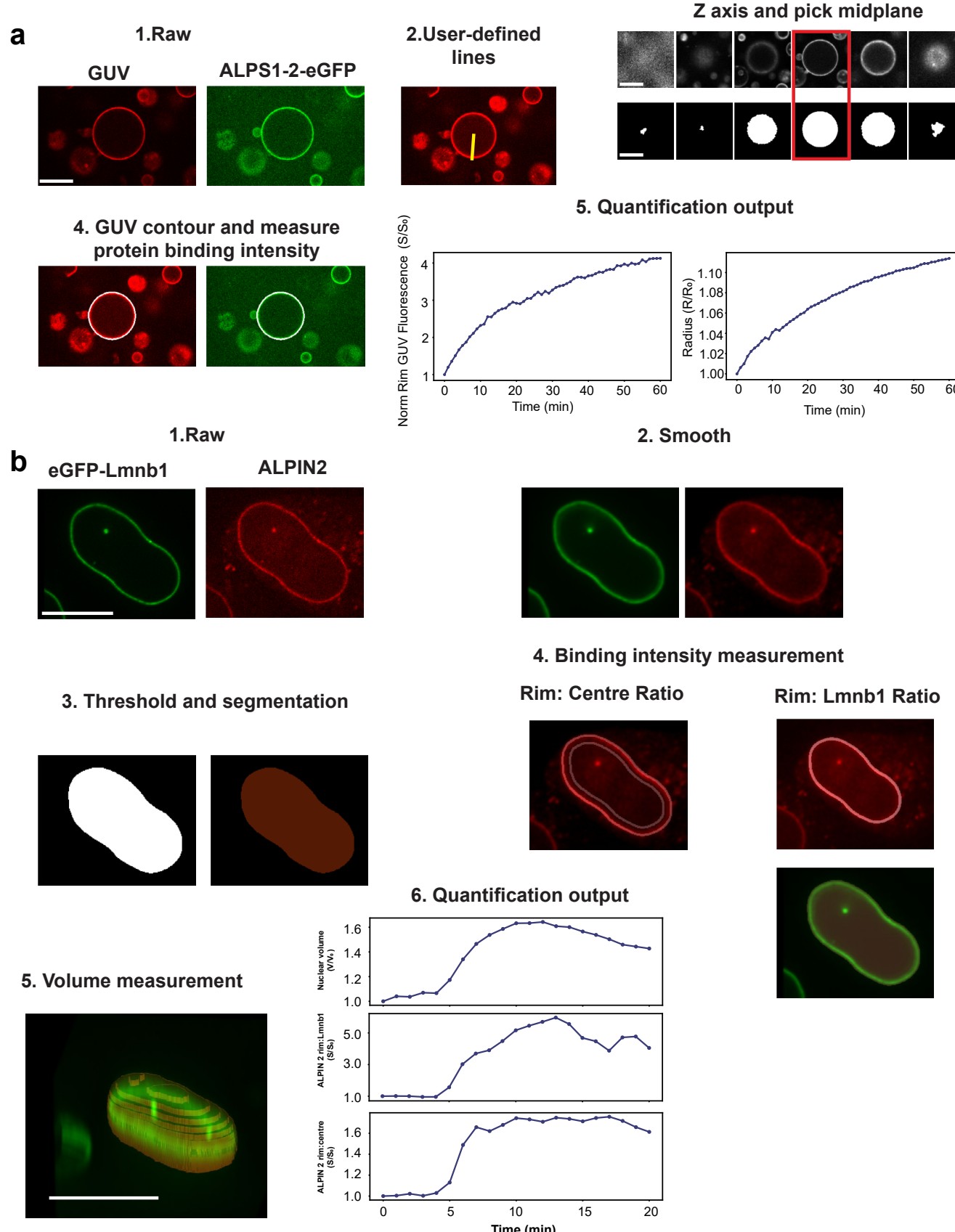

**Extended Data Fig. 6 | See next page for caption.**

**Extended Data Fig. 6 | Scheme of analysis for protein-membrane adsorption.**
**(a)** Overview of GUV membrane and ALPS1-2-eGFP adsorption analysis. Image stacks of GUV membrane (561 nm) and ALPS 1-2-eGFP (488 nm) were acquired, followed by manual selection of a radial line from the GUV centre to the membrane edge for automatic cropping (Step 2). Afterwards, a watershed segmentation algorithm was applied to the 3D cropped stack (Step 3), and the slice with the largest segmented area along z-axis (Midplane slice, Step 3 middle image with red edge) was selected to generate a membrane contour (white outline in Step 4) for quantifying GUV radius and ALPS 1-2-eGFP binding intensity. Quantified outputs were plotted and exported for downstream analysis (Step 5). Scale bar, 20 μm. R, radius. S, adsorption signals. **(b)** Overview of nuclear membrane volume and ALPIN/cPla$_2$ binding analysis. Here, a eGFP-Lmnb1 and ALPIN2 labelled U2OS cell undergoing 270 mOsm hypoosmotic shock was chosen as an example. Raw 3D image stack was processed by gaussian smoothing and intensity thresholding (Step 2), followed by masked-object thresholding and segmentation of individual nuclei using a marker-controlled watershed algorithm (Step 3). The mid slice with largest segmented nuclear area was chosen to generate membrane contour for downstream analysis. Two strategies were used to quantify protein binding to the nuclear membrane (Step 4): (1). a rim:centre ratio comparing binding intensity at the membrane contour with that at the inner outline marking the nucleoplasm (Step 4 left panel); and (2) a rim: Lmnb1 ratio comparing binding intensity at the contour (561 nm channel, Step 4 right panel top image) with Lmnb1 intensity across segmented nuclear area at 488 nm channel (Step 4 right panel bottom image). Nuclear volume was calculated by summing all voxels within the segmented stack (Step 5). Binding ratio and nuclear volume were plotted and exported for downstream analysis (Step 6). Scale bar, 20 μm. V, volume. S, ratiometric signals. Statistical source data are provided in Statistical Source Data Extended Data Fig 6.

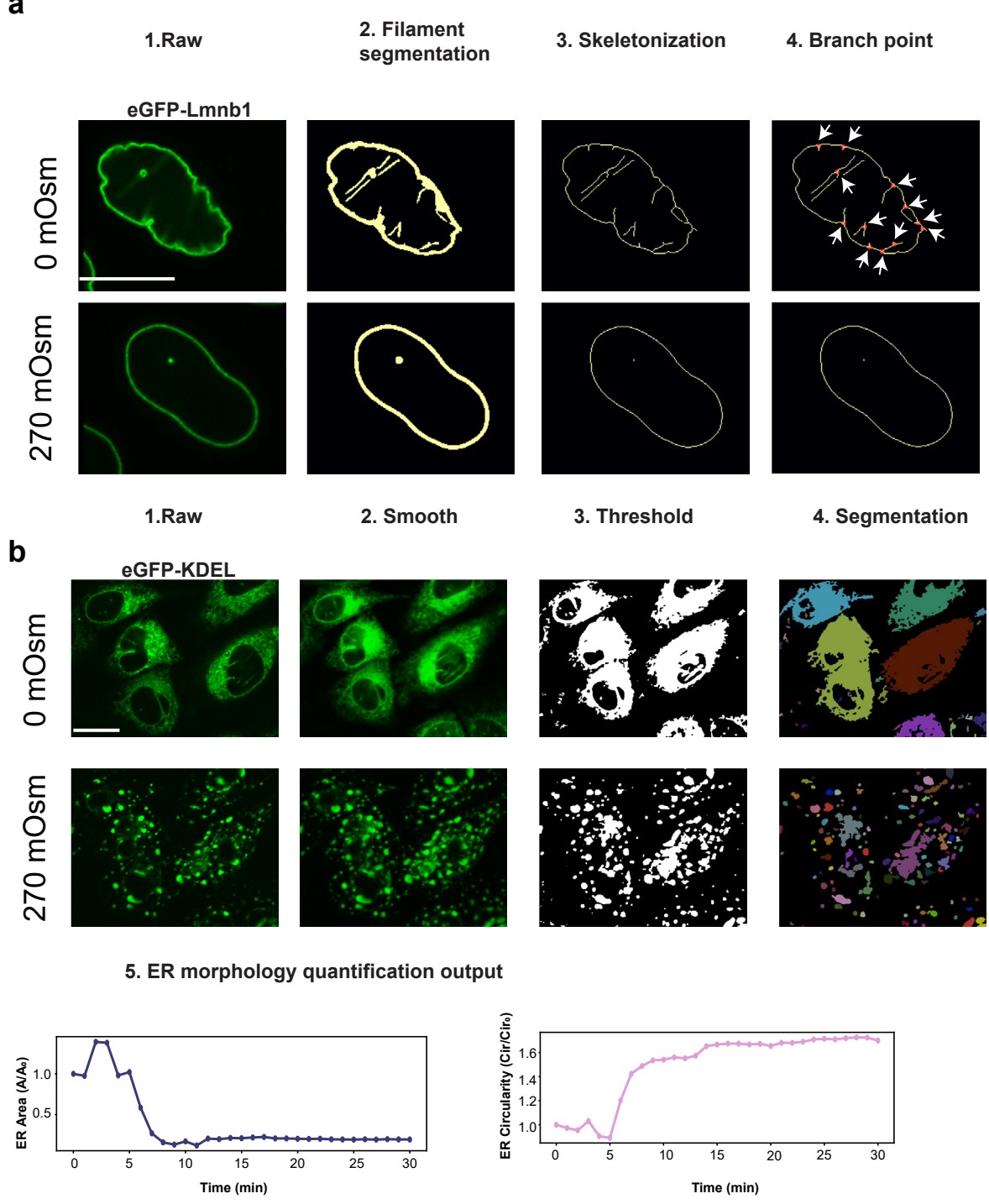

**Extended Data Fig. 7 | Scheme of analysis for ER and Lamin morphology.**
**(a)** Overview of the Lamin branchpoint analysis to quantitatively measure nuclear envelope invaginations across hypoosmotic shock and pre-lytic phase of ferroptotic cell death. 270 mOsm hypoosmotic stress on eGFP-Lmnb1 and ALPIN2 labelled U2OS cells was used here as a demonstration example. Specifically, gaussian smooth followed by a custom-built filament-based filter algorithm (Step 2) was applied to the previously selected nuclear middle slice (Extended Data Fig. 6b). The resulting segmentation mask was skeletonized to a 1-pixel thick outline (Step 3). Branch points (pixels with >3 neighbors) were calculated to estimate nuclear envelope invagination (Step 4, branchpoints are indicated by white arrows and red circles). Scale bar, 20 μm. **(b)** Overview of the ER morphology analysis to quantify the change of ER area and circularity across osmotic and ferroptotic stress. 270 mOsm hypoosmotic stress on

eGFP-KDEL and cPla₂-mKate2 labelled U2OS cells was presented here as a demonstration example. ER image stack (488 nm channel) first underwent intensity normalization and 3D-gaussian blur (Step 2), followed by thresholded segmentation (Step 3 and 4) via a marker-controlled watershed algorithm (If the ER is tagged with SEC61B, a custom-built filament thresholding algorithm was used in place of marker-controlled watershed algorithm). The slice with the largest total ER area was taken as the middle plane. Since ER network was morphologically complex and overlapped across neighboring cells, it was technologically infeasible to quantify ER morphology for individual cell. Instead(step5), the maximum ER object area and the mean circularity were computed to summarize the ER morphology across the entire FOV and exported for downstream analysis. A, area. Cir, circularity. Scale bar, 20 μm.

# Reporting Summary

## Statistics

For all statistical analyses, confirm that the following items are present in the figure legend, table legend, main text, or Methods section.

| n/a | Confirmed | |
|---|---|---|
| ☐ | ☒ | The exact sample size (*n*) for each experimental group/condition, given as a discrete number and unit of measurement |
| ☐ | ☒ | A statement on whether measurements were taken from distinct samples or whether the same sample was measured repeatedly |
| ☐ | ☒ | The statistical test(s) used AND whether they are one- or two-sided *Only common tests should be described solely by name; describe more complex techniques in the Methods section.* |
| ☒ | ☐ | A description of all covariates tested |
| ☐ | ☒ | A description of any assumptions or corrections, such as tests of normality and adjustment for multiple comparisons |
| ☐ | ☒ | A full description of the statistical parameters including central tendency (e.g. means) or other basic estimates (e.g. regression coefficient) AND variation (e.g. standard deviation) or associated estimates of uncertainty (e.g. confidence intervals) |
| ☐ | ☒ | For null hypothesis testing, the test statistic (e.g. *F*, *t*, *r*) with confidence intervals, effect sizes, degrees of freedom and *P* value noted *Give P values as exact values whenever suitable.* |
| ☒ | ☐ | For Bayesian analysis, information on the choice of priors and Markov chain Monte Carlo settings |
| ☒ | ☐ | For hierarchical and complex designs, identification of the appropriate level for tests and full reporting of outcomes |
| ☐ | ☒ | Estimates of effect sizes (e.g. Cohen's *d*, Pearson's *r*), indicating how they were calculated |

*Our web collection on statistics for biologists contains articles on many of the points above.*

## Software and code

Policy information about availability of computer code

| Data collection | U2OS cell and GUV membrane confocal imaging data were collected in NIS-Elements Software (4.13.04) and NIS-Elements Software (5.21.03). Intravital confocal microscopy of zebrafish larvae were collected in NIS-Elements Software (3.22.14). Multi-photon laser imaging data of zebrafish larvae were collected in Leica LAS X 4.8.1.29271. Super Resolution imaging data of U2OS cell were collected in ZEN Black 3.0 SR FP2, v16.0.20.306. FLIP Imaging data were acquired in Leica LAS X 3.5.7.23225. |
|---|---|
| Data analysis | FLIP imaging data were processed in LAS X 3.5.7.23225. segmented in FIJI (ImageJ) and analyzed in custom Python3 scripts (https://github.com/zazadovv/ER_Nucleus.git). Structured-illumination microscopy (SIM) reconstruction was performed in ZEN 3.0 SR FP2 (16.0.20.306 black). Segmentation of SIM masks were performed in Imaris v10.2 before visualizing in custom written python3 scripts (https://github.com/zazadovv/ER_Nucleus.git). Zebrafish image stacks were deconvolved and denoised using default method and denoise.ai tools in Nikon NIS elements (5.21.03). Multiphoton imaging results were processed using Leica LAS X, 4.8.1.29271. All live, permeabilized cell and GUV membrane imaging experiments were prepossessed, segmented and analyzed in custom Python3 scripts (Nuclear membrane analysis: https://github.com/joeshen123/Nuclear-Membrane-Binding-4 Analysis.git, ER membrane analysis: https://github.com/joeshen123/ER-Structure-Analysis.git , GUV membrane binding analysis: https://github.com/joeshen123/GUV-Protein-Binding-Analysis-Program.git ).GUV equilibrium binding isotherm was calculated and plotted using Curve Fitting Box (V 25.1) from Matlab R2025a before imported and modified in Adobe AI. Representative images were prepared in FIJI (ImageJ) and plots were generated with seaborn or matplotlib from Python 3 as vector graphs and subsequently modified in Adobe AI. |

For manuscripts utilizing custom algorithms or software that are central to the research but not yet described in published literature, software must be made available to editors and reviewers. We strongly encourage code deposition in a community repository (e.g. GitHub). See the Nature Portfolio guidelines for submitting code & software for further information.

## Data

Policy information about availability of data

All manuscripts must include a data availability statement. This statement should provide the following information, where applicable:

- Accession codes, unique identifiers, or web links for publicly available datasets
- A description of any restrictions on data availability
- For clinical datasets or third party data, please ensure that the statement adheres to our policy

> Numerical source data supporting the findings of this study are provided with the publication. Additional data are available from the corresponding author(s) upon reasonable request.

## Research involving human participants, their data, or biological material

Policy information about studies with human participants or human data. See also policy information about sex, gender (identity/presentation), and sexual orientation and race, ethnicity and racism.

| | |
|---|---|
| Reporting on sex and gender | N/A |
| Reporting on race, ethnicity, or other socially relevant groupings | N/A |
| Population characteristics | N/A |
| Recruitment | N/A |
| Ethics oversight | N/A |

Note that full information on the approval of the study protocol must also be provided in the manuscript.

# Field-specific reporting

Please select the one below that is the best fit for your research. If you are not sure, read the appropriate sections before making your selection.

☒ Life sciences ☐ Behavioural & social sciences ☐ Ecological, evolutionary & environmental sciences

For a reference copy of the document with all sections, see nature.com/documents/nr-reporting-summary-flat.pdf

# Life sciences study design

All studies must disclose on these points even when the disclosure is negative.

| | |
|---|---|
| Sample size | Sample sizes were guided by prior experience and published literature. No statistical methods were used to pre-determine sample size or effect size. Time-lapse U2OS cell experiments were performed with 2–5 biological replicates, with key experiments repeated in at least three independent replicates to ensure reproducibility. Single time-point U2OS cell assays (Fig. 1a (right panel), 2c, Extended Data Fig.1a, Extended Data fig. 2g, and Extended Data Fig. 2h) were conducted in one independent experiment performed on a single day, using cells from two independent frozen vials. Equilibrium GUV experiments (extended data Fig. 2a) were performed as a single independent experiment on a single day, using GUVs from two independently prepared lipid mixtures. For zebrafish imaging, each experiment represents a biologically independent replicate using embryos derived from distinct clutches. Experiments were repeated across multiple days, with one embryo imaged per experiment for each UV- or IR-induced tissue wounding condition, and repeated on more than six different live embryos to ensure reproducibility. |
| Data exclusions | Data exclusion criteria were predefined. Imaging datasets were excluded if affected by artificial fluorescence fluctuations (e.g., immersion oil bubbles), incompletely acquired Z-stacks or broken tracks were either cropped or completely discarded. All raw imaging files of cell experiments were loaded into custom written automatic programs for analysis, unless otherwise stated. For the nuclear membrane analysis program, segmented objects with size less than 8000 pixel2 (~140 um^2) and/or touching the borders of the image boundary were excluded from the analysis.   Nuclei were removed from analysis If when segmented objects moved more than 130 pixel (~20 μm) between successive frames or broken tracks appeared due to leakage of nucleoplasmic fluorescent markers (cPla2-mKate2, ALPIN) when these markers were used for nuclear segmentation, these nuclei were removed from analysis. Multiple tracks that were detected as single were deleted from the analysis. For the ER analysis program, segmented ER vesicles size less than 30-50 pixel2 (~ 1 μm2) were precluded from analysis. Since the ER was analyzed per FOV, no tracks were involved, and all time points were included in the analysis unless ER segmentation was precluded by the leakage of eGFP-KDEL. In long-term cell imaging experiments, datasets were excluded only when imaging artifacts prevented reliable analysis and data extraction. Specifically, exclusions were made if (i) after fluorescent markers leaked from their designated organelle localization into the surrounding cytoplasmic space during acquisition, (ii) cells became clumped, or (iii) automated tracking produced broken or fragmented tracks. For GUV experiments, the raw imaging files were loaded into a semi-automatic custom written analysis program where individual GUV was randomly selected by the user for analysis. For zebrafish live embryo imaging, datasets were excluded if tailfin tissue drift exceeded 100 |

| | |
|---|---|
| | pixels (~50 μm) in the XY plane or 20 pixels (~10 μm) in the Z-direction following UV- or IR- laser damage. All other biologically independent replicates were retained for statistical analyses, and no additional outliers were excluded. |
| Replication | Typically, experiments were repeated at least on two separate days (using cells thawed from different vials). Only the experiments shown in Fig. 1a (right panel), 2c, Extended Data fig.1a, Extended Data fig.2a, Extended Data Fig. 2g, and Extended Data fig. 2h were not replicated on different days, yet nevertheless conducted using cells or GUVs prepared from at least two independent stock sources, i.e., storage vials or lipid mixtures, respectively. For zebrafish embryos imaging experiments were repeated on two separate days using independent clutch of embryos. Statistical analysis were performed across biological replicates. Replication included either variation in biological source material and/or independent imaging sessions conducted on separate experimental days. |
| Randomization | Selection of animal larvae and cells for experiments is randomized without predetermined grouping, bias or selection. |
| Blinding | Analysis was performed objectively without blinding. Data processing and quantification were carried out using an automated computational pipeline, with no manual intervention or subjective data selection prior to analysis. |

# Reporting for specific materials, systems and methods

We require information from authors about some types of materials, experimental systems and methods used in many studies. Here, indicate whether each material, system or method listed is relevant to your study. If you are not sure if a list item applies to your research, read the appropriate section before selecting a response.

### Materials & experimental systems

| n/a | Involved in the study |
|---|---|
| ☒ | ☐ Antibodies |
| ☐ | ☒ Eukaryotic cell lines |
| ☒ | ☐ Palaeontology and archaeology |
| ☐ | ☒ Animals and other organisms |
| ☒ | ☐ Clinical data |
| ☒ | ☐ Dual use research of concern |
| ☒ | ☐ Plants |

### Methods

| n/a | Involved in the study |
|---|---|
| ☒ | ☐ ChIP-seq |
| ☒ | ☐ Flow cytometry |
| ☒ | ☐ MRI-based neuroimaging |

## Eukaryotic cell lines

Policy information about cell lines and Sex and Gender in Research

| | |
|---|---|
| Cell line source(s) | ATCC |
| Authentication | Since cell lines were obtained from commercial sources, no authentications were performed. |
| Mycoplasma contamination | Cell lines were not tested for Mycoplasma contamination |
| Commonly misidentified lines (See ICLAC register) | No commonly misidentified lines were used in the study |

## Animals and other research organisms

Policy information about studies involving animals; ARRIVE guidelines recommended for reporting animal research, and Sex and Gender in Research

| | |
|---|---|
| Laboratory animals | Adult casper zebrafish were reared in 2.8-liter polycarbonate tanks at a density of 10 fish per liter. Fish were maintained in salinity-conditioned system water at 28 °C under a 14:10-hour light:dark photoperiod cycle. |
| Wild animals | Study did not involve any wild animals. |
| Reporting on sex | Sex are indeterminate in the larval stage at which the presented experiments are conducted. |
| Field-collected samples | The study did not involve samples that were directly collected from the field. |
| Ethics oversight | Experiments are conducted according to institutional animal healthcare guidelines with the approval of the Institutional Animal Care and Use Committee (IACUC). MSKCC Protocol Number:11-01-002 |

Note that full information on the approval of the study protocol must also be provided in the manuscript.

## Plants

Seed stocks

N/A

Novel plant genotypes

N/A

Authentication

N/A

