## [Peer Review File · Nature Cell Biology]

Endoplasmic reticulum disruption stimulates nuclear membrane mechanotransduction

Corresponding Author: Dr Philipp Niethammer

Version 0:

Decision Letter:

Dear Dr Niethammer,

Thank you again for submitting your manuscript "Buffering of nuclear membrane tension and mechanotransduction by the endoplasmic reticulum revealed by quantitative ALPIN imaging", to Nature Cell Biology. We are very sorry for the delay in sharing our decision with you. At this stage, we have not received comments from Rev#3 (wound healing/zebrafish expert), despite repeated emails to the reviewer. However, we feel that we have received enough expert feedback to make our decision, in the interest of time. We will forward comments from Rev#3 to you if/when they are received. We apologize again for the delay.

Your manuscript has now been seen by 2 referees, who are experts in lipid biology, lipid probes (Referee #1) and nuclear mechanotransduction (Referee #2), whose comments are pasted below. In light of their advice, we regret that we cannot offer to publish the study in Nature Cell Biology.

As you will see, although the reviewers found the work interesting, they raised serious concerns that question the strength of the data and of the novel conclusions that can be drawn at this stage. The reviewers provided consistent feedback indicating that the work would require substantial additional evidence and controls. We have discussed the reviewers' feedback and regrettably find that the dataset is too preliminary for further consideration at the journal. We're very sorry to be sharing this news.

Although we cannot publish your paper, it may be appropriate for another journal in the Nature Portfolio. If you wish to explore the journals and transfer your manuscript please use our manuscript transfer portal. You will not have to re-supply manuscript metadata and files, unless you wish to make modifications. For more information, please see our [manuscript transfer FAQ](http://www.nature.com/authors/author_resources/transfer_manuscripts.html?WT.mc_id=EMI_NPG_1511_AUTHORTRANSF&WT.ec_id=AUTHOR) page.

We are very sorry that we could not be more positive on this occasion, but we thank you for the opportunity to consider this work.

With kind regards,
Melina

Melina Casadio, PhD
Senior Editor, Nature Cell Biology
Consulting Editor, Nature Structural & Molecular Biology
ORCID ID: <https://orcid.org/0000-0003-2389-2243>

Reviewers' comments:

Reviewer #1 (Remarks to the Author):

This paper focuses on the activation pathway of cPLa2, a nucleoplasmic phospholipase A2 with mechanotransduction properties. Notably its sensitivity to membrane stretch associated with changes in nuclear shape is known. Since cPLa2 is also Ca²⁺-sensitive, it is difficult to measure the real impact of inner nuclear membrane (INM) tension on cPLa2 membrane adsorption. Furthermore, as the NM is contiguous with the endoplasmic reticulum (ER), the latter may play a protective role in rapidly dissipating NM tension. The study uses several approaches ranging from in vitro reconstitution to in vivo demonstration on zebrafish to address these challenges. One original point was to develop an intra-nuclear probe called

ALPIN, which is sensitive to membrane tension but Ca²⁺ insensitive.

The authors elegantly demonstrated the parallel recruitment of ALPIN and cPlA2 to the INM during osmotic shock, cell compression, in permeabilized or ferroptotic cells using robust imaging quantification. In particular, this recruitment is enhanced when they interfere with the network morphology of the ER, suggesting a buffering role of this organelle in INM tension. In agreement with the cellular data, impacting the ER in zebrafish appears to enhance cPlA2 recruitment to the INM. The study is well conducted, and the evidence is sound overall. The text is clear and fairly well detailed. However, critical controls are needed and some data are missing to validate all the authors' hypotheses. In addition, in many figure panels, only one representative curve is shown, whereas it would be appreciated to have more measurements.

Major points:

1- The demonstration of the paper is based on several assays designed to fragment the ER. However, while the images show deformation or vesiculation of the ER, which is well quantified, they do not demonstrate discontinuity between nuclear and ER membranes. A weak but consistent eGFP-KDEL signal can be seen between "vesicles" in some images. If a membrane link exists, it is not clear how it would oppose lipid diffusion between ER and NM. The absence of ER-NM contiguity needs to be shown at higher resolution, e.g. by electron microscopy or super-resolution 3D imaging. Real-time membrane diffusion measurements of ER lipids or proteins (via FRAP, or FLIP etc.) with appropriate controls is also highly recommended to demonstrate altered ER-NM contiguity.

2- The phospholipase activity of cPlA2 is never measured, thus the real impact of ER-NM membrane continuity combined with the various treatments that increase membrane tension is not studied in terms of mechanotransduction. The data presented should be coupled with measurements of fatty acid release.

3- The role of membrane tension independently of that of Ca²⁺ in the recruitment of cPlA2 to the INM has not been demonstrated in vivo. In this respect, experiments in zebrafish using ALPIN expression should be carried out.

Minor points:

4- Fig S3. It is impossible to distinguish Sytox Orange from cPlA2-mKate2 from the images. The authors need to redo these controls, knowing that other lysis markers of different colors exist, even in the Sytox family. Quantification of results is also recommended.

5- Fig S1b, c, Fig S2c, d, Fig S5a, S5b, etc. A single representative curve is shown, whereas the figure legend mentions many more measurements. A mean curve with errors bars, or better still, all the other curves (in shaded colors, for example), should be shown. A critical example is the time lag between the recruitment of cPlA2 to the INM and the ER circularity increase in Fig S2d: this is a point discussed in the text and cannot be based on a single curve.

6- Fig S4i. GUV measurements must be displayed in detail by showing GUV-binding isotherms and fitted lines.

7- Fig 2. ML162 experiments. More time series should be shown in fig 2b, c as only one measurement is displayed. The authors wrote that T(INM) using ALPIN and eGFP-KDEL were concomitantly monitored, but Fig2 suggests that measurements were performed on different cells. No time series is shown for cPlA2 or ALPIN recruitment to the INM.

8- Fig 3. The authors have measured rate of ALPIN sensor binding to the INM, but experimental method is missing and the temporal resolution of their timelapse microscopy measurements is not specified. This is important to get an idea of measurement accuracy. Are units missing from the rate values shown in the tables of fig 3?

9- Finally, I believe that some of the cartoons/panels in the supplementary figures could be switched to the main figures. I felt I spent more time looking at the sup figures than the main figures, so a rebalancing might be advisable.

Reviewer #2 (Remarks to the Author):

This is a very interesting but brief submission that proposes a newly designed "Ca²⁺ independent, intranuclear lipid packing probe (called) ALPIN". Decades of past work showing phospholipase binding to dilated membranes has been nicely extended in previous work of the Niethammer lab to the inner nuclear membrane. However, a persistent question has been the role of the ER that is contiguous, and so they also propose here to address this with their new probe "in the presence or absence of a contiguous ER" (sketched in Fig.4b). Although I am unconvinced that this latter issue is adequately addressed, some aspects of the work seem foundational to an advance for the broader topic. In particular, the following concerns temper enthusiasm.

1. Overall rigor for reproducibility requires attention throughout. For a start, Fig.1 needs more calibration between isotonic conditions and 90% dilution of isotonic (such as 30%, 60%, and then 90%). Based on Supplement Figs for confinement: What were the nucleus heights before and after the compression, and did all cells survive?

2. Rationale: the authors need to provide a summary in the Intro of patho-physiological conditions where osmotic stresses change so drastically for human cells - given the studies here of human U2OS osteosarcoma-derived cells. This is because most if not all of the studies here are done in extreme limits relevant perhaps to wounding of some fish cells in fresh water

(which is low Osm at 5-50 mOsm). Some comments in the text about pH buffering are also needed.

3. Rigor of kinetics: The authors use Sytox Orange for plasma membrane rupture, but it should be compared to how fast can rupture be detected through loss of some freely diffusing cytoplasmic probe such as GFP and also cytoplasmic entry of a low MW dye? Sytox Orange could be slower as it must enter the cell, then the nucleus, and bind DNA.

4. Rigor in claim "Ca²⁺ independent": In particular, do the cells have any residual calcium in Fig.3b (e.g. by elemental analyses)? The high negative charge of DNA and the well-known presence of Ca²⁺ in the nucleus add complexity. If there is any residual Ca in the key studies of Fig.3b, then the differences could reflect this, so that the ALPIN sensor remains sensitive to Ca inside cells (even if not sensitive in vitro with vesicle studies).

5. Rigor in functional assessment: Is the ALPIN sensor accumulating in nucleoli? If not, then what are the nuclear depots or accumulations, and are those dependent or even competing with the effects of NE membrane dilation or calcium? Furthermore, for Fig.1e: Does ALPIN compete for NE binding with cPLA2? Does more ALPIN binding occur with partial knockdown of cPLA2? If competition is not detected, then do they bind different membrane sites?

6. Relevance to death or to nuclear mechanosensing of viable cells is unclear. In particular, Fig.2,S3b seems designed to chemically drive death of U2OS cells (by ferroptosis with an inhibitor of glutathione peroxidase), but the kinetics of ER changes and cell death require greater clarity. Sytox orange staining needs to be shown in a different color in the images, but more important is the need to conduct similar experiments with other markers of death. The broader question raised is whether binding of ALPIN (and maybe even cPLA2) to the nuclear envelope can occur in cells that are fully viable (can they divide). In other words, is ALPIN a death sensor for the nucleus loosely analogous to Annexin for the plasma membrane?

7. Novelty of findings with ER, and related to the previous point: Fig.1a is reminiscent of ER stress, which can eventually lead to death

[PMID: 23850759, <https://doi.org/10.3389/fcell.2021.767866>].

Based on reports of massive cell death upon hypotonic exposure such as in [<https://doi.org/10.3389/fphys.2020.582781>], the authors need to provide longer term assessments of death.

Also, ER stress drugs such as Thapsigargin (Tg) should also be tested.

Lastly in this context, why does ER fragment under compression? One can certainly squash cells to death, and so long-term viability again needs careful quantitation.

8. Rigor in analyses of Fig.3B: are images collected every minute or more often? Does that frequency affect the rates in the Tables? What are the units for rates, and do the rates show a significance dependence on indicated Ca levels? Does this support or not the original claim of "we developed the Ca²⁺ independent, intranuclear lipid packing probe ALPIN"?

9. Relevance to mechanosensing or death is again raised in Fig.4. This is because wounding does cause some cell death [doi.org/10.7554/eLife.86269],

but that might well require longer times than studied.

The authors need to broadly and rigorously address this fate in this model and across their studies, especially given the resemblance of ER changes here to cited ER stress.

10. Rigor and interpretability for the two phases of response such as in Fig.1b & S4d-g: More images of later timepoints are needed for all. What are the decay times, and what sets these times?

11. Rigor and reproducibility in Fig.S4i: show the binding isotherms and show them for different sized vesicles to address the proposed curvature dependence of Fig.S4b.

12. Lastly, returning to "in the presence or absence of a contiguous ER" (sketched in Fig.4b), the authors need to summarize in this final figure (or its legend) exactly which Figure panels support the left cartoon and comment that the rest of the results support the right cartoon (i.e. absence of ER connections).

Minor:

1. References 1-5 are from two groups including the Niethammer lab, and they do not adequately reference past work on the listed topics of differentiation, ECM, etc.

2. Much earlier work than Ref.7 is more appropriate to cite, such as: "Regulation of phospholipase A2 activity by the lipid-water interface" Biochemistry. 1979. The authors need to provide a better summary with historical references of phospholipase activity tension/density regulation at membrane interfaces.

3. "Nuclear shape sensing" should probably be replaced with nuclear area sensing (or NE area sensing). For example, a sphere and ellipsoid differ in shape, but small and large spheres do not differ in shape.

Reviewer #2 (Remarks on code availability):

Not sure it's relevant.

**For Nature Portfolio general information and news for authors, see <http://npg.nature.com/authors>.

Version 1:

Decision Letter:

Dear Dr Niethammer,

Thank you for your email asking us to reconsider our decision on your manuscript, "Endoplasmic reticulum disruption stimulates nuclear membrane mechanotransduction". We are always willing to hear the authors' perspective, but we must first prioritize decisions on new submissions. We appreciate your patience while we considered this appeal.

I have now discussed your manuscript, the referees' comments and your rebuttal, in detail with my colleagues, and we would be willing to reconsider a revised manuscript provided the following issues can be addressed, and that nothing similar is accepted for publication at Nature Cell Biology or published elsewhere in the meantime.

In particular, please resubmit the manuscript with all three full reviewer reports verbatim, excluding none of their comments, as we noted some paragraphs were missing.

In addition, please pay close attention to our guidelines on statistical and methodological reporting (listed below) as failure to do so may delay the reconsideration of the revised manuscript. In particular please provide:

On resubmission please provide the completed Reporting Summary (found here <https://www.nature.com/documents/nr-reporting-summary.pdf>). This is essential for reconsideration of the manuscript and this document will be available to editors and referees in the event of peer review. For more information see below. Please also ensure that the presentation of statistical information in the revised submission complies with Nature Cell Biology's statistical guidelines (see below).

As part of a pilot aimed at increasing reporting completeness for light microscopy experiments, we ask our authors to fill out a short reporting table [https://www.nature.com/documents/Light_microscopy_reporting_table.xlsx] that is made available to reviewers and ultimately published in the event of acceptance of the manuscript. If your manuscript includes light microscopy data, please return the table via email with the Reporting Summary and Editorial Policy Checklist.

Please use the link below to submit the complete manuscript files, and include a point-by-point response to the complete reviewer comments, verbatim as provided in their reports.

Link Redacted

Please let us know how you wish to proceed and when we can expect your revised manuscript.

With kind regards,

Angela Parrish

Angela R Parrish, PhD
Locum Senior Editor
Nature Cell Biology

GUIDELINES FOR EXPERIMENTAL AND STATISTICAL REPORTING

REPORTING REQUIREMENTS – To improve the quality of methods and statistics reporting in our papers we have recently revised the reporting checklist we introduced in 2013. We are now asking all life sciences authors to complete a reporting summary (found here <https://www.nature.com/documents/nr-reporting-summary.pdf>) that collects information on experimental design and reagents. This document is available to referees to aid the evaluation of the manuscript. Please note that this form is a dynamic 'smart pdf' and must therefore be downloaded and completed in Adobe Reader. We will then flatten it for ease of use by the reviewers. If you would like to reference the guidance text as you complete the template, please access these flattened versions at <http://www.nature.com/authors/policies/availability.html>.

STATISTICS – Wherever statistics have been derived the legend needs to provide the n number (i.e. the sample size used to derive statistics) as a precise value (not a range), and define what this value represents. Error bars need to be defined in the legends (e.g. SD, SEM) together with a measure of centre (e.g. mean, median), Box plots need to be defined in terms of minima, maxima, centre, and percentiles. Ranges are more appropriate than standard errors for small data sets. Wherever statistical significance has been derived, precise p values need to be provided and the statistical test used needs to be stated in the legend. Statistics such as error bars must not be derived from $n < 3$. For sample sizes of $n < 5$ please plot the individual data points rather than providing bar graphs. Deriving statistics from technical replicate samples, rather than biological replicates is strongly discouraged. Wherever statistical significance has been derived, precise p values need to be provided and the statistical test stated in the legend.

Version 2:

Decision Letter:

Our ref: NCB-LE56323B

3rd October 2025

Dear Dr. Niethammer,

Thank you for submitting your revised manuscript "Endoplasmic reticulum disruption stimulates nuclear membrane mechanotransduction" (NCB-LE56323B). It has now been seen by the original referees and one new referee and their comments are below. The reviewers find that the paper has improved in revision, and therefore we'll be happy in principle to publish it in Nature Cell Biology, pending minor revisions to satisfy the referees' final requests and to comply with our editorial and formatting guidelines.

Thank you again for your interest in Nature Cell Biology Please do not hesitate to contact me if you have any questions.

Sincerely,

Angela R Parrish, PhD
Locum Senior Editor
Nature Cell Biology

Reviewer #1 (Remarks to the Author):

The authors carried out the important controls requested and integrated additional measurements for statistics when needed. They confirmed the vesiculation of the ER and the lack of continuity between NM and ER in their experimental setup, in particular through super-resolution imaging and FLIP experiments. Overall, the revised article reads well and is fairly clear.

Minor point: I am a little surprised by the choice of some images to illustrate timelapse imaging experiments. For example, cPlA2 INM-adsorption during shock and during the pre-lytic phase is not convincing in Fig. S2b (e.g. at 12' and 252' compared to 0'). This does not visually reflect the quantification (red curve) shown in S2c, nor even the quantification by linescan in FigS2b middle panel. Is it possible to find images series that better illustrate the adsorption of cPlA2 to the INM?

Reviewer #3 (Remarks to the Author):

The authors have addressed all issues I had raised in a satisfactory manner. The manuscript was already very strong in the initial version and is now further improved. I strongly recommend publication in NCB.

Reviewer #4 (Remarks to the Author):

In this revised version of the manuscript, using a novel biosensor ALPIN inside the nucleus, together with high-resolution imaging, the authors presented results showing that the ER acts as a nuclear membrane buffer in response to a hypo-osmotic stress or Ca²⁺-gated signaling. They revealed that osmotic stress-induced ER vesiculation led to disruption of the ER reservoir that enhances NMMT. In response to the comments from the three reviewers, the authors performed necessary additional experiments to strengthen the conclusion of the study. Some issues raised by the reviewers were also clarified and explained by the authors. Although the underlying molecular mechanism(s) of ER vesiculation remains elusive, as acknowledged by the authors, this study contributes to the understanding of the role of ER in nuclear membrane mechanosensing, possibly as a danger-signaling trigger.

Version 3:

Decision Letter:

Dear Dr Niethammer,

I am writing on behalf of my colleague Dr Angela Parrish, who is out of the office.

I am pleased to inform you that your manuscript, "Endoplasmic reticulum disruption stimulates nuclear membrane mechanotransduction", has now been accepted for publication in *Nature Cell Biology*.

Over the next few weeks, your paper will be copyedited to ensure that it conforms to *Nature Cell Biology* style. Once your paper is typeset, you will receive an email with a link to choose the appropriate publishing options for your paper and our Author Services team will be in touch regarding any additional information that may be required.

Publication is conditional on the manuscript not being published elsewhere and on there being no announcement of this work to any media outlet until the online publication date in *Nature Cell Biology*.

Please note that *Nature Cell Biology* is a Transformative Journal (TJ). Authors may publish their research with us through the traditional subscription access route or make their paper immediately open access through payment of an article-processing charge (APC). Authors will not be required to make a final decision about access to their article until it has been accepted. <https://www.springernature.com/gp/open-research/transformative-journals> Find out more about Transformative Journals

Authors may need to take specific actions to achieve compliance with funder and institutional open access mandates. If your research is supported by a funder that requires immediate open access (e.g. according to [Plan S principles](https://www.springernature.com/gp/open-science/plan-s-compliance) or the [NIH public access policy](https://www.springernature.com/gp/open-science/us-federal-agency-compliance)) then you should select the gold OA route, and we will direct you to the compliant route where possible. Because authors warrant under our subscription licensing terms that they haven't committed to licensing any version of their article under a licence inconsistent with the terms of our agreement – including the applicable embargo period – publication under the subscription model isn't suitable for authors whose funders require no embargo.

If you have not already done so, we strongly recommend that you upload the step-by-step protocols used in this manuscript to protocols.io (<https://protocols.io>), an open online resource that allows researchers to share their detailed experimental know-how. All uploaded protocols are made freely available and are assigned DOIs for ease of citation. Protocols and Nature Portfolio journal papers in which they are used can be linked to one another, and this link is clearly and prominently visible in the online versions of both. Authors who performed the specific experiments can act as primary authors for the Protocol as they will be best placed to share the methodology details, but the Corresponding Author of the present research paper should be included as one of the authors. By uploading your Protocols onto protocols.io, you are enabling researchers to more readily reproduce or adapt the methodology you use, as well as increasing the visibility of your protocols and papers. You can also establish a dedicated workspace to collect your lab Protocols. Further information can be found at <https://www.protocols.io/help/publish-articles>.

Nature Cell Biology encourages authors presenting evidence for cell, biological, molecular, and genetic interactions to consider communicating these findings using Biofactoid (<https://biofactoid.org/>). This tool helps users share a searchable representation of interactions (e.g. binding, gene expression, post-translational modification) between genes, gene products, or chemicals. Information added to Biofactoid, with author attribution, is shared on social media and public databases, such as Pathway Commons, where it can be discovered and analyzed in the context of a large and growing corpus of knowledge.

With kind regards,

Melina Casadio, PhD
Senior Editor, Nature Cell Biology
Consulting Editor, Nature Structural & Molecular Biology
ORCID ID: <https://orcid.org/0000-0003-2389-2243>

** Visit the Springer Nature Editorial and Publishing website at http://editorial-jobs.springernature.com?utm_source=ejp_NCB_email&utm_medium=ejp_NCB_email&utm_campaign=ejp_NCB for more information about our career opportunities. If you have any questions please click [here](mailto:editorial.publishing.jobs@springernature.com).

We would like to thank the reviewers for their interest and thorough assessment of our manuscript. Based on their critique we entirely revised the manuscript adding a substantial number of experiments, consistent quantification, and a more focused narrative that avoids aspects that, we suspect, previously caused confusion.

Major additions include:

1. Super-resolution and FLIP assays that independently confirm complete functional separation of the nuclear membrane and ER.
2. Quantitative long-term imaging of ER vesiculation, nuclear morphology/volume, ALPIN/cPLA₂-INM adsorption, and cell death after osmotic shock and ferroptosis, placing ER buffering in its full temporal context.
3. Comprehensive statistical integration of all quantitative data.
4. Streamlined text and figures for greater clarity.
5. Independent replication of key experiments by Zaza Gelashvili (now co-first author).
6. An experimental registration sheet linking every figure to the raw and numerical data.

Altogether, we believe that our extensive revisions significantly improved the overall strength, focus, and accessibility of our findings.

Please find below a point-by-point discussion of all the comments.

Kind regards,

P. Niethammer (for the authors).

Reviewer #1 (Remarks to the Author):

This paper focuses on the activation pathway of cPla2, a nucleoplasmic phospholipase A2 with mechanotransduction properties. Notably its sensitivity to membrane stretch associated with changes in nuclear shape is known. Since cPla2 is also Ca²⁺-sensitive, it is difficult to measure the real impact of inner nuclear membrane (INM) tension on cPla2 membrane adsorption. Furthermore, as the NM is contiguous with the endoplasmic reticulum (ER), the latter may play a protective role in rapidly dissipating NM tension. The study uses several approaches ranging from in vitro

reconstitution to in vivo demonstration on zebrafish to address these challenges. One original point was to develop an intra-nuclear probe called ALPIN, which is sensitive to membrane tension but Ca^{2+} insensitive.

The authors elegantly demonstrated the parallel recruitment of ALPIN and cPlA2 to the INM during osmotic shock, cell compression, in permeabilized or ferroptotic cells using robust imaging quantification. In particular, this recruitment is enhanced when they interfere with the network morphology of the ER, suggesting a buffering role of this organelle in INM tension. In agreement with the cellular data, impacting the ER in zebrafish appears to enhance cPlA2 recruitment to the INM.

The study is well conducted, and the evidence is sound overall. The text is clear and fairly well detailed. However, critical controls are needed and some data are missing to validate all the authors' hypotheses. In addition, in many figure panels, only one representative curve is shown, whereas it would be appreciated to have more measurements.

Major points:

1- The demonstration of the paper is based on several assays designed to fragment the ER. However, while the images show deformation or vesiculation of the ER, which is well quantified, they do not demonstrate discontinuity between nuclear and ER membranes. A weak but consistent eGFP-KDEL signal can be seen between “vesicles” in some images. If a membrane link exists, it is not clear how it would oppose lipid diffusion between ER and NM. The absence of ER-NM contiguity needs to be shown at higher resolution, e.g. by electron microscopy or super-resolution 3D imaging. Real-time membrane diffusion measurements of ER lipids or proteins (via FRAP, or FLIP etc.) with appropriate controls is also highly recommended to demonstrate altered ER-NM contiguity.

> We thank the reviewer for pointing out this central, story-relevant concern. As suggested, we conducted additional super-resolution imaging and FLIP experiments to confirm that no functional connections exist between the vesiculated ER and the NM that would allow significant biomolecule exchange. Our new data (Fig. 1c-d, S1e) are unambiguous and in line with published results from the Lippincott-Schwartz group that previously demonstrated lack of functional connections between osmotically generated ER vesicles (1).

2- The phospholipase activity of cPlA2 is never measured, thus the real

impact of ER-NM membrane continuity combined with the various treatments that increase membrane tension is not studied in terms of mechanotransduction. The data presented should be coupled with measurements of fatty acid release.

> Thank you for emphasizing the importance of directly linking membrane mechanics to cPLA₂ catalytic output. In principle, we would compare arachidonic-acid (AA) release after osmotic shock in cells that do vs. do not undergo ER vesiculation. Unfortunately, it is currently not feasible to conduct the requested experiment in a meaningful way:

We need Ca²⁺ to split the ER from the nucleus, but Ca²⁺ also activates cPLA₂ in conjunction with T_{INM}. Thus, when using AA as readout for cPLA₂ activity, the direct effects of Ca²⁺ on cPLA₂ membrane adsorption/activity cannot be distinguished from its effects on ER vesiculation and T_{INM}. For this reason, we developed ALPIN—a Ca²⁺-independent tension sensor that reports T_{INM} without cPLA₂'s Ca²⁺ dependency.

Importantly, we and others already measured cPLA₂ activity upon hypotonic swelling ((2); see also PMID 9497423 and 33060331). Furthermore, we showed that only swollen, but not unswollen HeLa cell nuclei attract neutrophils *in vivo* via cPLA₂ ((2), Fig. 6).

3- The role of membrane tension independently of that of Ca²⁺ in the recruitment of cPLA₂ to the INM has not been demonstrated *in vivo*. In this respect, experiments in zebrafish using ALPIN expression should be carried out.

> We previously showed *in vivo* is that T_{INM} is indispensable, in addition to Ca²⁺, for cPLA₂-INM recruitment, and thus activation: In zebrafish larvae, raising Ca²⁺ with an ionophore does **not** translocate cPLA₂ to the inner nuclear membrane (INM) unless hypotonic shock is applied simultaneously ((2), Fig. S2D; see also Shen et al., 2022 and Niethammer et al., 2024). Shen et al. (3) quantified this Ca²⁺/T_{INM} co-requirement in more detail, showing that T_{INM} increases cPLA₂ membrane affinity by several orders of magnitude—explaining its physiological necessity.

Figure 1. ALPIN (red) expression in larval tail fin indicative of perinuclear aggregates.

We tested ALPIN for *in vivo* T_{INM} measurements, but over-expression from mRNA injection drives the probe into perinuclear condensate-like aggregates (Figure 1), a behaviour never observed in cultured cells and possibly linked to the probes persistently exposed hydrophobic sites, and/or organism-specific factors. Creating inducible, low-level transgenic lines may resolve this, yet would require 6–12 months and its success is uncertain. Because the core point—the T_{INM} dependence of $c\text{Pl}a_2$ activation *in vitro* AND *in vivo*—is already well documented (see above), and independently

confirmed by our ALPIN-cell culture experiments (this paper), we feel that optimizing ALPIN for zebrafish is, albeit desirable in general, beyond the scope of this focused, mechanistic study.

Minor points:

4- Fig S3. It is impossible to distinguish Sytox Orange from $c\text{Pl}a_2$ -mKate2 from the images. The authors need to redo these controls, knowing that other lysis markers of different colors exist, even in the Sytox family. Quantification of results is also recommended.

> We appreciate the reviewer's suggestion and now provide timelapse analysis with Sytox blue instead of Sytox orange (e.g., Fig. 3, S1b, S2c, S3b).

5- Fig S1b, c, Fig S2c, d. Fig S5a, S5b, etc. A single representative curve is shown, whereas the figure legend mentions many more measurements. A mean curve with errors bars, or better still, all the other curves (in shaded colors, for example), should be shown. A critical example is the time lag between the recruitment of $c\text{Pl}a_2$ to the INM and the ER circularity increase in Fig S2d: this is a point discussed in the text and cannot be based on a single curve.

> We agree, and now provide multiple sample integration for all our statements. Nuclear compression data have been removed, since they were tangential to the focus and methods of this paper.

6- Fig S4i. GUV measurements must be displayed in detail by showing GUV-binding isotherms and fitted lines.

> Yes, we now provide the requested data in Fig.S2a

7- Fig 2. ML162 experiments. More time series should be shown in fig 2b, c as only one measurement is displayed. The authors wrote that T(INM) using ALPIN and eGFP-KDEL were concomitantly monitored, but Fig2 suggests that measurements were performed on different cells. No time series is shown for cPla2 or ALPIN recruitment to the INM.

> We agree that the lack of time-lapse data was a weakness in the previous version, and we fixed this now. To clarify, our ER morphology analyses are performed on the same fields of views (FOVs) from where the cells for the ALPIN analysis are derived. We just did not find a reliable way to segment individual cells based on the eGFP-KDEL channel, which is why the ALPIN data is based on nuclear segmentation, whereas the ER circularity and area analyses are based on FOVs. We better clarified this point in the revised version.

8- Fig 3. The authors have measured rate of ALPIN sensor binding to the INM, but experimental method is missing and the temporal resolution of their timelapse microscopy measurements is not specified. This is important to get an idea of measurement accuracy. Are units missing from the rate values shown in the tables of fig 3?

> The adsorption rates were determined by the Ordinary Least Squares of the linear portion of the binding plots. We updated the methods. The temporal resolution is 1 frame/min. Adsorption rate unit: min^{-1} . The steep increase reflects the ALPIN adsorption in the first frame acquired after manual mixing. Extensive mixing is required given the high viscosity of PVP360 solution, but inevitably limits the time resolution of the experiment. More details are available in our Experimental Registration Sheet, which connects numerical data analysis with raw data.

9- Finally, I believe that some of the cartoons/panels in the supplementary figures could be switched to the main figures. I felt I spent more time looking at the sup figures than the main figures, so a rebalancing might be advisable.

> We completely agree and apologize for the inconvenience. The former organization of the figures was not optimal and confusing, which we fixed in the revised version.

Reviewer #2:

This is a very interesting but brief submission that proposes a newly designed "Ca²⁺ independent, intranuclear lipid packing probe (called) ALPIN".

> We very much appreciate the reviewer's interest.

Decades of past work showing phospholipase binding to dilated membranes has been nicely extended in previous work of the Niethammer lab to the inner nuclear membrane.

However, a persistent question has been the role of the ER that is contiguous, and so they also propose here to address this with their new probe "in the presence or absence of a contiguous ER" (sketched in Fig.4b). Although I am unconvinced that this latter issue is adequately addressed,

> We acknowledge the implicit critique (underlined), which is reminiscent of rev #1's first comment. We experimentally addressed it by super-resolution imaging and FLIP experiments in the revised version (**Fig. 1c-d**).

some aspects of the work seem foundational to an advance for the broader topic.

> We appreciate that the reviewer noticed and explicitly mentioned the fundamental aspects of our work.

In particular, the following concerns temper enthusiasm.

1. Overall rigor for reproducibility requires attention throughout.

> Yes, so we have improved the data presentation throughout the paper, and now consistently provide statistical integration. Further, all numerical data pertaining to the individual subfigures are now provided in an "Experimental Registration Sheet", which connects the depicted data with numerical analysis and raw data storage location.

In addition, Zaza Gelashvili has become a first author, because he independently reproduced all key aspects of the work with new timelapse experiments. We consider independent reproduction by a different researcher as gold standard. These additional efforts hopefully underline how serious we are taking any reproducibility concerns in my lab.

For a start, Fig.1 needs more calibration between isotonic conditions and 90% dilution of isotonic (such as 30%, 60%, and then 90%). Based on Supplement Figs for confinement: What were the nucleus heights before and after the compression, and did all cells survive?

> We now provide an ER circularity time-lapse analysis including 220, 270, and 303 mOsm osmotic shock, which shows that ER vesiculation is elicited by $\Delta\Pi > 220$ mOsm. We removed the cell compression experiments, because they are methodologically tangential to our story. Anyways, all cells survived the compression for the time of acquisition.

Nevertheless, when cells are stressed, e.g., by critical compression or swelling, their long-term survival is likely to be affected, as the reviewer noted. As a main physiological function of the NMMT-cPl₂ actually is to alert immune cells to such affected cells, the question is: for how long before cells eventually lose PM integrity is T_{INM} present to drive danger signaling? Our osmotic perturbation long-term timelapse experiments now show that osmotically shocked cells ($\Delta\Pi=270$ mOsm) remain alive for several hours, even under persistent osmotic stress, whereas the first reversible NMMT responses are observable already a few minutes after stress onset (see e.g., **Fig. S1b & S2c**). So, cell death and initial NMMT are temporally well separated.

2. Rationale: the authors need to provide a summary in the Intro of pathophysiological conditions where osmotic stresses change so drastically for human cells - given the studies here of human U2OS osteosarcoma-derived cells. This is because most if not all of the studies here are done in extreme limits relevant perhaps to wounding of some fish cells in fresh water (which is low Osm at 5-50 mOsm). Some comments in the text about pH buffering are also needed.

>Thank you for raising this point. We omitted a more detailed discussion of this issue because we have previously discussed it many times before in our primary research papers and reviews. We cited some of these papers (2,4), and now tried to better clarify the role of NMMT as death-transcendent

danger signaling mechanism. If this does not suffice, we are of course happy to include an additional explanation paragraph, such as the one below.

Briefly, in mammals, several epithelia are naturally bathed in fresh-water-like fluids ($\approx 10\text{--}50$ mOsm) (5). Saliva and the luminal fluid of the oral cavity and oesophagus, for example, fall in this range; thus any breach of these surfaces exposes underlying cells to hypotonic shocks comparable to those used here (Enyedi et al., 2013). Even steeper colloid-osmotic drops occur when a plasma membrane ruptures: cytoplasmic proteins escape, extracellular oncotic pressure collapses, and water rushes into the nucleus and swells it. Before lysis, necrotic cell swelling occurs without any external osmotic shock, in part caused by progressive ion balance dysregulation (6). Therefore, necrosis is also called 'oncosis' (from onkos, swelling), based on its most pronounced morphologic hallmark: cell swelling despite extracellular isotonicity.

L-15 medium is buffered by phosphates as stated in the methods. In the permeabilization experiments, pH is well calibrated by NaOH to 7.3-7.4. as stated in the methods.

3. Rigor of kinetics: The authors use Sytox Orange for plasma membrane rupture, but it should be compared to how fast can rupture be detected through loss of some freely diffusing cytoplasmic probe such as GFP and also cytoplasmic entry of a low MW dye? Sytox Orange could be slower as it must enter the cell, then the nucleus, and bind DNA.

>Our new time-lapse experiments now address this point. We observe loss of cytoplasmic ALPIN (that accumulates after nuclear leakage in the cytoplasm) concomitant with an increase of the nuclear Sytox blue signal. As expected, Sytox entry and cytoplasmic fluorescent protein loss occurs simultaneously (**Fig. S2c**, bottom-left, **S1b**, bottom, **Fig. 3a**, bottom-left), confirming the reliability of the Sytox readout.

4. Rigor in claim "Ca²⁺ independent": In particular, do the cells have any residual calcium in Fig.3b (e.g. by elemental analyses)? The high negative charge of DNA and the well-known presence of Ca²⁺ in the nucleus add complexity. If there is any residual Ca in the key studies of Fig.3b, then the differences could reflect this, so that the ALPIN sensor remains sensitive to Ca inside cells (even if not sensitive in vitro with vesicle studies).

> We appreciate the skepticism. We suspect that we have not been clear enough in discussing our experimental design/logic. To clarify: ALPIN contains no known Ca^{2+} -binding sites and consequently we find that it is not Ca^{2+} sensitive (Fig. S2a). What Ca^{2+} modulates in the Fig. 2g (former 3b) is ER–nuclear-membrane contiguity, not ALPIN's per se affinity for the inner nuclear membrane; this is why we developed ALPIN in the first place. Residual, nuclear Ca^{2+} , if it exists, should not have any bearing on these results. We tried to further clarify the issue in the revised version.

5. Rigor in functional assessment: Is the ALPIN sensor accumulating in nucleoli? If not, then what are the nuclear depots or accumulations, and are those dependent or even competing with the effects of NE membrane dilation or calcium?

>This is a very interesting point. We now show and colocalization of ALPIN with a nucleolar marker. Nucleoli and ALPIN disappear with osmotic shock as previously reported (7,8). cPLA_2 does not show nucleolar localization, and ML162 treatment, unlike osmotic shock, does not cause nucleolar dissipation. Please also note that in permeabilized cells, ALPIN's nucleolar localization does not change, yet it binds to the INM upon colloid nuclear swelling. Thus, there is no consistent coupling of sensor-INM adsorption and its nucleolar localization/dissipation.

The localization itself, however, is indeed interesting and probably related to the previously described nucleolar sequestration of unstructured domains to prevent their aggregation (9). Conceivably, ALPIN's constitutively exposed hydrophobic domains make ALPIN more prone to aggregation than cPLA_2 , whose Ca^{2+} regulation prevents constitutive exposure hydrophobic residue exposure.

Furthermore, for Fig.1e: Does ALPIN compete for NE binding with cPLA_2 ? Does more ALPIN binding occur with partial knockdown of cPLA_2 ? If competition is not detected, then do they bind different membrane sites?

> We already showed that ALPIN and cPLA_2 compete for the same lipid-packing–defect sites on membranes (3); the two probes therefore recognise identical INM targets. In U2OS cells endogenous cPLA_2 seems to be not abundant enough to mask ALPIN's sensor function: the sensor still gives a robust, quantifiable signal. Partial knock-down of cPLA_2 would be expected to raise ALPIN intensity modestly, but this would only improve sensitivity—

importantly, masking does not cannot create a false-positive read-out and is thus not a concern for our measurements. Other C2/PLAT-domain proteins (e.g. ALOX5/12) could in principle compete as well; their relative abundance will dictate the exact occupancy, yet none of our mechanistic conclusion should be affected by this.

6. Relevance to death or to nuclear mechanosensing of viable cells is unclear. In particular, Fig.2,S3b seems designed to chemically drive death of U2OS cells (by ferroptosis with an inhibitor of glutathione peroxidase), but the kinetics of ER changes and cell death require greater clarity. Sytox orange staining needs to be shown in a different color in the images, but more important is the need to conduct similar experiments with other markers of death.

> We absolutely agree, the lack of long-term kinetics was a weakness of the previous version. We fixed this now by imaging ER morphology, nuclear volume, biosensor-INM adsorption, and Sytox blue over several hours. Sytox dyes are perhaps the most established markers of PM integrity loss, and cell death. In addition, we now show that cytoplasmic ALPIN or cPLA₂ leaks out of cells at the same time as Sytox starts staining the nucleus (Fig. S2c, bottom-left, S1b, bottom, Fig. 3a, bottom-left). The fact that two independent PM integrity readouts behave in the same way, underlines the validity of our cell death assessment.

The broader question raised is whether binding of ALPIN (and maybe even cPLA₂) to the nuclear envelope can occur in cells that are fully viable (can they divide). In other words, is ALPIN a death sensor for the nucleus loosely analogous to Annexin for the plasma membrane?

> ALPIN and cPLA₂ behave as sensors of acute mechanical/ colloid-osmotic stress to the nucleus, not as markers of cell death. In U2OS cells exposed to a sustained 270 mOsm hypotonic shock, both probes accumulate on the INM ~2-4 hours before the first cells become Sytox-positive; once cell volume recovers through regulatory volume decrease, they dissociate and the cells assume largely normal morphology, indicating that recruitment occurs while plasma-membrane integrity is completely intact (Fig. S1b,c; S2). The same reversibility (just faster) is seen in vivo: during zebrafish tail-wounding, cPLA₂ is rapidly and transiently recruited to the nuclear envelope, drives a brief burst of leukocyte recruitment and vascular leakage, and then disengages as repair proceeds, all without detectable cell death (2,4,10). These

observations, together with the available cPLA₂ knock-out phenotypes that point to defects in inflammation and mechanoadaptation, show that T_{INM} (as detected by via cPLA₂ and ALPIN) acts as an early, physical “danger signal” triggered when physical stress exceeds a cell’s adaptive limits but before irreversible damage occurs, unlike annexin V, which –to our best knowledge—labels membranes only after rupture. Given its physiological role as danger signaling mechanism, of course, NMMT and cell death are expected to be coupled to some extent. But INM adsorption itself does not rely on cell death and is thus not a reliable cell death marker (in the usual sense of the word). Our new data makes this clear.

7. Novelty of findings with ER, and related to the previous point: Fig.1a is reminiscent of ER stress, which can eventually lead to death [PMID: 23850759, <https://doi.org/10.3389/fcell.2021.767866>]. Based on reports of massive cell death upon hypotonic exposure such as in [<https://doi.org/10.3389/fphys.2020.582781>], the authors need to provide longer term assessments of death. Also, ER stress drugs such as Thapsigargin (Tg) should also be tested. Lastly in this context, why does ER fragment under compression? One can certainly squash cells to death, and so long-term viability again needs careful quantitation.

> We have now added extended time-lapse sequences that follow ER morphology, ALPIN/cPLA₂ recruitment, and Sytox influx for 6 h after hypotonic shock, showing that vesiculation and nuclear-membrane tension rise well before any loss of viability and fully reverse through regulatory volume decrease. It has been already reported that thapsigargin, just like ionomycin, triggers ER vesiculation (11).

We did not intend to present ER vesiculation as a new discovery, and we apologize if this was the impression. The novelty of our work lies in testing the biomechanical crosstalk between the ER and the nucleus during mechanotransduction. To our best knowledge, our approach is unprecedented. We exploit the vesiculation response as a perturbation method to test whether the contiguous ER membrane reservoir limits nuclear-membrane mechanotransduction. Disrupting that reservoir amplifies NMMT, resolving a long-standing, fundamental objection to the NMMT concept. Namely, sceptics believed that an intact ER should completely prevent NMMT. Our work refutes this view.

We removed the cell compression data, because they were tangential to the paper's focus, and seemed to add confusion. We suppose that the ER vesiculates under compression for the same reason it fragments under hypotonic shock, ionomycin, thapsigargin: Ca²⁺ elevation disrupts ER contiguity (12,13), as also directly shown by our work (Fig. 2e).

8. Rigor in analyses of Fig.3B: are images collected every minute or more often? Does that frequency affect the rates in the Tables? What are the units for rates, and do the rates show a significance dependence on indicated Ca levels? Does this support or not the original claim of "we developed the Ca²⁺ independent, intranuclear lipid packing probe ALPIN"?

>Images were collected every minute, and the adsorption rate unit is min⁻¹ (now added in the revised figure). Further acquisition details are available in the ERS that details numerical data and raw data storage.

We apologize, if we were not clear enough in explaining the logic of this experiment: ALPIN-INM interactions are Ca²⁺ insensitive (Fig. S2a), because ALPIN does not have Ca²⁺ binding sites like cPLA₂. The increased ALPIN signal after 50 μM Ca²⁺ represents vesiculation-dependent T_{INM}.

9. Relevance to mechanosensing or death is again raised in Fig.4. This is because wounding does cause some cell death [doi.org/10.7554/eLife.86269], but that might well require longer times than studied. The authors need to broadly and rigorously address this fate in this model and across their studies, especially given the resemblance of ER changes here to cited ER stress.

> Our focus is on the danger-signal phase that begins within seconds of wounding, when T_{INM} pulls cPLA₂ to the inner nuclear membrane and triggers arachidonic-acid release in cells both at and beyond the wound margin. This biochemical alarm is what directs inflammatory (4) and vascular responses (10); what those cells may do hours later is immaterial to that primary function. Indeed, cPLA₂ activity continues—and may even intensify—after plasma-membrane rupture because pressurised nuclei retain T_{INM}. We have shown that such “death-transcendent” signalling operates from living and already-lysed cells alike (2,14). The present work extends this concept by demonstrating that ER vesiculation stabilises T_{INM}, thereby prolonging cPLA₂ engagement with the nuclear envelope. Thus, although ER morphology may

resemble ER stress, the main mechanistic point is that vesiculation amplifies nuclear membrane mechanotransduction; ultimate cell fate does not alter this conclusion. Nevertheless, we now provide long-term multiparametric imaging for our cell culture experiments to follow cell fate from initial perturbation to cell death.

10. Rigor and interpretability for the two phases of response such as in Fig. 1b & S4d-g: More images of later timepoints are needed for all. What are the decay times, and what sets these times?

> We have replaced most endpoint measurements with long term, time-lapse series (new Fig. 1b, 2b, S1c, S2d–e). These data show that ER vesiculation and cPLA₂/ALPIN recruitment rise together during the swelling burst and return to baseline within ~15 min of the volume peak. The decay kinetics track the regulatory-volume decrease (15), which expels ions and water, relaxes nuclear-membrane tension, and causes both the ER and the probes to recover synchronously. The revised manuscript now makes this two-phase sequence—and the role of RVD in setting the decay time—explicit.

11. Rigor and reproducibility in Fig.S4i: show the binding isotherms and show them for different sized vesicles to address the proposed curvature dependence of Fig.S4b.

> The revised Fig. S2a now presents full binding isotherms on giant unilamellar vesicles (GUVs). We chose GUVs because, at 10–20 μm in diameter, they match the nearly flat geometry of the nucleus; at this scale membrane curvature does not generate appreciable lipid-packing defects, and curvature is therefore irrelevant to the nuclear mechanotransduction addressed in this study. We tried to make this more explicit in the text.

ER vesiculation and Ca²⁺ elevation change nuclear membrane tension, not curvature, so additional measurements on smaller, highly curved vesicles would probe a parameter that the nuclear envelope simply does not experience. We acknowledge that integrating curvature with tension sensing is an interesting extension for future work, but it lies outside the biological context examined here.

12. Lastly, returning to "in the presence or absence of a contiguous ER" (sketched in Fig.4b), the authors need to summarize in this final figure (or

its legend) exactly which Figure panels support the left cartoon and comment that the rest of the results support the right cartoon (i.e. absence of ER connections).

>We prepared a revised summary figure 3f to better summarize our experimental results.

Minor:

1. References 1-5 are from two groups including the Niethammer lab, and they do not adequately reference past work on the listed topics of differentiation, ECM, etc.

>The entire intro has been rewritten to highlight the central physiological role of NMMT in critical cell deformation sensing, instead of differentiation/ECM topics. In any case, we are happy to add any additional references that the reviewer feels is important.

2. Much earlier work than Ref.7 is more appropriate to cite, such as: "Regulation of phospholipase A2 activity by the lipid-water interface" Biochemistry. 1979. The authors need to provide a better summary with historical references of phospholipase activity tension/density regulation at membrane interfaces.

>Thank you. We included this classic reference into the intro along with some other background references about tension dependent protein adsorption to membranes.

3. "Nuclear shape sensing" should probably be replaced with nuclear area sensing (or NE area sensing). For example, a sphere and ellipsoid differ in shape, but small and large spheres do not differ in shape.

> We replaced the term shape sensing with nuclear membrane mechanotransduction (NMMT) to more explicitly state what we are studying.

Reviewer#3:

Due to the high quality of the data presented I only have one major comments:

>Thank you very much for your encouraging assessment.

1) The data derived from imaging of live zebrafish embryos would strongly benefit from more quantification. Highly convincing representative images are presented, but there is little information on reproducibility. Figure legends mention that translocation with ER fragmentation was observed in 5 wounding experiments, and translocation without fragmentation in 4, but it remains unclear how many cells were observed. Likewise, the correlation between these cases and the distance from the wound site is only anecdotal. Thus, authors should more clearly show how many cells, stratified by distance from the wound site, show translocation with and without ER fragmentation.

>The entire paper is improved and rewritten. All n=1 graphs are removed. All major results have been independently reproduced by Zaza Gelashvili (now co-first author) in long-term timelapse format.

Re distance relationship of wound and translocation. Please see (4) Fig. 2b, d. There we have shown the requested correlation.

minor comments:

1) line 3: First sentence mentions eGFP-Sec61beta, which is shown in Fig. 1c, but the following section does not refer to this figure.

2) Figure S4c: I suggest to draw the cartoons to scale, such that it's more obvious that the second construct contains the second ALPS domain.

3) Plots showing adsorption of sensors to INM: while the methods very clearly describe how this was quantified, legibility could be improved by briefly explaining in the figure legends what the "ratio" indicates (e.g. Fig. 1e)

> We think that all these points are now addressed and improved in the revised version.

References

1. King C, Sengupta P, Seo AY, Lippincott-Schwartz J. ER membranes exhibit phase behavior at sites of organelle contact. *Proc Natl Acad Sci USA*. 2020 Mar 31;117(13):7225–35.
2. Enyedi B, Jelcic M, Niethammer P. The Cell Nucleus Serves as a Mechanotransducer of Tissue Damage-Induced Inflammation. *Cell*. 2016 May 19;165(5):1160–70.
3. Shen Z, Belcheva KT, Jelcic M, Hui KL, Katikaneni A, Niethammer P. A synergy between mechanosensitive calcium- and membrane-binding mediates tension-sensing by C2-like domains. *Proc Natl Acad Sci USA*. 2022 Jan 4;119(1).
4. Enyedi B, Kala S, Nikolich-Zugich T, Niethammer P. Tissue damage detection by osmotic surveillance. *Nat Cell Biol*. 2013 Sep;15(9):1123–30.
5. Sawinski VJ, Goldberg AF, Loiselle RJ. Osmolality of normal human saliva at body temperature. *Clin Chem*. 1966 Aug;12(8):513–4.
6. Hirata Y, Cai R, Volchuk A, Steinberg BE, Saito Y, Matsuzawa A, et al. Lipid peroxidation increases membrane tension, Piezo1 gating, and cation permeability to execute ferroptosis. *Curr Biol*. 2023 Apr 10;33(7):1282-1294.e5.
7. Zatsepina OV, Dudnic OA, Chentsov YS, Thiry M, Spring H, Trendelenburg MF. Reassembly of functional nucleoli following in situ unraveling by low-ionic-strength treatment of cultured mammalian cells. *Exp Cell Res*. 1997 May 25;233(1):155–68.
8. Zatsepina OV, Dudnic OA, Todorov IT, Thiry M, Spring H, Trendelenburg MF. Experimental induction of prenucleolar bodies (PNBs) in interphase cells: interphase PNBs show similar characteristics as those typically observed at telophase of mitosis in untreated cells. *Chromosoma*. 1997 Jun;105(7–8):418–30.

9. Frottin F, Schueder F, Tiwary S, Gupta R, Körner R, Schlichthaerle T, et al. The nucleolus functions as a phase-separated protein quality control compartment. *Science*. 2019 Jul 26;365(6451):342–7.
10. Gelashvili Z, Shen Z, Ma Y, Jelcic M, Niethammer P. Perivascular macrophages convert physical wound signals into rapid vascular responses. *BioRxiv*. 2024 Dec 12;
11. Subramanian K, Meyer T. Calcium-induced restructuring of nuclear envelope and endoplasmic reticulum calcium stores. *Cell*. 1997 Jun 13;89(6):963–71.
12. Venturini V, Pezzano F, Català Castro F, Häkkinen H-M, Jiménez-Delgado S, Colomer-Rosell M, et al. The nucleus measures shape changes for cellular proprioception to control dynamic cell behavior. *Science*. 2020 Oct 16;370(6514).
13. Lomakin AJ, Cattin CJ, Cuvelier D, Alraies Z, Molina M, Nader GPF, et al. The nucleus acts as a ruler tailoring cell responses to spatial constraints. *Science*. 2020 Oct 16;
14. Chang W, Gundersen GG. Swollen Nuclei Signal from the Grave. *Cell*. 2016 May 19;165(5):1051–2.
15. Hoffmann EK, Lambert IH, Pedersen SF. Physiology of cell volume regulation in vertebrates. *Physiol Rev*. 2009 Jan;89(1):193–277.

We would like to thank the reviewers for their interest and thorough assessment of our manuscript.

Reviewer #1 (Remarks to the Author):

The authors carried out the important controls requested and integrated additional measurements for statistics when needed. They confirmed the vesiculation of the ER and the lack of continuity between NM and ER in their experimental setup, in particular through super-resolution imaging and FLIP experiments. Overall, the revised article reads well and is fairly clear.

Minor point: I am a little surprised by the choice of some images to illustrate timelapse imaging experiments. For example, cPla2 INM-adsorption during shock and during the pre-lytic phase is not convincing in Fig. S2b (e.g. at 12' and 252' compared to 0'). This does not visually reflect the quantification (red curve) shown in S2c, nor even the quantification by linescan in FigS2b middle panel. Is it possible to find images series that better illustrate the adsorption of cPla2 to the INM?

> Thank you for the scientific perspective that helped us strengthen our manuscript. We believe the referee was referring to the representative montage inset shown in Fig. S1b. In the previous version, the time points were selected to illustrate the transition phases of cPla₂-INM adsorption rather than its peak levels, which may have caused the perceived mismatch and confusion with the quantification shown in Fig. S1 and the corresponding linescan analysis. In the revised version, we have replaced these panels with images showing peak cPla₂-INM adsorption, and retained the quantification unchanged and updated the indicated time points accordingly. We hope this clarifies the confusion between the montage and the quantification.

Reviewer #3 (Remarks to the Author):

The authors have addressed all issues I had raised in a satisfactory manner. The manuscript was already very strong in the initial version and is now further improved. I strongly recommend publication in NCB.

> Thank you very much !

Reviewer #4 (Remarks to the Author):

In this revised version of the manuscript, using a novel biosensor ALPIN inside the nucleus, together with high-resolution imaging, the authors presented results showing that the ER acts as a nuclear membrane buffer in response to a hypo-osmotic stress or Ca²⁺-gated signaling. They revealed that osmotic stress-induced ER vesiculation led to disruption of the ER reservoir that enhances NMMT. In response to the comments from the three reviewers, the authors performed necessary additional experiments to strengthen the conclusion of the study. Some issues raised by the reviewers were also clarified and explained by the authors. Although the underlying molecular mechanism(s) of ER vesiculation remains elusive, as acknowledged by the authors, this study contributes to the understanding of the role of ER in nuclear membrane mechanosensing, possibly as a danger-signaling trigger.

> Thank you very much !